# EDGAR v4.3.2 Global Atlas of the three major Greenhouse Gas Emissions for the period 1970-2012.

**Greet Janssens-Maenhout[1,*], Monica Crippa[1], Diego Guizzardi[7], Marilena Muntean[1], Edwin Schaaf[1], Frank Dentener[1], Peter Bergamaschi[1], Valerio Pagliari[1], Jos G.J. Olivier[2], Jeroen A.H.W. Peters[2], John A. van Aardenne[3], Suvi Monni[4], Ulrike Doering[5], A.M. Roxana Petrescu[6], Efisio Solazzo[1], Gabriel D. Oreggioni[1]**

[1]European Commission, Joint Research Centre (JRC), Via E. Fermi 2749 (T.P. 123), I-21027 Ispra (VA), Italy

[2]PBL Netherlands Environmental Assessment Bureau, Den Hague, The Netherlands

[3]European Environmental Agency, Copenhagen, Denmark

[4]Benviroc Ltd., Espoo, Finland

[5]Umweltbundesamt, Dessau-Rosslau, Germany

[6]University Amsterdam, The Netherlands

[7]Didesk Informatica, Verbania, Italy (consultant to the European Commission)

[*]Ghent University, Belgium

*Correspondence to*: G. Janssens-Maenhout (greet.maenhout@ec.europa.eu)

**Abstract:** The Emissions Database for Global Atmospheric Research (EDGAR) compiles anthropogenic emissions data for greenhouse gases (GHG) and for multiple air pollutants based on international statistics and emission factors. EDGAR data provides quantitative support for atmospheric modelling and for mitigation scenario and impact assessment analyses as well as for policy evaluation. The new version v4.3.2 of the EDGAR emission inventory provides global estimates, broken down to IPCC-relevant source-sector levels, from 1970 (the year of EU's first Air Quality Directive) to 2012 (the end year of the first commitment period of the Kyoto Protocol (KP)). Strengths of EDGAR v4.3.2 include global geo-coverage (226 countries), continuity in time, and comprehensiveness in activities. Emissions of multiple chemical compounds, GHG as well as air pollutants, from relevant sources (fossil fuel activities but also, for example fermentation processes in agricultural activities) are compiled following a bottom-up (BU), transparent to the extent possible and IPCC-compliant methodology. This paper describes EDGARv4.3.2 developments with respect to three major GHG ($CO_2$, $CH_4$, and $N_2O$) derived from a wide range of human activities apart from the land-use, land-use change and forestry (LULUCF) sector and apart from Savannah burning; a companion paper quantifies and discusses emissions of air pollutants. Detailed information is included for each of the IPCC-relevant source-sectors, leading to global totals for 2010 (in the middle of the first KP commitment period) (with 95% confidence interval in parentheses): 33.6 ($\pm$5.9) Pg $CO_2$/yr, 0.34 ($\pm$0.16) Pg $CH_4$/yr, and 7.2 ($\pm$3.7) Tg $N_2O$/yr. We provide uncertainty factors in emissions data for

the different GHGs and for three different groups of countries: OECD countries of 1990, countries with economies in transition in 1990, and the remaining countries in development (the UNFCCC non-Annex I parties). We document trends for the major emitting countries together with the European Union in more detail, demonstrating that effects of fuel markets and financial stability have had greater impacts on GHG trends than effects of income or population. These data (DOI:10.5281/zenodo.2658138) are visualised with annual and monthly global emissions grid-maps of 0.1°x0.1° for each source-sector; these data can be freely accessed from the EDGAR website http://edgar.jrc.ec.europa.eu/overview.php?v=432&SECURE=123.

## 1.   Historical evolution

An essential component of the UN Framework Convention on Climate Change (UNFCCC, 1992) is the collection of nationally reported inventories and information on these GHG emission inventory time series. At the time the UNFCCC was drafted, the 24 members of the OECD in 1990 and 16 other European countries and Russia were considered liable of "*the largest share of historical and current global emissions of GHG*" and taken up in Annex I to the UNFCCC. These Annex I countries and the European Union[1] submit annually complete inventories of GHG emissions from the 1990 base year[2] until the latest year for which full accounting is completed and reviewed (typically with two-year time lag) and these inventories are all reviewed to ensure transparency, completeness, comparability, consistency and accuracy[3]. This allows for most of these Annex I countries to track progress towards their reduction targets committed under the Kyoto Protocol (UNFCCC, 1997). Other (non-Annex I) countries are encouraged to submit their GHG inventories as part of their National Communications and Biennial Update Reports (BUR). The GHG inventories of non-Annex I countries were required to cover $CO_2$, $CH_4$ and $N_2O$ emissions for one year (1990 or 1994), without specific documentation and only subject to a brief review. However, the Paris Agreement (UNFCCC, 2015) requests to submit every 2 years BURs[4], which are subject to international consultation and analysis. Theoretically, UNFCCC should receive at the latest after two years national emissions inventories from each of the 197 countries, but as shown in Figure 1a not all countries did provide a national inventory and 154 countries did not provide a complete time series of

---

[1] This includes the 28 Member States of the European Union (EU) as of 1st of July 2013.

[2] For some economies in transition another year, such as 1988 or 1989 can be chosen under UNFCCC as base year. These GHG emissions are mainly sources but do include also carbon stock sinks for which the human-induced part needs to be assessed with care (Grassi et al., 2018).

[3] These 5 principles of a good reporting practice are defined in the UNFCCC guidelines for national GHG inventory, e.g. https://pdfs.semanticscholar.org/3c30/a1bd769dee5299746e0af825c7ab4ed55fba.pdf. EDGAR uses the term "comprehensiveness" to summarise these principles.

[4] The first BUR submitted should cover the inventory for the year no more than 4 years prior to the submission data, and subsequent BURS should be submitted every 2 years, but flexibility is given to least developed countries and small island developing states.

inventories. In addition many countries lack a well-developed statistical infrastructure, which is needed for a bottom-up (BU) inventory. Figure 1b presents the latest year that is covered with a national inventory, which dates for quite some countries more than 10 years ago: for most South-East Asian countries this is between 2004 and 2007 and for most African countries between 2000 and 2003.

As such, the collection of national reports/communications do not provide a complete, consistent and comparable global dataset, which can be used to understand the global budgets of the most important GHG emissions and their impact on climate. Very few bottom-up inventories of global anthropogenic emissions have been produced with continued effort for more than 2 decades. The Carbon Dioxide Information Analysis Centre (CDIAC) (Boden et al., 2017; Andres et al., 2014) and the Emissions Database for Global Atmospheric Research

(EDGAR) (Olivier et al., 2016a,b) provide global totals, whereas the IEA provides $CO_2$ estimates from fuel combustion only and the FAO $CH_4$ from agriculture only. While CDIAC ceased operation in September 2017, the Open-source Data Inventory for Anthropogenic $CO_2$ (ODIAC) (Oda et al., 2018) continued to use the CDIAC data and combined these with geospatial proxies (including night light satellite maps) to provide $CO_2$ grid-maps, as also EDGAR is doing (using other geospatial proxies). In addition, the new Community Emissions

Data System (CEDS) of Hoesly et al. (2018) built upon existing inventories to provide a new gridded dataset of all emission species for the Climate Model Inter-comparison Programme CMIP6.

The scientific community started to bring together these anthropogenic BU emissions with top-down estimates covering also the natural component to obtain the Global Carbon Budget (GCB) (Le Quéré et al., 2018) and the Global Methane Budget (Saunois et al., 2016). These budgets are important input for the periodic global

stocktake that the Paris Agreement envisages from 2023 onwards (with the submitted inventories for 2021). Even though significant progress in inventory compilation has been made, the overall uncertainty of the global total has become larger over time because the share of emissions from non-Annex I countries (with less developed statistical infrastructure) increased from less than 40% in 1990 to more than 60% in 2012, as shown in Figure 2.

To support both science and policy making with the monitoring and verification of the GHG emissions, it is important that emissions are estimated by using comparable methodologies, consistent source allocation and comprehensive coverage of the globe. The EDGAR v4.3.2 global inventory illustrates the result of a bottom-up technology-based compilation of country- and sector-specific emission time series 1970-2012. Furthermore, the monthly resolution and global grid-maps at a spatial resolution of 0.1°x0.1° allow direct use in atmospheric

models as well as in analyses of policy impacts. The first version of the Emissions Database for Global Atmospheric Research (EDGAR v2) answered the needs of the air quality community to map technological parameters of air pollution sources and was published by Olivier et al. (1996). Since then, several updated versions (Olivier, 2002) were released (EDGAR-HYDE, EDGAR v3.2, EDGAR 3.2 FT2000). Driven by the development of scientific knowledge on emission generating processes and by the availability of more recent

information, the EDGARv4 datasets were constructed as new factors and additional end-of-pipe abatement measures. The specification of the combustion technology and its end-of-pipe abatement is more important for air pollutants and aerosols than for greenhouse gases. $CO_2$ combustion emissions are fuel-determined and carbon capture and storage is not yet implemented at an operational level and is not considered here. However

abatement is considered for e.g. $CH_4$ recovery of coal mines and technology and end-of-pipe abatement are important for both adipic and nitric acid plants. Finally management of crop cultivation (e.g. for rice) or of manure are accounted for by technology-specific emission factors for $CH_4$ and $N_2O$.

Previous EDGAR versions v4.1 and v4.2 (available at http://edgar.jrc.ec.europa.eu/index.php#) are interim
frozen datasets without peer-reviewed documentation, but nevertheless extensively used by modellers. Illustrative examples of the EDGARv4 use are given in Table S5 of the Supplementary Information. The new online version EDGAR v4.3.2 is the main reference for the EDGARv4 datasets, and is the subject of this paper. We wish to stress that the EDGARv4.3.2 is the result of a steady improvement of the EDGARv4 database over more than a decade, also thanks to the feedback of users. For the main differences between EDGARv4.3.2 and
v4.2 we refer to the Supplementary of the paper, section 3 and Table S5 with the findings of atmospheric studies using EDGARv4 as input. For the main differences between EDGARv4.2 and v4.1 we refer to http://edgar.jrc.ec.europa.eu/Main_differences_between_EDGARv42_and_v41.pdf.

In this paper we focus on the three key GHG emission components of EDGARv4.3.2, describing the methodology, emission sources, activity data, emission factors and emission disaggregation (in space and time).
For CO2 we distinguish between (i) long-cycle carbon $CO_2$ from fossil fuel use and industrial processes (cement production, carbonate use of limestone and dolomite, non-energy use of fuels and other combustion, chemical and metal processes, solvents, agricultural liming and urea, waste and fossil fuel fires), and (ii) short-cycle carbon CO2 from biofuel use or short-cycle biomass burning (such as agricultural waste burning). The non-$CO_2$ GHG emissions are also provided to the IEA for the annual publication of emissions from fuel combustion
(Olivier and Janssens-Maenhout, 2016b). The EDGAR v4.3.2 frozen dataset for 1970-2012 is used to produce the updates from 2013 onwards, derived with a fast track (FT) approach (e.g. EDGARv4.3.2_FT2016). Under the FT update the activities are grouped into five main source sectors and for each of the latter trends of most recent activity statistics are used. These are derived by data provided by the latest IEA (2016) and BP (2017) statistics in terms of fuel trend indicators that are applied to the fossil fuel combustion sector. For the other main
sectors we use most recent commodity statistics from the US Geological Survey, the World Steel Association and the International Fertiliser Association, as explained in more detail in Olivier et al. (2016a). The methodology and activity data are also used to estimate corresponding gaseous and particulate air pollutant emissions, as part II of the EDGAR v4.3.2 release (Crippa et al., 2018). Other EDGARv4 air pollutants inventories are EDGAR v4.3.1 (Crippa et al., 2016a; Huang et al., 2017), EDGAR v4tox1 (Muntean et al., 2014)
and EDGAR v4tox2 (Muntean et al., 2018).

## 2.    Method

### 2.1    Bottom-up emission calculation

Annual country-specific emissions are calculated using international activity data and emission factors, updated according latest scientific knowledge and following IPCC (2006) methods. Emissions (*EM*) from a given sector *i*
in a country *C* accumulated during a year *t* for a chemical compound *x* are calculated with the country-specific activity data (*AD*), quantifying the activity for sector *i*, with the mix of *j* technologies (*TECH*) and with the mix

of $k$ (end-of-pipe) abatement measures (*EOP*) installed with share $k$ for each technology $j$, the emission rate with uncontrolled emission factor (*EF*) for each sector $i$ and technology $j$ and relative reduction (*RED*) by abatement measure $k$, as summarized in the following formula:

$$EM_i(C,t,x) = \sum_{j,k} \left[ AD_i(C,t) * TECH_{i,j}(C,t) * EOP_{i,j,k}(C,t) * EF_{i,j}(C,t,x) * \left(1 - RED_{i,j,k}(C,t,x)\right) \right]$$

(1)

The activity data are very sector dependent and vary from fuel consumption in energy units (*TJ*) of a particular fuel type, to the amount (*ton*) of products manufactured, and to the number of animals or the area (*ha*) and yield (*ton*) of cultivated crops. The technology mixes, (uncontrolled[5]) emission factors and end-of-pipe measures, are determined at different levels: country-specific, regional, country group (e.g. Annex I/ non-Annex I), or global. Technology-specific emission factors are used to allow a tier-2 approach, taking into account the different
management/technology processes or infrastructures (e.g. different distribution networks) under specific "technologies", and modelling explicitly abatements/reductions e.g. the CH4 recovery from coal mine gas at country level under the "end-of-pipe measures". Just as like national inventories, EDGAR v4.3.2 starts from accounting over a period of time, one calendar year, and over a well-defined region, the country in which the emissions took place.

The sector-specific total emissions of substance $x$ for country $C$ in year $t$ are then distributed in time and space using sector- and even technology-specific monthly shares $m$ and spatial proxy datasets $f$. The proxy datasets are expressed in function of coordinates (longitude, latitude) weighted at country level and with the Heaviside function equalling 1 when the grid cell belongs to the country area according to the following formula:

$$em_i(lon,lat,t,x) = EM_i(C,t,x) \frac{m_{i,j}(C)}{\sum_{k=1,..12} m_{i,j}(C)} \frac{f_{i,j}(lon,lat,t)}{\sum_{lon,lat} \left( f_{i,j}(lon,lat,t) \cdot H(C,lon,lat) \right)}$$

$$with\ H(C,lon,lat) = fraction\ of\ gridcell\ within\ C$$

(2)

While the monthly shares are more specified in a generic way (only varying with the latitudinal band and with the sectors), the spatial proxy datasets take into account point-source information at sub-sector level (facilities) that can change from year to year.

## 2.2 Sector definition and data sources

Table 1a provides a structured overview of all the emission sources included in the EDGARv4 database. The
25 energy-related sector (with largest share in total GHG emissions) require less detailed information on "technologies" than the agriculture- and waste-related sectors do require on the "practices" applied[6]. This

---

[5] Uncontrolled means without end-of-pipe abatement

[6] $CO_2$ emissions depend on the total mass and carbon content of the fuel and not much on the type of combustion technology, while CH4 emissions depend strongly on the types of fermentation processes in addition to the total mass and composition of the decomposing organic matter.

imbalance of the requirements for a higher level of detail for less important sources in terms of contribution to the national total is against the normal expectation (and time efficiency) of expanding more efforts on those sources having largest impact on the national totals (Pulles, 2018).

All the sources of Table 1a defined under the sectors and codes used in the IPCC (1996) guidelines, Chapter 1 of Vol. 1 Reporting Instructions and converted into the new IPCC (2006a) guidelines, Chapter 8 of Vol.1 Guidance and Reporting are considered, except the Land-Use, Land-Use Change and Forestry (LULUCF) sector.[7] Contrary to the other sectors, LULUCF is not covered by annual, statistical assessments of the goods ("trees") but needs geographical and/or remote sensing information as AD. For the emission sources and sinks related to carbon stock changes in the subcategory "Forest-land-remaining-forest-land", we refer to Petrescu et al. (2012), and for the large-scale biomass burning (including forest fires, Savannah burning, grassland and woodland fires), we refer to GFED (Van der Werf et al., 2010), GFAS (J. Kaiser et al, 2012) or FINN (Wiedinmeyer et al., 2011).

Most AD for the EDGAR v4.3.2 are taken from international statistics, screened for completeness and consistency by EDGAR routines, removing outliers (clerical errors, wrong units) and gaps in time (missing single year) with a linear interpolation of the previous and the following year. Preference is given to international statistics such as those of IEA (2014) and FAOSTAT (2014) over regional offices, such as EuroStat or national statistical bureaux in order to profit from international definitions (e.g. for fuel types by IEA), inter-comparability amongst countries and the data quality and control by IEA or FAO. For China and USA, national data from the Chinese Bureau for statistics and the US Energy Industry Administration respectively are consulted to assess and fill possible gaps in AD with consumption of fuels (fossil and bio) and of products (mainly metals, non-metallic minerals such as cement, chemicals, solvents). For EU28, the biofuel statistics of EuroStat are used as they are updated faster than the IEA fuel statistics.

Where possible, GHG emission factors are selected from IPCC (2006b) to ensure consistent and complete time series, which are comparable across countries. The representativeness of default emission factors and the effectiveness of implemented control measures for the different regions are assessed based on expert judgement and consulting annual Inventory Reports of Annex I countries to the UNFCCC (2014, 2016) or National Communications and Update Reports from some most important non-Annex I countries to UNFCCC (2014, 2012, 2017). Clean Development Mechanism projects (UNEP DTU, 2011) are taken into account in non-Annex I countries to account for abatement measures of $CH_4$ and $N_2O$ emissions via $CH_4$ recovery from coal mining and landfills and $N_2O$ reduction in nitric and adipic acid production.

Industrial process emissions have been calculated with the minerals statistics of the US Geological Survey (USGS, 2014). For the agricultural activities we consulted EU's Common Agricultural Policy Regionalised Impact (CAPRI) model and derived implied (weighted) emission factors which are representing country-specific technologies and practices. For the waste sector we applied the IPCC First Order Decay (FOD) model of IPCC (2006c) that is driven by the annual per capita generated municipal solid waste, the fraction deposited in

---

[7] EDGAR includes autoproducer emissions in 1A1a and not in the industrial sector where they are generated.

landfills, and the fraction degradable organic carbon for the solid waste disposal emissions, whereas the chemical and biochemical oxygen demand are used to calculate the wastewater emissions.

Table 1b details the applied sector-specific and, where needed, region-specific data sources (activity statistics and emission factor with model parameters) on fuel balances, traded industrial products, crops, livestock and waste. For the agriculture and waste more detailed description with the model parameters is given under the "emission factor" heading. EDGAR v4.3.2 aims to collect all underlying human activity statistics and not to model the emissions directly in function of the income, population or other proxy data. Table 1b uses the same main categories of Table 1a: energy, fugitive, industrial processes, solvents and products use, agriculture, waste and other (indirect N2O emissions and fossil fuel fires). All emissions data can be downloaded also at subcategory level and are unambiguously identified with the IPCC (1996) code.

### 2.3    Temporal profiles for the monthly distribution of the annual emissions

The legal reporting obligations under UNFCCC require time series of annual inventories, in line with the output of most national statistics infrastructures with accurate, annual accounting. For the atmospheric models, a higher temporal resolution is essential. Temporal profiles in EDGARv4 have been developed in 2010 under the European Commission's 6[th] Framework Programme Research projects CIRCE (Climate change and Impact Research: the Mediterranean Environment)[8], because the global air quality models needed monthly disaggregated air pollutant emission gridmaps as input. The temporal profiles are a bottom-up estimate of the monthly variations for major sectors, based on the insights of regional air quality models. Recently, the temporal profiles have been revised and extended, as documented by Crippa et al. (2019).

Table S4a summarizes the sector specific monthly profiles applied to the aggregated sectors for each GHG in the northern hemisphere. The largest variation is found in the temporal profiles for the agricultural sector (see Fig. S2a in the Supplement), then in the emissions from residential heating and the smallest variation is present for the road transport and power generation sector. Covering regions from all over the world, a reverse profile is applied to the southern hemisphere, reflecting the opposite seasonality. No seasonal pattern is used for the equatorial region, defined within the range of [30°S, 30°N] latitude. For more refined time profiles (hourly) and in-depth analysis of the temporal distribution we refer to Crippa et al. (2019). Comparison of the EDGAR v4.3.2 monthly profiles and those used for other global emission products (Andres et al., 2011; Hoesly et al., 2018; Janssens-Maenhout et al., 2015) is given in Figure S2b.

### 2.4    Proxy data for the spatial distribution of the country total emissions

For visualisation and as an input to atmospheric chemistry transport and climate models, the EDGAR v4.3.2 database distributes anthropogenic pollutant emissions over a uniform, global 0.1°x0.1° grid defined with lower left coordinates. In emission inventories the emissions can be emitted either from a single point source or

---

[8]https://www.cmcc.it/projects/circe-climate-change-and-impact-research-the-mediterranean-environment, dataset document under Pozzer et al. (2012)

distributed over a line source (e.g. roads) or over an area source (e.g. agricultural fields), depending on the source sector or subsector. The line and area sources are distributed over the grid cells with the proxy data covering the globe entirely or partially, whereas the point sources are allocated to individual grid cells and reported as the area average of the sum of the point sources for that grid cell.

The proxy datasets that are used to grid different sector-specific sources are given in Table S4b of the Supplement. A detailed description is available in the EDGAR gridding manual (Janssens-Maenhout et al., 2013).

The spatial grid-maps are graphical representations of the country totals, making use of spatial proxy data. EDGAR tries to allocate as much as possible the human activity to the places where it is likely located: on the
place of the industrial facilities (several point source databases), or using the road network, or the housing. Alternatives can be night light satellite data, as used by Oda et al. (2018) for those emission sources that were not yet covered with point sources (such as power plants) or the population data as proposed by Andres et al. (2016). We feel our proxy data more in line with our BU approach of allocating the sectoral emissions to the place of the emission source. We do not recommend an uncertainty analysis of the proxy data itself, but a sensitivity
assessment of the representativeness of the selected proxy data using atmospheric transport modeling. EDGARv4.3.2 gridmap uncertainties are currently subject of scrutiny and further investigated under the European (Horizon 2020) research projects CO2 Human Emissions (CHE, https://www.che-project.eu/) and Observation-based System for monitoring and verification of greenhouse gases (VERIFY, http://verify.lsce.ipsl.fr/).

**2.5    Uncertainty assessment of the greenhouse gas emissions**

Uncertainties associated with emission of greenhouse gases stem from several sources, broadly described in Vol.1, Chapt.3, section 3.1.5 of the IPCC (2006) Guidelines. The uncertainties in this section are those caused by 'statistical random sampling error', that can primarily be thought of epistemic nature (lack of knowledge, thus reducible by gathering more data) but also including an aleatory component (uncertainty due to intrinsic
randomness, therefore uncompressible) (see e.g  Beven et al., 2016). As already pointed out by, e.g., Gütschow et al. (2016), the heterogeneity of reporting, lack of documentation, differences, ranges of uncertainties, sector aggregation, all factor to make difficult to compare, compile, and combine the multiple sources of information and to convey to a robust, coherent, estimate of uncertainty.

This section presents an analysis of the relative uncertainty per country grouping and gas, calculated using the Eq
(4) and the parameters reported in Table 2, which also identifies a few countries as examples of GHG emissions reporting. (expressed in $CO_{2eq}$)[9]

$$\sigma_{GHG} = \frac{\sqrt{(\sigma_{CO2}EM_i(CO_2))^2 + (\sigma_{CH4}.25.EM_i(CH_4))^2 + (\sigma_{N2O}.298..EM_i(N_2O))^2}}{EM_i(CO_2) + 25EM_i(CH_4) + 298EM_i(N_2O)} \tag{4}$$

---

[9] following the standards of the European Commission's policy documents

In accordance to IPCC tiered approach to infer uncertainties to emission factors as well as to activity data, the analysis here assumes that countries adhering to the 24 member countries of the OECD in 1990 (24OECD90)[10] were economically stable and thus they would already have, or be able to, build a good statistical infrastructure and have the lowest uncertainties in their inventories. On the same line, the 16 countries with Economies in Transition of 1990 (16EIT90)[11] have experienced greater economic instability, and their inventories are more uncertain than those of the 24OECD90, but less uncertain than those from the other remaining non-Annex I countries. Exceptions to the country grouping are made for the following new or historic trading nations, China, Russia and India, because of global proliferation of emission-regulated goods, as Crippa et al. (2016b) analysed for air pollution.

All uncertainties are reported in Tables 3, 4 and 5 within twice the standard deviation ($\sigma$) of the mean value, corresponding to a 95% confidence interval of the sample. This is a larger uncertainty range than the $\pm 1\sigma$ selected by the Global Carbon Budget 2017 (Le Quéré, 2018), but in line with IPCC recommendations. For comparative shares and trends in biofuel or non $CO_2$ GHG emissions, data on gases and sources are much more uncertain than for fossil fuel CO2. While Denier van der Gon et al. (2015) indicate that the biofuel combustion activity (and corresponding short cycle carbon CO2) is difficult to estimate for the different countries in Europe, Tian et al. (2015) estimate the large uncertainties in $CH_4$ and $N_2O$ budgets. The uncertainties in these emissions are caused by the scarcity and limited accuracy of the corresponding international activity statistics combined with the use of less representative country-wide emission factors (Olivier, 2002; Olivier et al., 2010). Using Eq (4) the uncertainty estimate in the global total anthropogenic $CO_2$ emissions is of $\pm 9.0\%$, that is slightly higher than the estimate of 8.4% by Andres et al. (2014), most probably since EDGAR v4.3.2 also includes the highly uncertain waste incineration, urea and liming activities (IPCC (2006a) reports an uncertainty associated with the default emission factors for $CO_2$ of 40%, for waste incineration), which are not part of the analysis by Andres et al. (2014).

$CO_2$ uncertainty can vary significantly among countries (Marland et al., 1999; Olivier et al., 2014). Larger uncertainties of about $\pm 15\%$ are obtained for non-Annex I countries, whereas uncertainties of less than $\pm 5\%$ are obtained for the 24 OECD90 countries for the time series from 1990 (Olivier et al, 2015) reported to UNFCCC. For emissions of $CH_4$ and $N_2O$, we estimate uncertainties of $\pm 32\%$ and $\pm 42\%$, respectively, for 24OECD90 countries and $\pm 57\%$ and $\pm 93\%$ for the other countries. These are based on the default uncertainty estimates of IPCC (2006) and in line with Bun et al. (2010). These are higher than the estimates of $\pm 25\%$ and $\pm 30\%$ by UNEP (2012) but justified by the large uncertainties reported by Tubiello et al. (2015) for the FAO activity statistics of $\pm 30\%$ and $\pm 50\%$ for crop and livestock.

---

[10] Australia, Austria, Belgium, Canada, Switzerland, Germany, Denmark, Spain, Finland, France, United Kingdom, Greece, Ireland, Iceland, Italy, Japan, Luxembourg, The Netherlands, Norway, New Zealand, Portugal, Sweden, Turkey, USA

[11] Bulgaria, Belarus, Cyprus, Czech Republic, Estonia, Croatia, Hungary, Lithuania, Latvia, Malta, Poland, Romania, Russia, Slovakia, Slovenia, Ukraine

As for the uncertainty of the emission grid-maps, Fig. 9 of Andres et al. (2016)[12] reports a population map's uncertainty in excess of 150% for Europe, Western USA, China, etc. Such uncertainty, when propagated into the emission calculations will likely outweigh the combined uncertainty of activity data and emission factors, especially for CO2. Also according to Hogue et al (2016) is the uncertainty on CO2 mapping "with 1°x1° grid cells for the United States is typically on the order of ±150%". In light of the high impact of spatial proxy on the overall uncertainty, the authors wanted to focus on a complete uncertainty assessment of the emission grid-maps in collaboration with the atmospheric modelling community, evaluating carefully a useful covariance matrix, and refer to the ongoing sensitivity assessments in the CHE project mentioned in section 2.4. Observation-based verification of European $CH_4$ and $N_2O$ emissions using inverse modelling (e.g. Bergamaschi et al., 2015; 2017) indicates that the relatively low uncertainty estimates for some countries are not consistent with the relatively large uncertainty estimates of others, and for $CH_4$ a common uncertainty band in the upper range is considered more appropriate.

### 3. Results bottom-up versus top-down

The atmospheric composition can be addressed either top-down (TD), using atmospheric composition (and measurements, such as total column measurements by satellite imagery), or bottom-up (BU), summing up the different emissions released by the different activities. Both are needed and complementary to each other: BU estimates relate to drivers and are of prime interest to policy makers, whereas TD estimates relate to observations. The two approaches have atmospheric transport models in common as link and allow to cross-check the consistency between the two approaches. Several assessments have been carried out: in the air quality community (e.g. Solazzo and Galmarini, 2015) as well as in the carbon cycle community under the Global Carbon Project with the GCB of Le Quéré et al. (2018) and the global methane budget of Saunois et al. (2016). EDGARv4.3.2 only focuses on the BU calculated anthropogenic part of the emissions and gets only posterior feedback on the use of the datasets by atmospheric modellers on the grid-maps of which examples are listed in Table S5. Although the posterior feedback on the prior emission gridmaps is very useful, it remains limited because of the uncertainties related to the transport model, the atmospheric chemistry model, the meteorology input and the in-situ or space-borne observations. However, the use of the emission grid-maps indicated the sensitivity of the emission grid-maps to the choice of spatial proxy data. The spatial representativeness needs to be checked by measured data, such as from remote sensing (e.g., Yu et al., 2017). This was so far most successful for air pollutants: $NO_x$ (Ding et al., 2017), $SO_2$ (Liu et al., 2018), CO (Hooghiemstra et al., 2011) and $CH_4$ (Bergamaschi et al., 2015, Saunois et al., 2017). In the 2018 the GCB used the spatial patterns of the EDGAR grid-maps and might give feedback in the future.

---

[12] Note Andres et al. (2016) limited the result by saying Case for CDIAC

### 3.1 The global CO2 budget

Table 3 summarises the main features of eight global $CO_2$ atlases and/or inventories: EDGAR v4.3.2, GCB (Le Quéré et al., 2016, 2018), the PKU-FUEL (Wang et al., 2013) and the ODIAC2016 (Oda et al., 2018; Oda and Maksyutov, 2011), CDIAC (Andres et al., 2014), EIA (2014), IEA (2014) and BP (2017) in temporal and spatial characteristics, sector break-down, methodology and $CO_2$ totals for major source categories in 2010 (which, for PKU-FUEL, was approximated by the latest available year, 2007).

Despite the substantially different levels of detail for the fuel use calculations, the global totals are relatively similar. At global level the differences in $CO_2$ emissions between IEA (2014) and EDGAR v4.3.2 are around 4%, which can be explained largely by the difference in overall emission factors used (differences due to different default values for the carbon content and oxidation factors in IPCC (2006) and IPCC (1996)). This yields 2%, 1% and 0.5% higher $CO_2$ emissions from coal, oil and gas combustion respectively, and increases overall fossil fuel emissions by about 1.3%. In addition, the latest IEA statistics for recent years show more updated values for fuel consumption than for years further in the past. Marland et al. (1999) compared for the first time the EDGAR and CDIAC datasets. Andres et al. (2012) followed this further with a more detailed analysis of the differences between the global $CO_2$ datasets available in 2012, including the 2012 version of CDIAC, IEA, EIA and EDGAR v4.2 (EC-JRC/PBL, 2011). One of the remaining differences is that the flaring in EDGAR v4.3.2 is twice as high as in CDIAC and EIA, which is explained by the different estimation method for the activity data (reported energy statistics in CDIAC and EIA versus satellite night lights of flaring from NOAA –NGDC (2015) and Elvidge et al. (2016) in EDGAR). Although the different EDGAR datasets deviate less than 0.5% for Annex I countries, this deviation becomes 3.4% for non-Annex I countries (see Figure S3 in the Supplementary).

Larger differences are seen for the non-combustion $CO_2$ emissions. Fig. 6a examines the most important ones comparing process emissions of the non-metallic sector (cement, lime, dolomite limestone, ceramics and glass production) of EDGAR v4.3.2, Le Quéré et al. (2016) and Xi et al. (2016). $CO_2$ from cement production in EDGAR v4.3.2 is 13% (19%) lower than in Xi et al. (2016) (based on CDIAC) because of the correction for the fraction of clinker in the cement produced. The EDGARv4.3.2 data provides cement production emission estimates very close to the estimates of Andrew (2018) as reported in Figures 3 and 4. A further large difference is found for developing countries, especially those with emerging economy. Fig. 6b zooms in with the total $CO_2$ emissions regionally on China and compares EDGAR v4.3.2 estimates per sector with those of Guan et al. (2012) and Liu et al. (2015), who brought a large underestimation respectively overestimation in the Chinese $CO_2$ inventory to the broad attention of scientists and media. Guan et al. (2012) indicated the 1.4 Gton $CO_2$ gap in the national total compared to the sum of the provincial statistics and proposed 9.1 Gton $CO_2$ in 2010. The EDGAR v.4.3.2 estimate of 8.8 ton $CO_2$ for 2010 differ only by -3%, which is composed of a difference of -19% for the fossil fuel combustion emissions and of +27% for the process emissions. In 2015 China revised its coal statistics with lower coal carbon content and the energy consumption was considerably decreased (for coal power plants with -12%). Liu et al. (2015) published coal carbon content for 4200 Chinese mines and analysed the impact on the total $CO_2$ from combustion in China. EDGAR v4.3.2 revised the activity data for 1990-2012 and obtained for 2010 an emission reduction for power generation of –8% and for the $CO_2$ total of -2% only.

Although Liu et al. (2015) reported 14% lower emissions compared to EDGAR, this is effectively only 6% (below the uncertainty range for China's $CO_2$ emissions) when correcting for the flaring, coke production, chemicals production and limestone which were not accounted for in their study. This illustrates the importance of clearly documented datasets for data comparisons and further understanding the sources of discrepancies. The higher estimate of Liu et al. (2015) can be understood by his 3 % lower average net calorific value[13] than the default of IPCC (2006) used by EDGAR.

### 3.2    The global $CH_4$ budget

Table 4 compares the EDGAR v4.3.2 global $CH_4$ estimates of 0.34 ($\pm$0.16) Pg $CH_4$ /yr with four other global datasets (the bottom-up inventories of US EPA, 2012 and GAINS[14] Eclipse v5 of Höglund-Isaksson et al., 2012); and the global budgets of Kirschke et al., 2013 and Saunois et al., 2016). Even though the global total $CH_4$ emissions for the bottom-up inventories vary less than 4%, global annual emissions from the agricultural and fossil fuel production sectors vary with $\pm$22% and $\pm$17%, respectively. The top-down inventory estimates are 16% $\sim$ 29% larger than the bottom-up ones.

Figure 7a illustrates the origin of the large variations in the estimated fugitive emissions of oil and gas production (including extraction, transmission and distribution). Large uncertainties in $CH_4$ from venting and flaring at oil & gas extraction facilities have been reported by e.g. Lyon et al. (2015) or Peischl et al. (2015). The $CH_4$ venting of oil and gas extraction facilities is, in particular during the times of the Soviet Union, now believed to be larger than previously thought (e.g. in EDGAR v4.2 or US EPA), after Höglund-Isaksson (2017) used ethane-methane ratios as an indicator. Additionally, gas distribution is a relative large source of uncertainty, in particular in countries with old gas distribution city networks using steel pipes now distributing dry rather than wet gas, with potentially more leakages. Based on IPCC (2006), EMEP/EEA (2009, 2013) and Marcogaz (2013), the emission factors for steel and grey cast iron pipelines vary in the range of 0.1 $\sim$ 7 ton/km/yr whereas this is a factor 2 lower for PVC and polyethylene pipelines. The difference in composition of the gas distribution networks is taken into account in EDGAR v4.3.2 with country-specific variations in emission factors. The high $CH_4$ emissions during the natural gas transmission in the Russian reporting to UNFCCC (2016) might also account for all or part of accidental $CH_4$ releases, which are not negligible according to Höglund-Isaksson (2017). These are not included in the EDGAR datasets.

China is currently the top emitter of $CH_4$ because it has become the largest coal producer and it is a major rice cultivator. While the fugitive $CH_4$ emissions from coal production in China are increasing, emissions from rice cultivation are decreasing, as shown in Fig. 7b. The emission factor $CH_4$/ha/yr for irrigated rice fields has been reduced from 1970 to 2000 by ~1/3 by changing farming practices, as reported by Li et al. (2002), resulting in

---

[13] This difference in average net calorific value results from a 8% difference in non-oxidation fraction and a 2% difference in energy-specific carbon content.

[14] Greenhouse Gas - Air Pollution Interactions and Synergies (GAINS) project of IIASA under http://gains.iiasa.ac.at/models/

0.47 kg CH$_4$/ha/yr for the last decade. A comparison with Peng et al. (2016) illustrates the large range of emission factors used: the emission factor in EDGARv4.3.2 for rice cultivation is twice as high than in Peng et al. (2016). Also for the coal mining the CH$_4$ emission factor for China in EDGARv4.3.2 is 9% higher than in Peng et al. (2016). EDGAR v4.3.2 revised emission factors for coal mining with local data from Peng et al. (2016), weighted by coal mine activity per province. These emission factors are at the lower end of IPCC (2006b) recommendations and yield EDGAR v4.3.2 estimates of 17.2Tg in 2008 and 21.2Tg in 2012, which are comparable to estimates of Peng et al. (2016) within ±2Tg.

Total CH$_4$ emissions in EDGAR v4.3.2 in 2005 are 2% (3%) lower than in the v4.2 (4.1) version, which has been used in global inverse modelling studies of Monteil et al. (2011), Bergamaschi et al. (2013, 2015, 2017), Ganesan et al. (2015), Kort et al. (2008), Miller et al. (2013). Except of the Chinese coal mining, no other major shortcomings to v4.2 were indicated in these global studies. More regional inverse modelling studies are nowadays able to "verify"[15] the CH$_4$ emissions better (such as Henne et al. (2016) for Switzerland) and first atmospheric model runs with EDGARv4.3.2 CH$_4$ emissions started recently. Total emissions have not changed significantly for either EU28 or the USA, but there are changes in the patterns of emissions: the -2.5% (-0.2%) change in the EU28 estimates of v4.3.2 compared to those of v4.2 (v4.1) is still within the range of the inverse model simulations of Bergamaschi et al. (2018), while the -4.7% (-3.4%) change in USA in EDGAR v4.3.2 compared to v4.2(v4.1) are not in line with the suggested +50 ~ 70% higher anthropogenic emissions based on the inverse modelling study of Miller et al. (2013). The latter might be explained at the emissions side by delayed reporting of statistics on fracking for shale gas and oil and the not well characterised and highly uncertain emission factors as indicated by US EPA (2015) and at the modelling side by large uncertainties of inverse models and the potential contribution of natural sources. For China the EDGAR v4.3.2 estimate for fugitive emissions from coal mining yields a 38% lower CH$_4$ emissions total in 2008, which is in line with Saunois et al. (2016), Brandt et al., (2014) and Kirschke et al. (2013), suggesting lower CH$_4$ emissions in particular in northern China where coal mining takes place.

**3.3    The global N$_2$O budget**

An overview of the global N$_2$O budget is not yet available as like for CO$_2$ and CH$_4$. Recent efforts from the modelling community to provide input for the global N$_2$O budget by Tian et al. (2018) report anthropogenic emission estimates for 2006 of 10.8 Tg N$_2$O/yr, confirming the 2005 global total by US EPA (2012) of 10.9 but a full overview of the global nitrous oxide budget is still forthcoming. The bottom-up estimate of EDGAR v4.3.2 of 7.2 (±3.7) Tg N$_2$O/yr for 2005 differs from this with 34%, which is still within the uncertainty range. The bottom-up estimate of GAINS by Winiwarter et al. (2018) differs in similar way by 29%. It is noted that the differences within each source category very remain large (see Table 5). A comparison at European level between the EDGAR v4.3.2 and the N-budget of Leip et al. (2011) shows relative moderate discrepancies also at sector-specific level with for the total and the agricultural sectors 26% respectively 37% smaller estimates by

---

[15] The term "verify" is selected in consultation with the EC policymakers for Climate and refers to the detection of biases in emission inventories.

EDGAR compared to Leip et al. (2011) but for the non-agricultural sectors 19% larger estimates. Höglund-Isaksson et al. (2010) provided GAINS estimates for EU27 that are respectively 28%, 42% and 20% larger than the total, agriculture and non-agricultural sectors estimates of Leip et al. (2011).

Although in EDGAR v4.3.2 the agricultural sector is contributing most to the anthropogenic direct and indirect N$_2$O emissions, the production of chemicals, such as nitric acid, glyoxal, caprolactam and adipic acid production, and its use as anaesthesia or for aerosol spray cans also plays an important role. In 1970 the chemicals sector contributed 20% to the total, but this has been significantly reduced to less than 8% because of technological developments. Figure 8 shows the impact of technological developments from old plants to higher pressure plants or plants with non-selective catalytic reduction, reducing the N$_2$O emissions by factors of 2 and 10, respectively. The N$_2$O emissions of nitric and adipic acid plant facilities which EDGAR v4.3.2 estimated are in line with the estimates of US EPA (2012) and by GAINS for the year 2005. However a discrepancy evolves when looking at the 2010 values, because of the relative large reduction between 2007 and 2010 in EDGAR and the relative constant trend in GAINS. While EDGAR assumes abatement technologies for nitric and adipic acid plants in China following the reporting under the Clean Development Mechanism, Schneider et al. (2010) assumes that abatement was not used at least for the new adipic acid plants. The latter assumption was followed by Winiwarter et al. (2018) and explains the differences in the global nitric and adipic acid N$_2$O emission estimates between GAINS and EDGAR.

## 4. Discussion of the trends

### 4.1 Global Greenhouse Gases 1970-2012

A country-based statistical analysis including 4 decades of GHG emissions (EDGAR v4.2) and GDP (Purchasing Power Parity data of the Penn World Tables 7 of Feenstra et al., 2013) was carried out to investigate the possible causality between emissions and income. The results, summarised in Paruolo et al. (2015) showed that no presence of causality could be statistically proven. This reflects a complex link between the very heterogeneous economic activities (ranging from manufacturing to services) and emissions, and justifies the meticulous bottom-up inventory compilation using statistics instead of modelling.

Figure 3 shows the global trend of GHG emissions in CO$_2$-equivalent (100 year time horizon), using the GWP-100 values of AR4 (IPCC, 2007)[16]. The GHG total is composed of all sources (excluding LULUCF) of CH$_4$ and N$_2$O but only CO$_2$ from long cycle C fossil sources and excluding the short cycle[17] C for the CO$_2$ accounting,

---

[16] In the latest UNFCCC revision of the reporting guidelines adopted by COP (2014), it was decided to use for the reporting from 2015 onwards the global warming potential coefficients (GWP-100) from AR4 (IPCC, 2007) with 25 for CH$_4$ and 298 for N$_2$O.

[17] The IPCC (2006) methodology for CO2 accounts the emissions from short cycle C (released by combusting biofuels, agricultural waste burning or field burning) under the Agriculture, Forestry and Land Use (AFOLU)

conform IPCC (2006). The estimated global total GHG in 2010 of 44.7 Pg $CO_{2eq}$ was shown to be 0.7% lower than the estimates for the 2010 global total (without LULUCF) in the UNEP (2012, 2015) Emission Gap reports. The share of each gas to the total GHG is relatively stable and yields for $CO_2$ 76.8% (+2.1pp, -1.2 pp), for $CH_4$ 18.1% (-2.5 pp, +1.9pp) and for $N_2O$ 5.1% (+0.4pp, -0.6pp), where in between brackets the percent point impact

of the evolution of the GWP-100 value from the SAR (IPCC, 1996b)[18] to the AR5 (IPCC, 2014)[19] is given.

In the global GHG emissions time series, the trend was shown to be dominated by $CO_2$ as it has the largest share and the largest increase. In the 1970s $N_2O$ increased at the same rate as $CO_2$ (2.6%/yr), while $CH_4$ was half as fast. In the 1980s and 1990s, $N_2O$ and $CH_4$ increases were very small, while $CO_2$ continued albeit at a slower rate (1.6%/yr). In the last decade 2002-2012 $CO_2$ and $CH_4$ growth rates increased with respectively 3.2%/yr and

2.0%/yr. While over the four decades (1970-2012) the global total GHG increased in line with global population (91% versus 88%), the inter-annual and regional emission variations do not always reflect the rates in population increase but are instead better explained by the global fuel markets and economy, with the 1973 and 1979 oil crises, the dissolution of the Soviet Union (1989-1991), the growth of the Chinese economy, after they joined the World Trade Organisation in 2002 and the 2008 global financial crisis.

## 4.2    Greenhouse gas trend analysis for regions and top emitting countries

Figure 4 shows the GHG trends for the major regions: 24OECD90 (split into USA, EU15 and the rest), 16EIT90 (with Russia and EU13 and the rest) and non-Annex I (for which China, India and Brasil are shown separately). The gas-specific GHG trend is also available per country in Janssens-Maenhout et al. (2017) and downloadable from  edgar.jrc.ec.europa.eu/overview.php?v=CO2andGHG1970-2016.  To  understand  the  trends  of  the  total

GHG (in CO2eq) the decomposition with the trends of $CO_2$, $CH_4$ and $N_2O$ trend for the same regions is given in the figures S4.a, S4.b and S4.c respectively and with a discussion per country group in the Supplementary.

Focussing on the top four emitting countries and regions, Figure 5 compares the reported UNFCCC (2004, 2012, 2014, 2016, 2017) emissions of China, USA, EU28, Russia and the emission estimates of EDGAR v4.3.2. There is a very good agreement between the UNFCCC reported values and the EDGAR v4.3.2 estimates for the EU28,

whereas for USA and Russia the EDGAR v4.3.2 estimates are lower than those reported by UNFCCC (2016). For the USA this is explained by lower $N_2O$ emissions in EDGAR v4.3.2, although $N_2O$ emissions reported by USA to UNFCCC (2014, 2016) are within the large uncertainty range for the EDGAR v4.3.2 estimates. For Russia $CH_4$ emissions reported to UNFCCC (2016) are 37% higher than those estimated by EDGAR v4.3.2, but also this is within the uncertainty range. The largest difference is found in the estimation of gas pipeline

transmission emissions, which are 4 times higher in the UNFCCC inventory of Russia than in EDGAR v4.3.2. The relatively low emission factor for gas pipelines, used by EDGAR, is in line with the recommendations of

---

sector (see  IPCC (2006), Vol.2, sector 2.3.3.4 related to biomass combustion and methodologies for harvested wood products).

[18] In SAR (IPCC, 1996b): GWP-100 of $CH_4$ = 21 and GWP-100 of $N_2O$ = 310

[19] In AR5 (IPCC, 2014): GWP-100 of $CH_4$ = 28 and GWP-100 of $N_2O$ = 265

Lelieveld et al. (2005). For China, a very good agreement between the EDGAR v4.3.2 estimate and the UNFCCC (2004, 2012, 2017) reported values is obtained, taking into account the importance of the coal statistics revision. In order to evaluate the latter effect, two time series of emission are calculated by EDGAR, with and without coal statistics revision. The revision includes a decrease of the 2010-2012 values, and yields an increase for the 1990-2009 values of about +3% for 2005 and 1994. It is evident that the previous estimates of the UNFCCC inventory in 2005 and 1994 would need to be revised in order to evaluate the emissions change from 2005 to 2012. Even if relative uncertainty in EDGAR estimates for China could be reduced, it is evident that the size of the Chinese inventory has large impact on the global absolute uncertainty.

## 5.    Discussion of the grid-maps

In this section, the gridded EDGAR datasets at 0.1°x0.1° are further screened to identify hot spots and to check for anomalies. An overview of the region-specific totals and their sector-specific composition for the year 2012 is given in Figs. 9, 12 and 15 for the different substances. The sector-specific country totals are provided in the overview Table 6a per region and 6b per sector for 2012.

### 5.1    $CO_2$ emissions and urban hot spots

The 2012 grid-map of $CO_2$ emissions from both long-cycle and short-cycle carbon in Fig. 9 with the relative sectorial breakdown for selected world regions (Europe, North America, Latin America, Africa, Middle East, Oceania, Russia and China) clearly shows the fossil fuel combustion activities, representing 90.6% of the total $CO_2$ emissions. In this section we include for completeness biofuel emissions, which were omitted from the comparisons with UNFCCC reporting, because UNFCCC assumes carbon neutrality for all agricultural and biofuel $CO_2$ emissions in a country for any individual year. In the 24OECD90 countries 75.2% of $CO_2$ emissions are produced by the power, road transport and residential sectors, while these sectors represent only 60.9% in non-Annex I countries. The share of the industrial combustion and production sectors (mining/manufacturing) of non-Annex I countries reaches 36.8%. The $CO_2$ shares of the fuel combustion in the power generation, road transport, buildings and manufacturing sectors vary for the different regions from 16~50%, 5~27%, 6~39% and 9~22% of total emissions (see Table 6a and 6b) respectively. Interestingly, agricultural waste burning[20] represents 10% of $CO_2$ emissions in Latin America (mainly due to sugarcane crop residues burning) and 22% of $CO_2$ emissions in Africa derives from the transformation industry (charcoal production using as input primary solid biomass). Industrial emissions are distributed at the point-source locations of the power/heat plants or industrial facilities (e.g. cement factories) using the capacity of the plants or facilities as a weighting factor.

In the grid-maps hotspots are particularly visible over cities, of which the top 4 are emitting 2.75% of the global total[21] and coincide with the cities of Shanghai, Huangshi, Shenyuang and Moscow. In fact, 5% of the 0.1°x0.1°

---

[20] Note that the agricultural waste burning is not including the Savannah burning.

[21] At a rate of more than 125 Mton/(0.5° x 0.5°)

grid cells are emitting more than 5Mton/(0.1°x0.1°)/yr and account for 34.08% of the global total. It is therefore interesting to look at the contribution of the various sectors in megacities, as shown in Fig. 10. Emissions from the road transport sector (Fig 10a) for the 20 selected cities seem to be more important in suburban areas than in the centre of the megacity. For power plants more heterogeneity was found (Fig. 10b) with larger power plants typically located on the periphery of the city in the 24OECD90 countries, while for major cities of the 16EIT90 and non-Annex I countries, several larger power plants are located within the central city areas. The remaining share of $CO_2$ emissions was shown to be mainly from the buildings sectors and the industrial manufacturing emissions.

The evolution over time from 1970 to 2012 shows a different pattern for the residential sector than for the road transport sector. Fig. 11a shows that while the residential sector decreased over these 4 decades in America and Europe, it increased in Asia and Africa. The difference in $CO_2$ emissions from the road transport sector meanwhile presents in Fig. 11b a more homogeneous picture with increases from 1970 to 2012 in almost all regions. Please note that Fig. 11 includes both long-cycle and short-cycle carbon fuel use, but Fig. S5a-d in the Supplementary presents these separately and shows e.g. the use of the vegetal waste and dung for residential heating in India and the biofuel use for car transport in Brasil.

### 5.2    $CH_4$ emission maps

Because $CH_4$ is mainly released from fermentation processes (enteric, manure, landfills or rice) or diffusion processes (coal mine leakage or gas distribution losses), the 2012 $CH_4$ emission grid-map with sector contributions for major world regions (Fig. 12) does not mirror the same human activities as the $CO_2$ map. The $CH_4$ shares for enteric fermentation, fossil fuel production & transmission and solid & water waste treatment range from 9~59%, 8~68% and 11~37% of the global total respectively, depending on the region. For 24OECD90 countries enteric fermentation (with 31.1% share), fossil fuel production (28.1%) and landfills (21.4%) are the three dominant sectors, whereas in the 16EIT90 countries, $CH_4$ emissions are dominated by fossil fuel production (49.4% share). The non-Annex I countries show a similar high share of enteric fermentation and fossil fuel production as the 24OECD90 countries, but rice cultivation and domestic wastewater together give much higher emissions than solid waste disposal. Rice cultivation was shown to contribute significantly to the total $CH_4$ inventory of China (21.5% or 14.2 Tg in 2012), which is almost 11 times the $CH_4$ emissions of rice cultivation in India (3.8 Tg), despite the larger area for rice fields in India than in China (425 compared to 303 thousand km$^2$). This is explained by the fact that India typically has one harvest per year from 1/3 rain-fed fields and 2/3 irrigated fields, whereas China has multiple harvests per year from irrigated rice fields. Rain-fed rice fields in India are modelled with a five times lower emission factor than the irrigated fields in China. Figure 13a and 13b show the opposing trends with mainly positive 2012-1970 increments in enteric fermentation (mainly cattle) (a) and mainly negative increments in $CH_4$ emissions from rice cultivation (b). The $CH_4$ trend from rice cultivation in Asia was shown to be remarkably stable with the exception of Thailand where increased activity is noticed. The remaining non-Annex I countries of Africa and Latin-America show similar high contributions from enteric fermentation (25.8 Tg versus 20.9 Tg respectively in 2012). However, the total $CH_4$ emissions from the African continent are higher than those of Latin-America because of the 3.5 times larger $CH_4$ emissions from fossil fuel production (gas and oil production). Interestingly, both continents show

significant $CH_4$ emissions from charcoal production, which compares to 16% (Africa) and 15% (Latin-America) of their gas and oil production emissions of $CH_4$.

Hot spots of $CH_4$ are estimated for fossil fuel production, typically at gas & oil production facilities or at coal mines, as shown in Fig. 14. In North America a shift over the period 2005-2012 from coal mining in the North-East (-21%) to gas & oil production in particular in North-Dakota, Montana and Texas (+65%) took place. The USA is nowadays the largest producer of both shale gas and tight oil, which are making up almost half of total US gas and oil production (EIA, 2015). In Europe a much larger decrease of -87% in coal production happened earlier while gas production increased by 30%. Consequently the EU28 needed to rely on oil and gas imports and expanded its transmission and gas distribution network with corresponding increase in $CH_4$ leakages. Aside of the USA, also the Middle East was shown to be a global world player on the oil and gas market, shifting from oil production (with a decrease of 71% over the period 1976-1985) to gas production (with a 9.3-fold increase from 1985 to 2012), mainly driven by Iran, Saudi Arabia and Qatar. The African countries with the highest $CH_4$ emissions from fossil fuel production were in decreasing order of importance Algeria and Nigeria (for oil and gas production) and South Africa (for coal mining). Similarly to Nigeria, which showed an approximate doubling of $CH_4$ emissions from oil (and gas) production over the last 4 decades, Mexico and Venezuela also showed similar levels of $CH_4$ emissions from oil and gas production (increasing with a factor 1.6 over the 4 decades). For gas production, Russia has shown the largest $CH_4$ venting and leakage, overtaking the USA in 1985.

Coal mining has become important for China, which is since 1982 the largest bituminous coal producer in the world, overtaking the USA. Moreover China was shown to be also the largest coal importer since 2011 (overtaking Japan), as domestic coal produced in mainly the western and northern inland provinces of China faced a bottleneck in transportation, lacking southbound rail lines (Tu, 2012) towards the southern coast that has the highest coal demand. Not only did EDGAR v4.3.2 revise the country-specific coal mining emission factors, but also the spatial distribution was considerably updated with hot spots at the location of the mining activity. For coal mine activities in China (split in brown and hard coal), the coal mine database of Liu et al. (2015) provided over 4200 coal mine locations, which is 10 times more than that available for EDGAR v4.2. For Europe, the closure of mines since the 1990s has been taken into account using European Pollutant Release Transfer Register (EPRTR, 2012).

### 5.3    $N_2O$ emissions including indirect sources

Unlike the $CO_2$ and $CH_4$ grid-maps, the gridded $N_2O$ emissions for the year 2012 in Fig. 15 with the share of different sectors for world regions showed a quite uniform global coverage distribution, due to the predominance of soil emissions and indirect emissions (distributed with the N-deposition map of Dentener et al. (2006)), also from the seas surface. Over land, most $N_2O$ is emitted from the agricultural soils (the use of animal manure as fertiliser, the application of N containing fertilisers and cattle in pasture), representing from 35% to 86% of total $N_2O$ emissions depending on the region. Fertilising farmland with pasture or animal waste as fertiliser or crop residues has not increased so much as the use of nitrogen fertilisers. Figure 16 shows the increased use (by the difference 2012-1970) of nitrogen fertiliser, in particular in Asia.

### 6. Conclusion and outlook

In line with the ESSD guidelines of Carlson & Oda (2018), we aim with this publication not only for free open access to all calculated data (with their uncertainty) but also for a complete documentation of the EDGARv4 products that has been compiled in transparent way, to the extent possible.

#### 6.1 Strengths and applications of EDGAR v4.3.2

The scientific global emission inventory database EDGAR v4.3.2 provides a comprehensive dataset of anthropogenic emissions of $CO_2$, $CH_4$ and $N_2O$ in time series 1970-2012 (with monthly resolution) and spatially disaggregated grid maps with 0.1°x0.1° resolution. An advantage of EDGAR v4.3.2 is that the bottom-up emissions calculation methodology is applied to all countries and the results are available with regular updates based on a robust statistical data infrastructure and provide direct information to policy makers in the standard structure as used for the Annex I countries. EDGAR v4.3.2 may provide useful information to countries with less strong statistical data infrastructure for their future inventory requirement. In particular the time series of EDGAR v4.3.2 can complete the emission trends for non-Annex I countries, as illustrated for the case of China, where the coal statistics revision impacts also the 2005 and the 1994 inventory with +3%.

For the atmospheric modelling community EDGAR v4.3.2 enables models to use historical emission grid-maps for a top-down assessment of the total budget, making use of in-situ and remote sensing atmospheric observation records. The results of inverse atmospheric models provide an evaluation of the nationally collected emission data with regard to their uncertainty and as such support the scientific review and updates of emission inventory methodologies. For recent years (e.g. 2010) total anthropogenic budgets of 33.6 (±5.9) Pg $CO_2$ /yr , 0.34 (±0.16) Pg $CH_4$ /yr and 7.2 (±3.7) Tg $N_2O$ /yr are obtained. The current evaluation capacity of inverse models using atmospheric measurements remains limited where the models struggle with an accurate separation of the natural emissions component from the total. Although modelling uncertainties and the uncertainties of natural emissions remain large, the atmospheric models provide observationally constrained top-down input and it is expected that inverse models increasingly contribute to the independent verification of the total fluxes. Moreover, the impact of updates of recommended tiered emission factors (such as from IPCC (1996; 2006), the upcoming refinement in 2019, or selected region-specific data) on the resulting emissions can be assessed at global scale. EDGAR v4.2 evaluated the impact of the update of the $N_2O$ emission factor for direct soil emissions from the use of fertilisers (synthetic or manure or crop residues) by IPCC (2006b) with a 20% lower value than what the IPCC (2000) Good Practice Guidance provided as default. The update of the EDGAR v4.2 version to v4.3.2 demonstrated e.g. the necessity to take up region-specific emission factors for fugitive coal mining emissions in China, which are considerably lower than the IPCC lower tier 1 default values (e.g. Peng et al., 2016; Saunois et al., 2016).

With the 42 year long time series of EDGAR v4.3.2 we provide an important input to the analysis of global GHG trends. We find an accelerated increase of GHG emissions since the beginning of the 21[st] century compared to the three decades before, mainly driven by the increase in $CO_2$ emissions from countries with emerging economies. For the EU-28 the trend is determined by a rather stable share of $CO_2$ and a smooth but continuously decreasing $CH_4$ contribution, resulting in an overall reduction of total GHG emissions. Even though the uncertainty of global total emissions has increased mainly because of the increasing share of GHG emissions

from emerging economy countries, on European scale the uncertainty has decreased because of the progress in inventory compilation and the decrease in some sectors with more uncertain $CH_4$ emissions.

Overall the EDGAR v4.3.2 database aims at providing useful information for both the scientific and policy communities involved in understanding GHG emissions and budget, e.g. for the compilation of national inventories, the UNFCCC periodic global stocktake, analysis of co-benefits between air pollution and GHG emission mitigation strategies, interpretation of in-situ or space-borne Earth observation data, understanding and reducing of uncertainties.

## 6.2    Use, evaluation and limitation of the EDGARv4.3.2 dataset

The EDGARv4.3.2 provides a global picture of GHG emissions using a tier1-2 approach following IPCC (2006) guidelines and allowing comparison of country- and sector-specific sources. This global completeness comes at the expense of lacking or less accurate information at (i) higher resolution or subnational focus and (ii) detailed modeling of subsector emissions beyond tier 1-2.

Therefore, the potential users of the EDGARv4.3.2 dataset are recommended to carefully consider these limitations when:

- Applying it for region-specific, subnational or urban case studies, for which more detailed inventories should be used or constructed, using bottom-up and local information. EDGARv4.3.2 only uses national data and any subnational level is the result of a spatial distribution (top-down) making use of proxy data. In general large differences can be expected between the top-down spatially downscaled national emissions with proxy data and the bottom-up inventory with local data, sometimes not strictly reporting emissions that occur solely inside the small territory, as Gately et al. (2017) demonstrate. The EDGARv4.3.2 can be used to gap-fill between other regional inventories (e.g. in HTAPv2.2 of Janssens-Maenhout et al., 2015), or to bridge the gap between point sources and the national inventory (e.g. Theloke, J. et al., 2011). These gapfillings come at the expense of losing consistency within the reported emissions as inventory.

- Applying the dataset outside the period 1970-2012 can only be recommended when taking into account the fast track update from 2013 onwards based on recent statistics, for which we refer to the annual publication on the hyperlink http://edgar.jrc.ec.europa.eu/index.php#. For the years before 1970 we refer to the HYDE dataset. We would refrain from any linear extrapolation based on short term trends of the emission time series or of emission drivers, for which the causality proof for these short time-series is missing.

- Comparing with other gridded datasets at grid cell level, especially when using EDGARv4.3.2 disaggregated subsector emissions data. Several assumptions on the technological evolution and the spatial distribution flow into the EDGAR4.3.2 subsector emissions. The difference between two grid-maps can not be unambiguously attributed to missing activity data, the selected region-specific emission factor, or the assumed technology-share or the spatial distribution proxy data. In particular the latter factor is important, in particular for very localized sources or point sources (such as industrial

complexes or urban areas). Moreover the strength of point sources is very sensitive to the choice of the characteristic parameter (such as designed capacity or averaged annual emission estimate or given annual throughput) in the proxy dataset and can vary strongly over time. As Hogue et al. (2016) indicates: the largest uncertainty contribution in gridded emission datasets comes from the proxy data used for spatial disaggregation of national emissions. The subtraction of the sum of all the point sources from the sector-specific country total leaves a remaining emission that is composed of smaller sources and that is typically distributed with e.g. a population density proxy, as information is lacking. The uncertainties for the point sources and for the remaining smaller sources are highly different and larger than the uncertainty of the sector-specific country total. Information on the representativeness of the selected characteristic parameter for point sources is most critical and needs to be evaluated with measurements (such as in-situ atmospheric measurements of co-emitted pollutants) but would require an in-depth analysis beyond the scope of this paper.

The EDGARv4.3.2 is the result of continuous improvement of previous datasets, which have been used by modelers in inverse modeling studies to verify the level and distribution of the emissions. Feedback has been taken into account (e.g. Saunois et al., 2017 for CH4). The evaluation of the dataset with a more advanced uncertainty assessment did not take place yet.

## 6.3    Future perspective

EDGAR v4.3.2 demonstrates that inventories can be developed for all countries within the limitations of the quality of the available statistical data in order to contribute to the comprehensive picture needed for the UNFCCC periodic global stocktakes. In 2023 a first global stocktake is foreseen to track the progress of the collective efforts to reduce the emissions as promised under the NDCs. Comprehensive information on emissions for all world countries can help to assess and build trust in the effectiveness of the NDCs. In particular, the country estimates of EDGAR v4.3.2 can help countries with less developed statistical infrastructures to compile their inventories and complete time series.

EDGAR v4.3.2 yields not only grid maps for all greenhouse gases, but also for air pollutants, representing multi-pollutant sources as single point source with realistic ratios of the different pollutant emission rates. To analyse the co-benefits and trade-offs of integrated approaches towards climate and energy as well as air quality policies, it is of key importance to use the transparency framework of "measuring-reporting-verifying" for a world-wide evaluation of the emissions. A bridge between the inventory compilers and satellite community can yield more dynamic emission databases. So far the interpretation of satellite data is more successful with air pollutants, $NO_x$, $SO_2$, CO but also CH4. For interpreting CO2, Berezin et al. (2013) demonstrated a new methodology using ratios of $NO_2:CO_2$ to reveal the fossil fuel component of $CO_2$.

Emissions provided by the EDGAR database cannot be always considered as the best country or region-specific estimate. The use of a common denominator as technology-based methodology across the world implies for some regions the loss of more detailed knowledge and differences from the local inventories. However, the comprehensiveness of the EDGAR v4.3.2 grid-maps allows to generate per grid-cell the emission ratios of

different GHG and air pollutant gases or the sector-specific shares, as additional information for interpreting satellite retrievals measuring column-averaged dry-air mole fractions of total $CO_2$ or $CH_4$.

### 7.    Access to the data

Annual grid-maps for all GHGs and sectors covering the years 1970-2012 are available as txt (expressed in the unit: ton substance per grid cell) and NetCDF (expressed in the unit: kg substance/m2/s) with 0.1°x0.1° spatial resolution, available under DOI:10.5281/zenodo.2658138. (This is the GHG part with $CO_2$,- long-cycle carbon, CO2 – short-cycle carbon, $CH_4$ and $N_2O$ gridmaps and timeseries of the EDGAR dataset with PID: http://data.jrc.ec.europa.eu/dataset/jrc-edgar-edgar_v432_ghg_gridmaps.) In addition, monthly GHG global grid-maps are produced for 2010 and are available per sector and substance. The main features of the grid-maps are described (section 5) while focusing on the year 2012, although analogous considerations also pertain to previous years.

### 8.    Acknowledgement

The EDGAR database compilation was initiated by J. Olivier (PBL), but in 2008 handed over to JRC, where a new version v4 was developed with a relative large turnover of personnel. The EDGARv4 development profited from the substantial contribution of the non-JRC/non-PBL authors when they were affiliated to JRC. The authors are grateful to the IEA, K. Treanton and R. Quadrelli, for the collaboration and data exchange of the energy related statistics and to the JRC colleague, J. Wilson for the thorough review and English proofreading. Last but not least our thanks go to Prof. D. Carlson, who significantly improved the readability of the manuscript by his thorough revision and suggestions for abstract, structure, shortening and working with tables. We believe that his efforts were also helping to bridge different communities: on carbon modeling and on GHG inventory compiling.

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

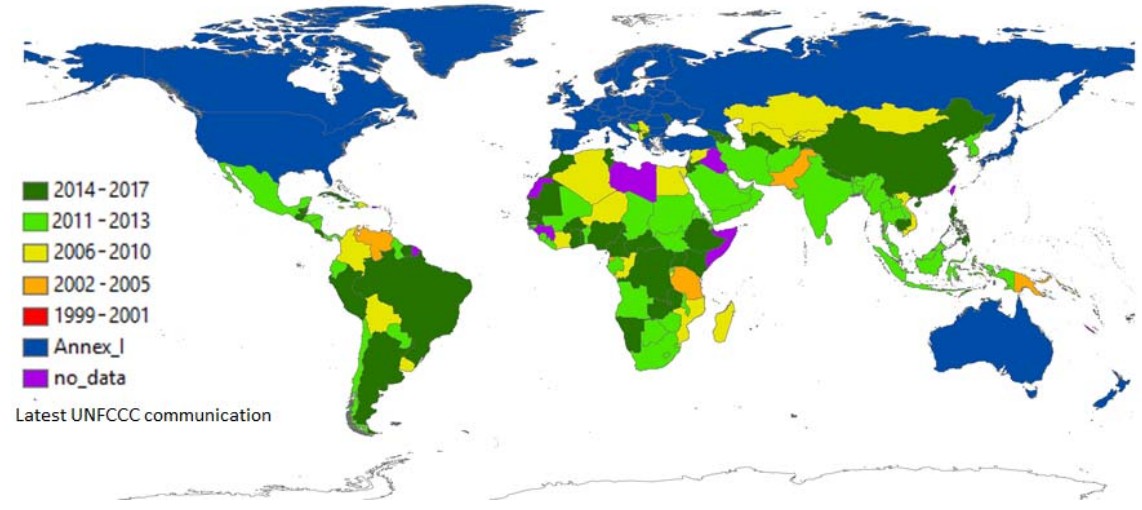

**Figure 1a - Inventory submission as received at UNFCCC (by January 2017) for all countries: expressed with the year of emission reporting in which the latest national communication to UNFCCC took place.**

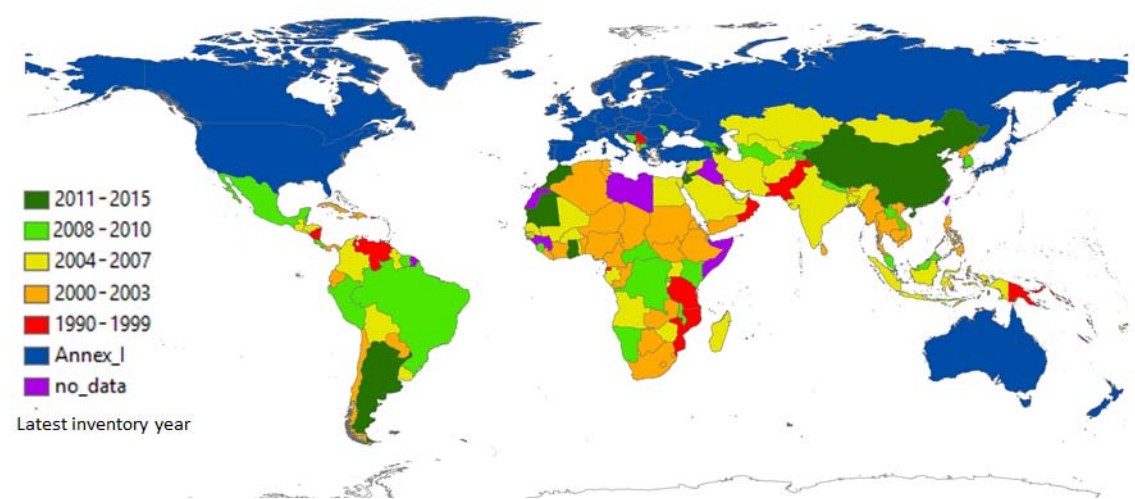

**Figure 1b - Inventory submission as received at UNFCCC (by January 2017) for all countries expressed with the latest year of emission that is covered in the inventory submitted to UNFCCC.**

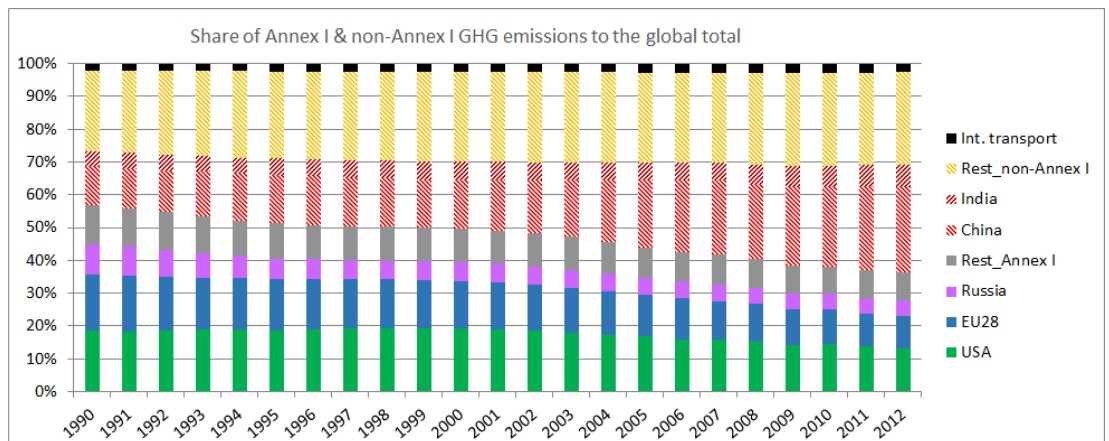

**Figure 2: Relative contribution of the Annex I and non-Annex I countries to the global total GHG emissions. The red, brown and orange dashed parts of the stack correspond to the non-Annex I share that increases from about 1/3 in 1990 to almost 2/3 in 2012.**

**Table 1a: Main category with all Source/Sink Categories conform to the IPCC Guidelines (1996). Note that neither large scale biomass burning nor land-use, land-use change and forestry emissions are included, although we do include biofuel combustion and agricultural activities (such as livestock and milk production, crop and rice production, agricultural waste burning, field burning, histosols and liming).**

| Main category of emission sectors | EDGAR_code | Emission sectors of data delivery | IPCC_1996 | IPCC_2006 |
|---|---|---|---|---|
| **Energy** comprises the production, handling, transmission and combustion of fossil fuels and biofuels and is calculated with energy statistics. For CO2 the short cycle C is split off from the long cycle C, because the short cycle CO2 emitted from the combustion of biofuel is assumed to neutralise the CO2 uptake during the same year the biofuel was grown. Any disequilibrium of this balance needs to be taken up under the Land-Use, Land-use change and forestry sector. As such the long cycle CO2 energy refers to fossil fuel combustion only, the short cycle CO2 energy refers to the biofuel combustion. All other substances include fossil and biofuel combustion. | ENE | Power industry | 1A1a | 1.A.1.a |
| | IND | Combustion for manufacturing | 1A2 | 1.A.2 |
| | RCO | Energy for buildings | 1A4 | 1.A.4+ 1.A.5.a+ 1.A.5.b.i+ 1.A.5.b.ii |
| | REF_TRF | Oil refineries and Transformation industry | 1A1b+ 1A1c+ 1A5b1+ 1B1b+ 1B2a5+ 1B2a6+ 1B2b5+ 2C1b | 1.A.1.b+ 1.B.2.a.iii.4+ 1.A.1.c+ 1.A.5.b.iii+ 1.B.1.c+ 1.B.2.a.iii.6+ 1.B.2.b.iii.3 |
| | TNR_Aviation_CDS | Aviation climbing&descent | 1A3a_CDS | 1.A.3.a_CDS |
| | TNR_Aviation_CRS | Aviation cruise | 1A3a_CRS | 1.A.3.a_CRS |
| | TNR_Aviation_LTO | Aviation landing&takeoff | 1A3a_LTO | 1.A.3.a_LTO |
| | TNR_Aviation_SPS | Aviation supersonic | 1A3a_SPS | 1.A.3.a_SPS |
| | TNR_Other | Railways, pipelines, off-road transport | 1A3c+ 1A3e | 1.A.3.c+ 1.A.3.e |
| | TNR_Ship | Shipping | 1A3d+ 1C2 | 1.A.3.d |
| | TRO | Road transportation | 1A3b | 1.A.3.b |
| **Fugitive** refers mainly to gas flaring and venting during oil and gas production, coalbed methane during underground or surface mining and CH4 distribution losses and evaporation during transmission and mainly distribution. This is based on fuel production statistics, supplemented nightlight observations. | PRO | Fuel exploitation | 1B1a+ 1B2a1+ 1B2a2+ 1B2a3+ 1B2a4+ 1B2c | 1.B.1.a+ 1.B.2.a.ii+ 1.B.2.a.iii.2+ 1.B.2.a.iii.3+ 1.B.2.b.ii+ 1.B.2.b.iii.2+ 1.B.2.b.iii.4+ 1.B.2.b.iii.5+ 1.C |

| | | | | |
|---|---|---|---|---|
| **Industrial Processes** refer to non-combustion emissions from either manufacturing of cement, lime, soda ash, carbides, ammonia, methanol, ethylene, methanol, adipic acid, nitric acid, caprolactam, glyoxal and other chemicals, or from production of metals and from the use of soda ash, limestone and dolomite, from production of ferrous and non-ferrous metals and from non-energy use of lubricants and waxes. The emission estimates use the volume of industrial product produced (and traded) from the industry statistics. | CHE | Chemical processes | 2B | 2.B.1+ 2.B.2+ 2.B.3+ 2.B.4+ 2.B.5+ 2.B.6+ 2.B.8 |
| | FOO_PAP | Food and Paper | 2D | 2.H |
| | IRO | Iron and steel production | 2C1a+ 2C1c+ 2C1d+ 2C1e+ 2C1f+ 2C2 | 2.C.1+ 2.C.2 |
| | NEU | Non energy use of fuels | 2G | 2.D.1+ 2.D.2+ 2.D.4 |
| | NFE | Non-ferrous metals production | 2C3+ 2C4+ 2C5 | 2.C.3+ 2.C.4+ 2.C.5+ 2.C.6+ 2.C.7 |
| | NMM | Non-metallic minerals production | 2A | 2.A |
| **Solvents and Products use** includes CO2 from solvents in paint, degreasing and dry cleaning, chemical products and other product use, as well as use of N2O as anaesthesia and in aerosol spray cans. Estimates are based on a combination of population and solvents statistics. | PRU_SOL | Solvents and products use | 3 | 2.B.9+ 2E+ 2F+ 2G+ 2D3 |
| **Agriculture** comprises the application of urea and agricultural lime, enteric fermentation, rice cultivation, enteric fermentation, manure management, fertiliser use (synthetic and manure), agricultural waste burning (in field) and is based on agricultural statistics. Large scale biomass burning from Savannah is not included. | AGS | Agricultural soils | 4C+ 4D | 3.C.2+ 3.C.3+ 3.C.4+ 3.C.7 |
| | AWB | Agricultural waste burning | 4F | 3.C.1.b |
| | ENF | Enteric fermentation | 4A | 3.A.1 |
| | MNM | Manure management | 4B | 3.A.2 |
| **Waste** comprises landfills and wastewater management, and waste incineration that is not producing energy (neither generation of electricity nor heat recovery, because these are accounted in the energy sector(non-energy). Estimates are based on a combination of population and solid and liquid waste product statistics. | SWD_INC | Solid waste incineration | 6C | 4.C |
| | SWD_LDF | Solid waste landfills | 6A+ 6D | 4.A+ 4.B |
| | WWT | Waste water handling | 6B | 4.D |
| **Other** refers to direct emissions from fossil fuel fires (coal fires & the Kuwait oil fires), N2O usage and indirect emissions from atmospheric deposition of NOx and NH3 from non-agricultural sources, for which other historical statistics are consulted. | FFF | Fossil Fuel Fires | 7A | 5.B |
| | IDE | Indirect Emissions | 7C | 5.A |
| | N2O | Indirect N2O from agriculture | 4D3 | 3.C.5+ 3.C.6 |

**Table 1b: Data sources for activity statistics and emission factors for the main categories of emission sources defined in Table 1a. (cfr. References in the Manuscript or the Supplementary).**

| | | Activity Data (AD): data source | Activity Data (AD): data references | Emission Factors (EF): data source | Emission Factors (EF): data references |
|---|---|---|---|---|---|
| energy balance statistics | fossil fuel | IEA energy balance statistics (version 2014, last year 2012) for 138 OECD and Non-OECD countries expressed in TJ for the 64 fuel types and 94 activities.[1] | IEA (2014) | IPCC (2006) Guidelines (GL) | IPCC (2006b) |
| | Biofuel | IEA final consumption (version 2014) of biogasoline (bioethanol), biodiesel and other liquid biofuel categories for OECD countries.[2] | IEA (2014); EIA (2013); US DA (2014) | IPCC (2006) GL | IPCC (2006b) |
| fossil fuel production statistics | coal | World Coal Association data of 2016 for hard coal and brown coal production data, separated into surface and underground mining.[3] | World Coal Association (2016) | IPCC (2006) GL, supplemented with EMEP/EEA (2013) Guidebook EF (specified in function of average depths of coal production).[4] | IPCC (2006b); EMEP/EEA (2013) |
| | gas and oil | *For exploration*: IEA (version 2014) gas & oil production data. *For transmission & distribution:* the leakage rate is calculated in function of the length of the pipelines and its construction material (grey cast iron, steel, polyethylene or polyvinylchloride), with data from Eurogas (2010) report and Marcogaz (2013) technical sheet, UNFCCC National Inventory Reports (2014) and CIA (2008, 2016). *For venting and flaring:* the total amount of gas flared and vented is calculated based on the difference in fuel produced and fuel sold from IEA fuel statistics (version 2014) supplemented with trends from CDIAC (timeseries till 2014), EIA (2014) and counterchecked against UNFCCC (2014) National Inventory Reports for most countries until 1994. The share that is flared is from 1994 onwards derived with NOAA satellite observation of the intensity of flaring lights as DMSP data of US Air Force Weather Agency and updated with VIIRS data of NOAA-NGDC (2015) | *Exploration*: IEA (2014). *Transmission/distribution:* Eurogas(2010); Marcogaz (2013); UNFCCC (2014); CIA (2008, 2016). *Venting & flaring:*IEA (2014); Andres et al (2014), EIA (2014); UNFCCC (2014); Elvidge et al. (2009, 2016) | *Exploration:*IPCC (2006) GL *Transmission/distribution*: IPCC (2006) GL, supplemented with data of UNFCCC national inventory reports (NIR). Gas transmission through large pipelines: relatively small country-specific emission factors of Lelieveld et al. (2005); Gas distribution: large and material-dependent leakage rates of IPCC (2006) GL *Venting*: CH4 EF are based on country-specific UNFCCC NIR data (and average value as default for all other countries); *For flaring:* CO2 EF is taken from IPCC (2006) GL (and excludes indirect emissions through gas venting). | IPCC (2006b); UNFCCC (2014, 2016); Lelieveld et al. (2005) |

---

[1]We note that (1) Hard coal and brown coal data for 1970-1978 were split using the 1979 shares of the fuel types. (2) For the countries of the former Soviet Union and former Yugoslavia, the pre-1990 data was allocated to the countries using the same sector-specific country shares of the new countries from 1990. (3) We used "Serbia-Montenegro" in the dataset, which includes Kosovo and Montenegro. (4) For the lumped sum IEA regions 'Other America', 'Other Africa' and 'Other Asia', the sector- and fuel-specific activity data have been disaggregated following the IEA definition of these regions and using the total production and consumption figures per country of coal, gas and oil from energy statistics reported by the US Energy Information Administration (EIA, 2014).

[2]We note that: (1) For Iceland, Israel and Mexico this is supplemented with the biofuel consumption reported by EIA (2013). (2) For Japan, Argentina, Brasil, China, India, Indonesia, Malaysia, Peru, Philippines and Thailand the biofuel data are supplemented with the data from US DA (2014).

[3]We note that: (1) Abandoned and closed mines are taken up with very different shares to the total CH4 emissions in 2012 from coal mining in UK (with 24%), Romania, China, USA, Czech Republic, Germany and Ukraine (0.3%). (2) CH4 recovery from coal mining was estimated following IPCC (2006) for the 11 countries with largest coal mining in the past. These are in decreasing order of the share of the total CH4 emission from this sector (with absolute CH4 recovery in 2012): Czech Republic (60% with 41.1kton CH4/yr), Spain (36% with 5.0 kton/yr), Poland (33% with 157.0 kton/yr), USA (29% with 739.9 kton/yr), UK (25% with 19.1 kton/yr), Germany (24% with 45.0 kton/yr), Ukraine (16% with 97.7 kton/yr), Australia (15% with 186.0 kton/yr), China (9% with 1974.1 kton/yr), Russia (3% with 75.5 kton/yr), Kazakhstan (2% with 10.0 kton/yr).

[4]According to Peng et al. (2016) and Liu et al. (2015), Chinese underground coal mines are characterised by low quality coal and, as such, low EF, corresponding to the lower end of the range of EFs recommended by EMEP/EEA for coal mines in Europe.

| | | | | |
|---|---|---|---|---|
| Industrial processes | metallic and non-metallic minerals | Production data for cement, iron and steel, non-ferrous metals and various chemicals are based on Commodity Statistics of UN STATS (version 2014) often supplemented for recent years by USGS (2014). Iron and steel production is further split into technologies (basic oxygen furnace, open hearth, electric arc furnace) using data of the World Steel Association - WSA (2015). For production of lime, soda ash, ammonia, ferroalloys and non-ferrous metals, we combine USGS (2014) data and data reported to the UNFCCC (2014). Primary aluminium production statistics per country from UN are combined with smelter types (Horizontal and Vertical Stud Söderberg technologies as well as Centre Work, Point Feed, and Side Work Prebake technologies) characterised by the Aluminium Verlag (2007) and the International Aluminium Institute - IAI (2006). For primary magnesium production and die-casting global consumption was derived from production statistics from the US Geological Survey - USGS (2014) and the International Magnesium Association - IMA (1999) and reported country-specific die-casting companies. UN STATS (version 2014) | UN STATS (2014); USGS (2014); WSA (2015); and internal reports based on Aluminium Verlag (2007), IAI (2006), IMA (1999) | $CO_2$ from cement production is based on the Tier 1 emission factor for clinker production, whereas cement clinker production is calculated from cement production reported by USGS (2014). The implied clinker to cement ratio is based on either clinker production data from UNFCCC NIR (Annex I countries) and the China Cement Almanac, or ratios from the World Business Council for Sustainable Development - Cement Sustainability Initiative - WBCSD-CSI (2015). | IPCC (2006b), USGS (2014), UNFCCC (2014, 2016), WBCSD-CSI (2015) |
| | chemical industry | For the $CO_2$ sources from industrial production of silicon and calcium carbide, glyoxal and other chemical bulk products (acrylonitrile, black carbon, ethylene, ethylene oxide, methanol, and vinyl chloride) for which no international statistics were available, UNFCCC NIR is used, although limited to Annex I countries. Interpolations and extrapolations were only done to gap-fill single years with missing reported data in the time series 1970-2012, making use of the average of the previous and following years. Data of the International Fertilizer Industry Organisation - IFA (2015) are used for urea production, which accounts the fossil carbon in $CO_2$ from ammonia production, following IPCC (2006). Data of FAOSTAT (2014) are used for production of pulp, meat and poultry. Ammonia production data are taken from USGS (2014). UNSTATS (2014) Commodity Statistics are applied to estimate the emissions from bread production, while for paper, wine and beer we use FAO (2016c,d) production data. | UNFCCC (2014, 2016), IFA (2015), FAOSTAT (2014), USGS (2014) UN STATS (2014), FAO (2016c,d) | For the $N_2O$ sources of nitric acid, adipic acid and caprolactam, production as well as abatement data from 1990 onwards are based on UNFCCC NIR and SRI Consulting (2008). For nitric acid production in 1970, only old technology is assumed, with a gradual change in technology by 1990 into high pressure plants in non-Annex I countries and a mix of low and medium pressure plants in Annex I countries, in line with reported emissions to UNFCCC NIR. | IPCC (2006b), UNFCCC (2014, 2016), internal report based on SRI Consulting (2008) |
| Solvent statistics | | Activity data for paints, glues and adhesives, degreasing products, pesticides and vegetal oil are found in the UN Commodity statistics and supplemented with the UN Comtrade (2016) statistics details. Activity data 1990-2012 for other solvent use from UNFCCC NIR was integrated for Europe, USA, Australia and New Zealand and Japan and linearly extrapolated backwards in time. | UN Comtrade (2016), UN STATs (2014), UNFCCC (2014, 2016) | For $CO_2$, the national inventory reports of UNFCCC indicate a small amount of $CO_2$ emission per ton paint applied, or per ton degreasing and dry cleaning product or other chemical product used. $N_2O$ use as an anaesthetic and in aerosol spray cans is assumed proportional to the population. The average per capita $N_2O$ use reported by Annex I countries to UNFCCC NIR was used as region-specific default. | IPCC (2014, 2016); UNFCCC (2014, 2016) |

| Agriculture | crop (excluding rice) | Following IPCC (2006) methodology we apply FAO crop and livestock data, specified as livestock numbers for buffalo, camels, dairy and non-dairy cattle, goats, horses, swine, sheep, mules and asses and for poultry (turkeys, geese, chickens and ducks). These all contribute to manure and to enteric fermentation, except poultry that only produce manure. Historic data for countries of the former Soviet Union (1970-1990), Yugoslavia (1970-1992), Belgium and Luxemburg (1970-1999), Czechoslovakia (1970-2012) and Ethiopia and Eritrea (1970-2012) are split up, using the share in the first available year of statistics for the individual countries. Serbia and Montenegro data are merged from 2006 onwards, Sudan data are gap-filled for 2012-2014 with data of 2011 and the chicken data for Switzerland were corrected in 2007. For enteric fermentation by cattle, country specific methane emission factors are calculated following IPCC (2006) methodology using country specific milk yield (dairy cattle) and carcass weight (other cattle) trends from FAOSTAT (2014) to estimate the trends. For other animal types, regional emission factors from IPCC (2006) are used. | FAOSTAT (2014); IPCC (2006) | IPCC (2006) emission factors for $CO_2$, $CH_4$ and $N_2O$. $N_2O$ emissions from the use of animal waste as fertiliser are estimated taking into account both the loss of nitrogen that occurs from manure management systems before manure is applied to soils and the additional nitrogen introduced by bedding material. $N_2O$ emissions from fertiliser use and $CO_2$ from urea fertilisation are estimated based on IFA and FAO statistics. The $N_2O$ emission factor for direct soil emissions of $N_2O$ from the use of synthetic fertilisers and from manure used as fertilisers and from crop residues is taken from IPCC (2006), that updated the default IPCC emission factor in the IPCC Good Practice Guidance (2000) with a 20% lower value. | IPCC (2006b); FAOSTAT (2014); IFA(2015) |
|---|---|---|---|---|---|
| | Livestock | Livestock numbers of FAOSTAT (2014) are combined with estimates for animal waste per head to estimate the total amount of animal waste produced. Nitrogen excretion rates for cattle, pigs and chicken in developed countries are derived from the CAPRI model[5] for Europe (Leip et al., 2007) and for all other countries and animal types IPCC (2006) values are used. The trend in carcass weight was used to determine the trend in nitrogen excretion over time. Shares of different animal waste management systems are based on regional defaults provided in IPCC (2006) and regional trend estimates for diary and non-dairy cattle for the fractions stall-fed, extensive grazing and mixed systems from Bouwman et al. (2005). CH4 emissions from manure management are estimated by applying default IPCC emission factors for each country and temperature zone. Livestock fractions of the countries are calculated for 19 annual mean temperature zones for cattle, swine and buffalo and three climates zones for other animals (cold, temperate, warm). N2O emissions from manure management are based on distribution of manure management systems from UNFCCC NIR, Zhou et al. (2007) for China and IPCC (2006) for the rest of the countries. | FAOSTAT (2014); Leip et al. (2007); Bouwman et al. (2005); UNFCCC (2014, 2016), Zhou et al. (2007); IPCC (2006) | $CO_2$ emissions from liming of soils: from UNFCCC NIR and on the use of ammonium fertilisers for other countries from FAOSTAT (2014), as liming is needed to balance the acidity caused by ammonium fertilizers. Areas of cultivated histosols are estimated by combining the FAO climate and soil maps with the land-use map of the National Institute of Public Health and the Environment – RIVM. Different $N_2O$ emission factors are applied to tropical and non-tropical regions. Nitrogen and dry matter content of agricultural residues are estimated from the cultivation area and yield for 24 crop types (2 types of beans, barley, cassava, cereals, 3 types of peas, lentils, maize, millet, oats, 2 types of potatoes, pulses, roots and tubers, rice, rye, soybeans, sugar beet, sugar cane, sorghum, wheat and yams) from FAOSTAT (version 2014) and using emission factors of IPCC (2006). The fraction of crop residues removed from and/or burned in the field is estimated using data of Yevich and Logan (2003) and UNFCCC NIR. | IPCC (2006b); (FAO Geonetwork, 2011); (Goldewijk et al., 2007; FAOSTAT (2014); Yevich and Logan (2003) |

---

[5] The Common Agriculture Policy Regional Impact model (https://www.capri-model.org/dokuwiki/doku.php?id=start) provided tier 3 input to derive "technological parameters" and implied emission factors, representing country-specific practices.

| | | | | | |
|---|---|---|---|---|---|
| | Rice | The total area for rice cultivation, obtained from FAOSTAT (2014), is split between the different agro-ecological land-use types (rain fed, irrigated, deep water and upland) using data from the International Rice Research Institute - IRRI (2007). Methane emission factors for the various production land-uses are taken from the GAINS database (version of 2007) of the International Institute for Applied Systems Analysis - IIASA. | FAOSTAT (2014), IRRI (2007), IIASA (2007) | IPCC (2006) guidelines, for China updated with data of Peng et al. (2016) | IPCC (2006b); Peng et al. (2016) |
| solid waste product statistics | landfills | The per capita *MSW* generation rate (for 2000) and the fraction *MSW* disposed, incinerated and composted are based on the specification for 75 countries by IPCC (2006). For 151 other countries, the *MSW* generation rate and fraction disposed are assumed the same in comparable countries of the same region (with the world divided into 4 Asian regions, 5 African regions, 4 European regions, 2 regions in Oceania, 3 American regions and the Caribbean) and within the same income class (within the same GDP range).<br>The IPCC Waste Model provides for these 19 regions the average weight fraction *DOC* under aerobic conditions, which has been used as the default for all countries. For Annex I countries these three parameters have been updated after consultation of UNFCCC (2014) country-specific information on the parameters (within the expected range). The national total *MSW* reported to UNFCCC (2014) correlates best with total population (*POP*) for industrialised countries and with urban population in case of developing countries. As such the total *MSW* is calculated using the respective correlation for each country according to whether it is industrialised or developing. The DOC fraction of the total MSW landfilled is direct input to a First Order Decay (FOD) model for CH4 generation, described according to IPCC (2006) Guidelines by formula (3), and using default parameters for the Methane Correction Factor (MCF), the decomposition constant (k) and the Oxidation Factor (OX).<br>The MCF is characterised by the type of landfill: managed aerobic or anaerobic, unmanaged deep or shallow. Decomposition under anaerobic conditions is assumed to occur for 50% in the countries apart from 12 Annex I countries, which are corrected to a country-specific estimate based on their UNFCCC (2014) reports.<br>The volumetric fraction of CH4 in generated landfill gas is assumed constant and equal to 50% for all world countries. Finally, the amounts of recovered CH4 R (used or flared) are subtracted from the gross CH4 emissions, when reported in UNFCCC NIR and for 23 non-Annex I countries with CDM projects reported by UNEP Risø Centre. It is evident that these estimates are relatively uncertain, even though the source is declining considerably. | IPCC (2006c); UNFCCC (2014, 2016); UNEP Risø Centre (2011). | The MCF default is calculated as a linear variation between 0.4 and 1.0 with the urban population share and is corrected with reported data from UNFCCC NIR for 1990-2012 and linearly extrapolated backwards in time. The decomposition constant k is inversely proportional to the half-life value of the DOC and depends on climatic conditions, so that the exponential decaying reaction varies between 0.96 and 0.67. The IPCC Waste Model specifies default k values for 4 climatic zones (dry temperate, wet temperate, dry tropical and moist/wet tropical) are applied, except for the Annex I countries where the nationally measured value is selected instead. As Oonk (2010) indicated, the k-value of CH4 generation half-life or biodegradation rate is a less sensitive parameter in the emissions calculation with the FOD than the oxidation of CH4, for which data are missing. OX, which depends in part on the top layer design of the landfill and on climatic conditions, is by default zero and is only updated to a value between 3% and 10% when reported in UNFCCC NIR. | IPCC (2006c); UNFCCC (2014, 2016); Oonk (2010) |
| | incineration | Other waste sources are incineration, with activity data from UNFCCC NIR and IPCC (2006), extrapolated assuming a fixed ratio to landfilling, and secondly, composting, based on UNFCCC NIR for Annex I countries, Gupta et al. (1998) for developing countries and Sharholy et al. (2008) for India. | IPCC (2006c); UNFCCC (2014, 2016); Gupta et al. (1998); Sharholy et al. (2008) | IPCC (2006) Guidelines for combustion of fuel types | IPCC (2006b) |

| | Description | References (col 1) | Description (col 2) | References (col 2) |
|---|---|---|---|---|
| Wastewater statistics | The effect wastewater discharges have on the receiving environment depends on the oxygen required to oxidize soluble and particulate organic matter in the water and as such the Chemical Oxygen Demand (COD) and Biochemical Oxygen Demand (BOD) are used to characterise the quality of industrial and domestic wastewater. The total organically degradable material in wastewater for industry (TOWi) is estimated as kg COD /yr with country-specific data of FAO on meat, sugar, pulp, and of the Renewable Fuels Association on ethyl alcohol and 31 organic chemicals production. IPCC (2006c) default values for wastewater generation and CODs are used to derive TOWs for each industry type. The annual total population (both sexes) by country is obtained from the yearly revised UN world population prospects provides consistent time series 1950-2015 (used from 1970 onwards) for 228 world countries. With the country-specific percentage of population at mid-year residing in urban areas from the world urbanisation prospects, two additional time series, one for urban population and one for the counterpart of rural population were derived for each of the 228 countries. For domestic and commercial organically degradable material in wastewater (TOWd) we used default values for kg BOD /yr for rural and urban (low and high-income) areas and UNHABITAT country-specific shares of low-income and high-income urban population. Different wastewater treatments are specified with technology-specific CH4 emission factors. For domestic wastewater the sewer to waste water treatment plants (WWTP), sewer to raw discharge, bucket latrine, improved latrine, public or open pit and septic tank are distinguished. Regional or country-specific shares for 2000 are by default from IPCC (2006) and supplemented with data of improved sanitation over time from Van Drecht et al. (2009) and Doorn and Liles (1999). | IPCC (2006c); FAO, 2016a,b,c; UN SD (2016); Renewable Fuels Association – RFA (2016); UN DP (2014, 2015); UNHABITAT (2016a,b); World Bank (2016); Van Drecht et al. (2009); Doorn and Liles (1999). | For industrial CH4 emissions, on-site treatment in WWTP, sewer with and without city-WWTP, and raw discharge are distinguished with shares and regional emission factors from Doorn et al. (1997). For N2O, nitrogen in the effluent discharged to aquatic environments (N-effluent) was calculated following IPCC (2006) GL for each country, as a function of the human population and annual per capita protein consumption data from FAO. Other parameters are kept constant: the nitrogen fraction for protein is 0.16 kg N /kg protein (IPCC (2006) default values), the factor for non-consumed protein entering wastewater is 1.25 for the USA and 1.1 for all other countries, and the factor for industrial and commercial co-discharged protein into the sewer system is 0.25 for industrial N-effluent and 1.00 for domestic and commercial N-effluent. | IPCC (2006b); Doorn et al. (1997); FAO (2016a,b) |
| Indirect N2O | Indirect $N_2O$ emissions from leaching and runoff of nitrate are estimated from nitrogen input to agricultural soils as described above. Leaching and runoff are assumed to occur in all agricultural areas except non-irrigated dryland regions, which are identified with maps of FAO Geonetwork. Indirect N2O emissions from atmospheric deposition of nitrogen of NOx and NH3 emissions from non-agricultural sources, mainly fossil fuel combustion, are estimated using nitrogen in NOx and NH3 emissions from these sources as activity data, based on EDGAR v4.3.2 data for these gases. The same emission factor from IPCC (2006) is used for indirect N2O from atmospheric deposition of nitrogen from NH3 and NOx emissions, as for agricultural emissions. | IPCC (2006); FAO Geonetwork (2011) | The fraction of nitrogen lost through leaching and runoff is based on the study of Van Drecht et al. (2003). The updated emission factor for indirect N2O emissions from nitrogen leaching and run-off from the IPCC (2006) GL is selected, while noting that it is 70% lower than the mean value of the 1996 IPCC GL and the IPCC Good Practice Guidance (IPCC, 1997, 2000). | IPCC (2006b); Van Drecht et al. (2003). |
| fossil fuel fires | Fossil fuel fires include the Kuwait oil and gas fires with the amount of fuel burnt evaluated by Husain (1994) and the underground coal mine fires evaluated by Van Dijk et al. (2009), mainly for China and India. | Husain (1994); Van Dijk et al. (2009) | IPCC (2006) guidelines | IPCC (2006b) |

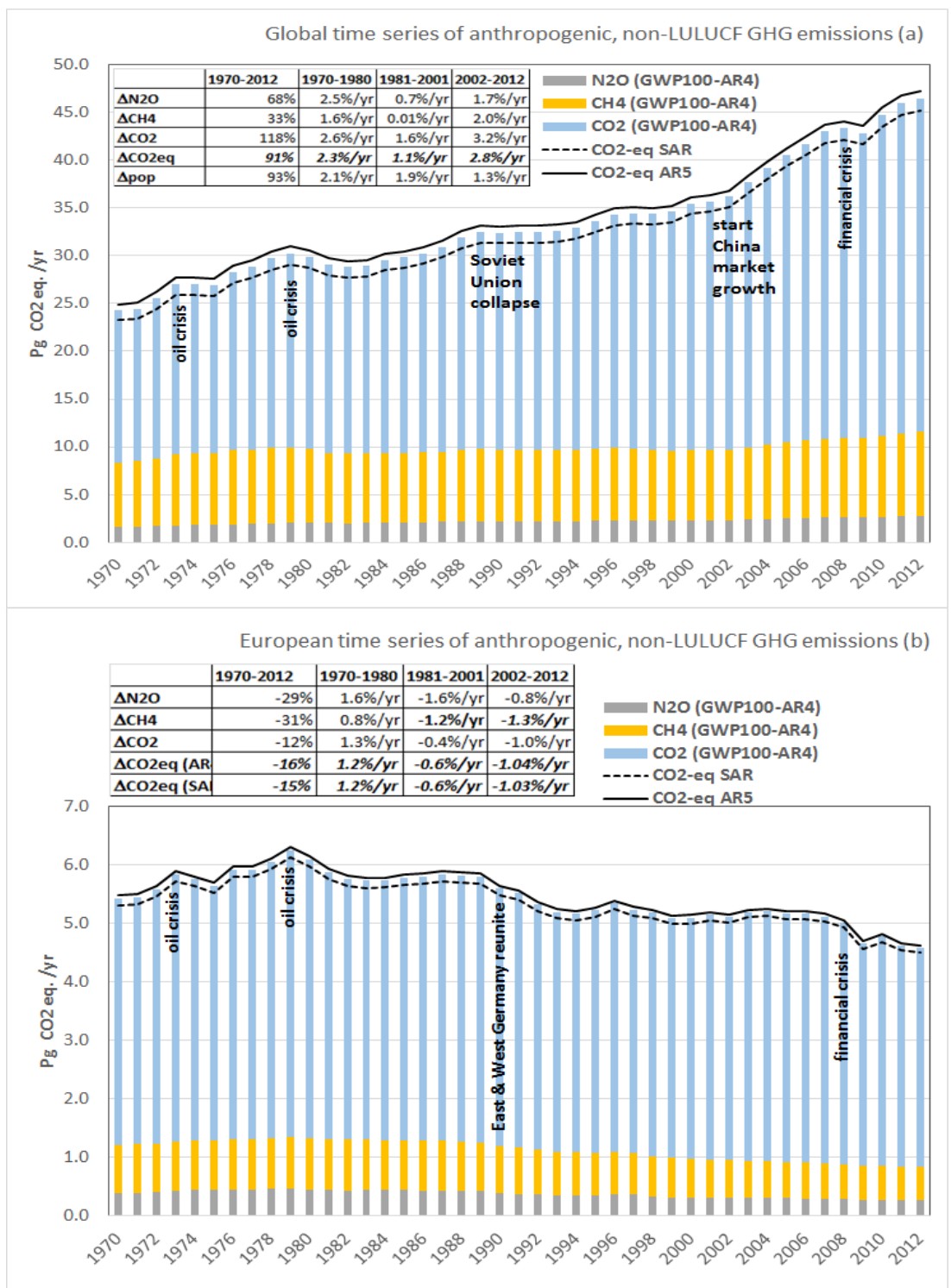

**Figure 3: (a) Timeseries 1970-2012 of fossil fuel CO$_2$, CH$_4$ and N$_2$O global emissions from human activities excluding the LULUCF sector. The stacked bars use AR4 GWP-100 values whereas the dashed line and full line indicate the total CO$_{2eq}$ of the three gases in the case the SAR and the AR5 GWP-100 values are respectively used.**

**Table 2: Relative uncertainty of the GHG inventory for countries/ country types (a) with the uncertainties per gas (b)**

| CO$_{2eq}$ | 1990 | 1995 | 2000 | 2001 | 2002 | 2003 | 2004 | 2005 | 2006 | 2007 | 2008 | 2009 | 2010 | 2011 | 2012 |
|---|---|---|---|---|---|---|---|---|---|---|---|---|---|---|---|
| 2σ China | 20.8% | 17.8% | 16.9% | 16.6% | 15.1% | 14.2% | 13.6% | 12.7% | 12.4% | 12.1% | 12.0% | 11.8% | 11.6% | 11.3% | 11.3% |
| 2σ India | 28.4% | 25.6% | 23.7% | 23.6% | 23.2% | 23.1% | 15.9% | 21.5% | 20.9% | 20.6% | 20.2% | 19.2% | 18.9% | 18.4% | 17.2% |
| 2σ Brasil | 33.4% | 33.3% | 30.2% | 30.5% | 31.1% | 31.8% | 31.8% | 30.3% | 30.1% | 29.5% | 29.2% | 30.0% | 29.0% | 28.6% | 28.3% |
| 2σ Rest_non-AnnexI | 23.4% | 22.7% | 22.1% | 21.9% | 21.8% | 21.8% | 21.7% | 21.6% | 21.5% | 21.4% | 21.3% | 21.3% | 21.1% | 21.1% | 21.1% |
| 2σ USA | 10.9% | 8.2% | 7.6% | 7.6% | 7.6% | 7.6% | 7.5% | 5.4% | 5.5% | 5.5% | 5.5% | 5.6% | 5.6% | 5.6% | 5.7% |
| 2σ EU15 | 12.7% | 10.4% | 9.6% | 9.4% | 9.3% | 9.1% | 9.0% | 5.9% | 5.9% | 5.9% | 5.9% | 6.0% | 5.9% | 6.0% | 6.0% |
| 2σ Rest_24OECD90 | 12.7% | 12.6% | 8.1% | 8.1% | 7.9% | 7.8% | 7.8% | 6.3% | 6.2% | 6.2% | 6.2% | 6.3% | 6.3% | 6.2% | 6.2% |
| 2σ Russia | 12.3% | 12.6% | 12.2% | 12.2% | 12.3% | 12.3% | 12.3% | 12.4% | 12.3% | 12.4% | 12.5% | 12.8% | 12.7% | 12.5% | 12.5% |
| 2σ EU13 | 13.0% | 12.7% | 12.8% | 12.7% | 12.9% | 12.7% | 12.8% | 10.7% | 10.5% | 10.4% | 10.5% | 11.2% | 10.7% | 10.7% | 10.8% |
| 2σ Rest_16EIT90 | 11.6% | 12.7% | 12.6% | 12.5% | 12.7% | 12.5% | 12.5% | 12.7% | 12.6% | 12.5% | 12.9% | 13.7% | 13.2% | 14.3% | 14.4% |

(a)

| CO$_2$ | 1990 | 1995 | 2000 | 2001 | 2002 | 2003 | 2004 | 2005 | 2006 | 2007 | 2008 | 2009 | 2010 | 2011 | 2012 |
|---|---|---|---|---|---|---|---|---|---|---|---|---|---|---|---|
| 2σ China | 12.0% | 12.0% | 12.0% | 12.0% | 9.0% | 9.0% | 9.0% | 9.0% | 9.0% | 9.0% | 9.0% | 9.0% | 9.0% | 9.0% | 9.0% |
| 2σ India | 12.0% | 12.0% | 12.0% | 12.0% | 12.0% | 12.0% | 12.0% | 12.0% | 12.0% | 12.0% | 12.0% | 12.0% | 12.0% | 12.0% | 9.0% |
| 2σ Brasil | 15.0% | 15.0% | 12.0% | 12.0% | 12.0% | 12.0% | 12.0% | 12.0% | 12.0% | 12.0% | 12.0% | 12.0% | 12.0% | 12.0% | 12.0% |
| 2σ Rest_non-AnnexI | 15.0% | 15.0% | 15.0% | 15.0% | 15.0% | 15.0% | 15.0% | 15.0% | 15.0% | 15.0% | 15.0% | 15.0% | 15.0% | 15.0% | 15.0% |
| 2σ USA | 10.0% | 9.0% | 9.0% | 9.0% | 9.0% | 9.0% | 9.0% | 5.0% | 5.0% | 5.0% | 5.0% | 5.0% | 5.0% | 5.0% | 5.0% |
| 2σ EU15 | 10.0% | 9.0% | 9.0% | 9.0% | 9.0% | 9.0% | 9.0% | 5.0% | 5.0% | 5.0% | 5.0% | 5.0% | 5.0% | 5.0% | 5.0% |
| 2σ Rest_24OECD90 | 10.0% | 10.0% | 9.0% | 9.0% | 9.0% | 9.0% | 9.0% | 5.0% | 5.0% | 5.0% | 5.0% | 5.0% | 5.0% | 5.0% | 5.0% |
| 2σ Russia | 10.0% | 10.0% | 10.0% | 10.0% | 10.0% | 10.0% | 10.0% | 10.0% | 10.0% | 10.0% | 10.0% | 10.0% | 10.0% | 10.0% | 10.0% |
| 2σ EU13 | 10.0% | 10.0% | 10.0% | 10.0% | 10.0% | 10.0% | 10.0% | 9.0% | 9.0% | 9.0% | 9.0% | 9.0% | 9.0% | 9.0% | 9.0% |
| 2σ Rest_16EIT90 | 10.0% | 10.0% | 10.0% | 10.0% | 10.0% | 10.0% | 10.0% | 10.0% | 10.0% | 10.0% | 10.0% | 10.0% | 10.0% | 10.0% | 10.0% |

| CH$_4$ | 1990 | 1995 | 2000 | 2001 | 2002 | 2003 | 2004 | 2005 | 2006 | 2007 | 2008 | 2009 | 2010 | 2011 | 2012 |
|---|---|---|---|---|---|---|---|---|---|---|---|---|---|---|---|
| 2σ China | 60.0% | 60.0% | 60.0% | 60.0% | 60.0% | 60.0% | 60.0% | 57.0% | 57.0% | 57.0% | 57.0% | 57.0% | 57.0% | 57.0% | 57.0% |
| 2σ India | 60.0% | 60.0% | 60.0% | 60.0% | 60.0% | 60.0% | 57.0% | 57.0% | 57.0% | 57.0% | 57.0% | 57.0% | 57.0% | 57.0% | 57.0% |
| 2σ Brasil | 60.0% | 60.0% | 60.0% | 60.0% | 60.0% | 60.0% | 60.0% | 57.0% | 57.0% | 57.0% | 57.0% | 57.0% | 57.0% | 57.0% | 57.0% |
| 2σ Rest_non-AnnexI | 60.0% | 60.0% | 60.0% | 60.0% | 60.0% | 60.0% | 60.0% | 60.0% | 60.0% | 60.0% | 60.0% | 60.0% | 60.0% | 60.0% | 60.0% |
| 2σ USA | 60.0% | 57.0% | 57.0% | 57.0% | 57.0% | 57.0% | 57.0% | 32.0% | 32.0% | 32.0% | 32.0% | 32.0% | 32.0% | 32.0% | 32.0% |
| 2σ EU15 | 60.0% | 57.0% | 57.0% | 57.0% | 57.0% | 57.0% | 57.0% | 32.0% | 32.0% | 32.0% | 32.0% | 32.0% | 32.0% | 32.0% | 32.0% |
| 2σ Rest_24OECD90 | 60.0% | 60.0% | 57.0% | 57.0% | 57.0% | 57.0% | 57.0% | 32.0% | 32.0% | 32.0% | 32.0% | 32.0% | 32.0% | 32.0% | 32.0% |
| 2σ Russia | 60.0% | 60.0% | 60.0% | 60.0% | 60.0% | 60.0% | 60.0% | 60.0% | 60.0% | 60.0% | 60.0% | 60.0% | 60.0% | 60.0% | 60.0% |
| 2σ EU13 | 60.0% | 60.0% | 60.0% | 60.0% | 60.0% | 60.0% | 60.0% | 57.0% | 57.0% | 57.0% | 57.0% | 57.0% | 57.0% | 57.0% | 57.0% |
| 2σ Rest_16EIT90 | 60.0% | 60.0% | 60.0% | 60.0% | 60.0% | 60.0% | 60.0% | 60.0% | 60.0% | 60.0% | 60.0% | 60.0% | 60.0% | 60.0% | 60.0% |

| N$_2$O | 1990 | 1995 | 2000 | 2001 | 2002 | 2003 | 2004 | 2005 | 2006 | 2007 | 2008 | 2009 | 2010 | 2011 | 2012 |
|---|---|---|---|---|---|---|---|---|---|---|---|---|---|---|---|
| 2σ China | 100.0% | 100.0% | 100.0% | 100.0% | 100.0% | 100.0% | 100.0% | 93.0% | 93.0% | 93.0% | 93.0% | 93.0% | 93.0% | 93.0% | 93.0% |
| 2σ India | 100.0% | 100.0% | 100.0% | 100.0% | 100.0% | 100.0% | 93.0% | 93.0% | 93.0% | 93.0% | 93.0% | 93.0% | 93.0% | 93.0% | 93.0% |
| 2σ Brasil | 100.0% | 100.0% | 100.0% | 100.0% | 100.0% | 100.0% | 100.0% | 93.0% | 93.0% | 93.0% | 93.0% | 93.0% | 93.0% | 93.0% | 93.0% |
| 2σ Rest_non-AnnexI | 100.0% | 100.0% | 100.0% | 100.0% | 100.0% | 100.0% | 100.0% | 100.0% | 100.0% | 100.0% | 100.0% | 100.0% | 100.0% | 100.0% | 100.0% |
| 2σ USA | 100.0% | 93.0% | 93.0% | 93.0% | 93.0% | 93.0% | 93.0% | 42.0% | 42.0% | 42.0% | 42.0% | 42.0% | 42.0% | 42.0% | 42.0% |
| 2σ EU15 | 100.0% | 93.0% | 93.0% | 93.0% | 93.0% | 93.0% | 93.0% | 42.0% | 42.0% | 42.0% | 42.0% | 42.0% | 42.0% | 42.0% | 42.0% |
| 2σ Rest_24OECD90 | 100.0% | 100.0% | 93.0% | 93.0% | 93.0% | 93.0% | 93.0% | 42.0% | 42.0% | 42.0% | 42.0% | 42.0% | 42.0% | 42.0% | 42.0% |
| 2σ Russia | 100.0% | 100.0% | 100.0% | 100.0% | 100.0% | 100.0% | 100.0% | 100.0% | 100.0% | 100.0% | 100.0% | 100.0% | 100.0% | 100.0% | 100.0% |
| 2σ EU13 | 100.0% | 100.0% | 100.0% | 100.0% | 100.0% | 100.0% | 100.0% | 93.0% | 93.0% | 93.0% | 93.0% | 93.0% | 93.0% | 93.0% | 93.0% |
| 2σ Rest_16EIT90 | | 100.0% | 100.0% | 100.0% | 100.0% | 100.0% | 100.0% | 100.0% | 100.0% | 100.0% | 100.0% | 100.0% | 100.0% | 100.0% | 100.0% |

(b)

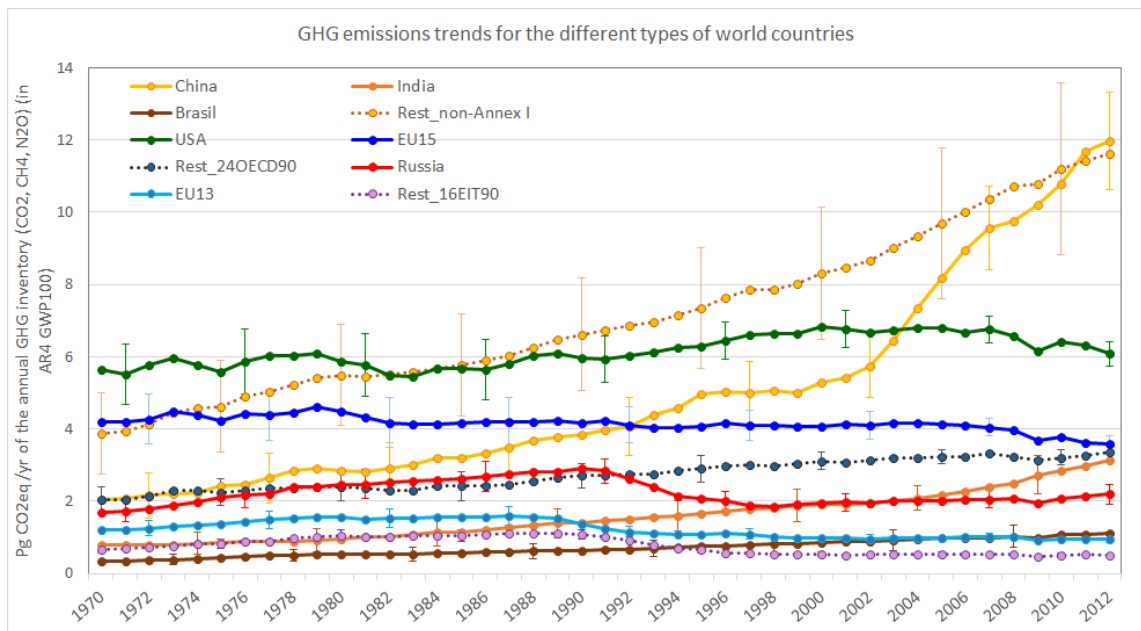

**Figure 4: Annual greenhouse gas time series 1970-2012 of EDGARv4.3.2 with periodic error bar indication for the different types of countries with top emitters: (i) non-Annex I countries with China, India, Brasil and Rest of non-Annex I countries, (ii) 24OECD90 countries with USA, EU15 and the remaining 8 OECD countries of 1990, (iii) 16EIT90 countries with Russia, EU13 and the remaining 2 newly independent Eurasian states. For the figures per gas we refer to figures S4a-c in the Supplementary.**

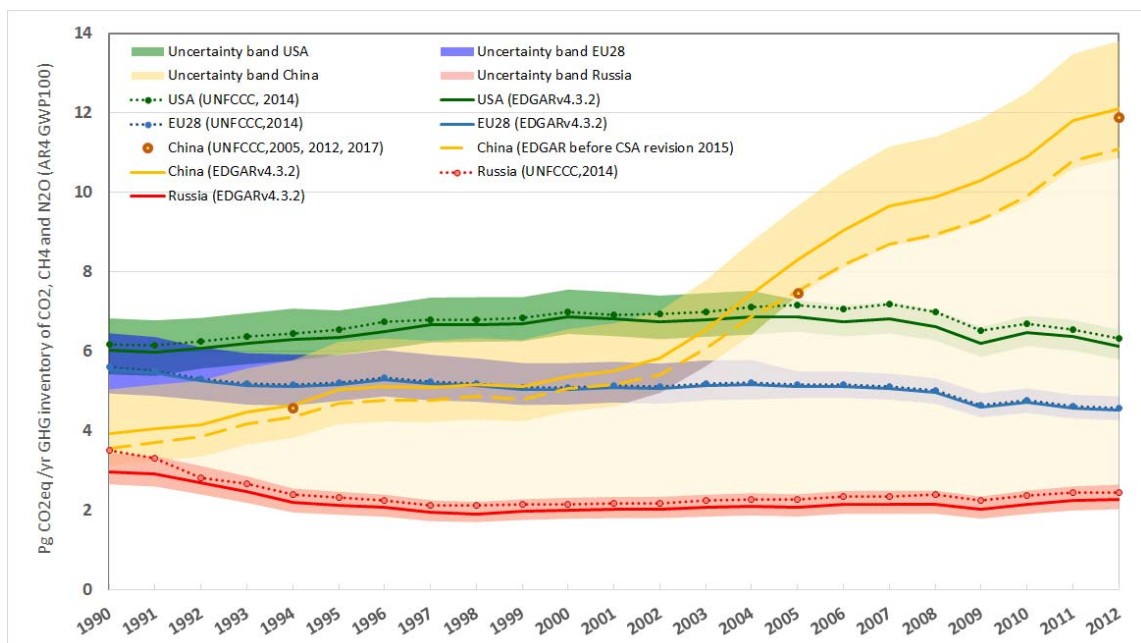

**Figure 5: GHG emissions of largest emitting countries and regions (USA, EU28, Russia, China) of EDGARv4.3.2 (solid line) with their uncertainty band compared to the reported UNFCCC time series of 2016 (dotted line). For China, two inventories were reported by national communications (1994, 2005) and a biennial update in 2017 added a new inventory value for 2012. The dashed yellow line gives the EDGARv4.3.1 estimate of the Chinese GHG emissions using the energy statistics before the Coal Statistics Abstract (CSA) revision of October 2015.**

**Table 3: Intercomparison of eight global $CO_2$ datasets (GCP, Le Quéré et al., 2016; PKU-FUEL, Wang et al., 2013; ODIAC2016, Oda et al., 2018; CDIAC, Andres et al., 2014; EIA, 2014; IEA, 2014; BP, 2016) with regard to their spatial and temporal coverage and their estimate of the global CO2 totals per source for 2010 (and 2007 for PKU-FUEL).**

| $CO_2$ totals in Pg/yr for 2010 | EDGARv4.3.2 | GCP[6] | PKU-FUEL (-CO2) | ODIAC2016[6] |
|---|---|---|---|---|
| Time series | 1970-2012 , fast track to 2015 | 1959-2015 | 2007 | 2000-2016 |
| spatial resolution | 0.1° x 0.1° | | 0.1° x 0.1° | 1*km* x 1*km* |
| temporal resolution | Monthly | Annual | Annual | Monthly |
| Geo-coverage | 226 countries | Global | 223 countries | global |
| activity split | 150 activities, 42 fossil and 15 bio fuels) | 5 main sectors, 42 fuel types | 64 fuel types | 6 data inputs (based on nighttime light, CARMA and CDIAC) |
| fossil fuel combustion | **30.5 (±5.3) 95%CI** | **Bottom-up estimate: 34.5 [Top down estimate: 35.6]** | **28.71** | **33.4** |
| non-combustion | **3.1 (±1.6) 95%CI** | | | **1.6** |
| CO2 totals in Pg/yr for 2010 | **CDIAC** | **EIA** | **IEA[7]** | **BP[6]** |
| Time series | 1751-2014 | 1980-2011 | 1971-2014 | 1965-2015 |
| temporal resolution | annual | Annual | annual | Annual |
| Geo-coverage | 224 countries | 224 countries | 137 countries, 3 regions | 67 countries, 5 regions |
| activity split-up | 5 main sectors, 42 fuel types | 6 main sectors, 42 fuel types | 64 activities, 42 fossil and 15 bio fuels) | 8 activities, 3 fossil and 3 other fuel types |
| fossil fuel combustion | **32.7** | **31.6** | **31.0** | **33.5** |
| non-combustion | **1.6** | | | |

---

[6] GCP, ODIAC and BP have used more recent energy statistics than EDGAR and IEA (2014), which explains the major difference in global CO2 emissions between them.

[7] The difference in the calculation for IEA and EDGAR are mainly the different carbon factors used: IPCC (1996) for IEA and IPCC (2006) for EDGAR. In addition, EDGARv4 supplements the charcoal production activity with fuelwood data of FAO, the venting and flaring activity with satellite data and the fossil fuel mine gas recovery with UNFCCC data, and EDGARv4 calculates the transformation losses which IEA neglects. The main difference meanwhile disappeared as IEA updated the carbon factors with IPCC (2006) values.

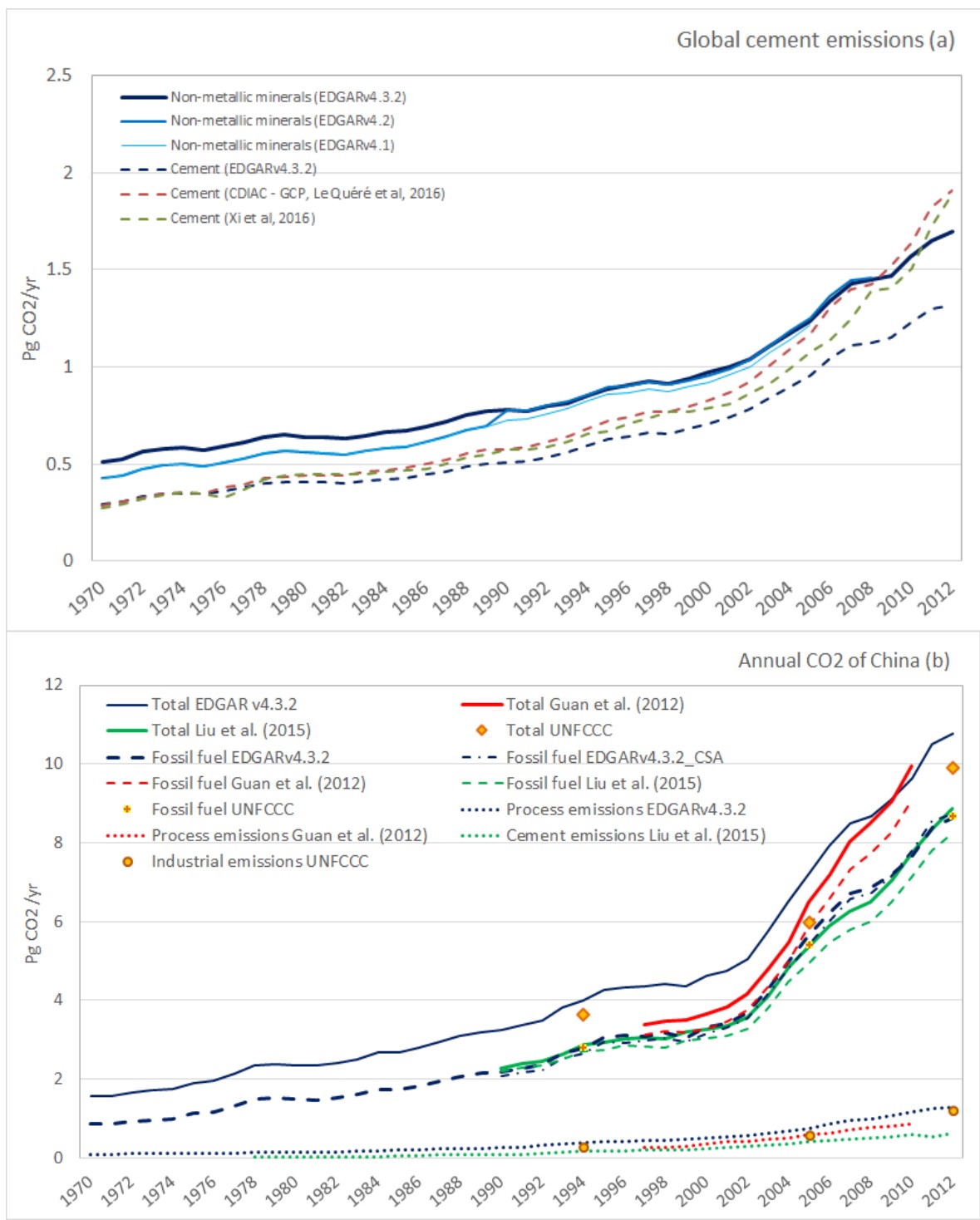

**Figure 6: Intercomparison of CO₂ emissions trends estimated by EDGAR and by others with: (a) details for cement process emissions globally with data of Le Quéré et al. (2016) and Xi et al. (2016), (b) details for China's sector-specific emissions with data of Guan et al. (2012) and Liu et al. (2015). Total is for all datasets subdivided into Fossil fuel combustion and Industrial process emissions (i.e. non-combustion industrial emissions, including cement)**

**Table 4: Intercomparison of the global total Pg CH₄ in 2010 by EDGARv4.3.2 and by four other global emission inventories: USEPA (2012), GAINS-ECLIPSEv5 CH₄ of Höglund-Isaksson et al. (2015), Kirschke et al. (2013) and the global methane budget of Saunois et al. (2016).** Note that the sector-specific global total is given in Tg CH4/yr for 2010 and in brackets for 2000. USEPA 2010 value is projected. For Kirschke, instead of 2010 (2000) we used the Maximum (Minimum) of the 2000-2009 range. For Saunois we used instead of 2010 [2000] the 2012 value [mean value of the 2000-2009 range]. The 2010 values are bold, the 2000 values are in italics.

| CH4 totals in Tg/yr for 2010 *[2000]* | EDGARv4.3.2 | USEPA (2012) | GAINS ECLIPSEv5 (2015) | Kirschke et al. (2013) Bottom up [Top down] | Saunois et al. (2016) Bottom up [Top down] |
|---|---|---|---|---|---|
| Time series | 1970-2012 | 1990-2005 (projected to 2030) | 1990-2010 | 1980-2009 | 2000-2012 |
| spatial resolution | 0.1°x0.1° | None | 1°x1° | | |
| temporal resolution | monthly | Annual | Annual | annual | Annual |
| Geo-coverage | 227 countries | 224 countries | 77 countries & 5 regions | global | Global |
| Agricultural sector | **154 (±92)** *[137]* | **147** *[136]* | **129** *[123]* | **Bottom up: 219** *[263]* **[Top down: 286** *[204]***]** | **Bottom up: 197** *[190]* **[Top down: 200** *[183]***]** |
| Waste & wastewater | **67 (±61)** *[59]* | **65** *[58]* | **51** *[46]* | | |
| energy and fossil fuel production | **121 (±91)** *[96]* | **129** *[107]* | **144** *[116]* | **Bottom up: 105** *(85)* **[Top down: 123** *[77]***]** | **Bottom up 164** *[142]* **[Top down: 147** *[136]***]** |
| Other | **21 (±20)** *[18]* | | **19** *[17]* | - | - |
| Total | **342 (±160)** *[293]* | | **342** *[302]* | **Bottom up: 368** *[304]* **[Top down: 409** *[273]***]** | **Bottom-up: 370** *[338]* **[Top down: 347** *[319]***]** |

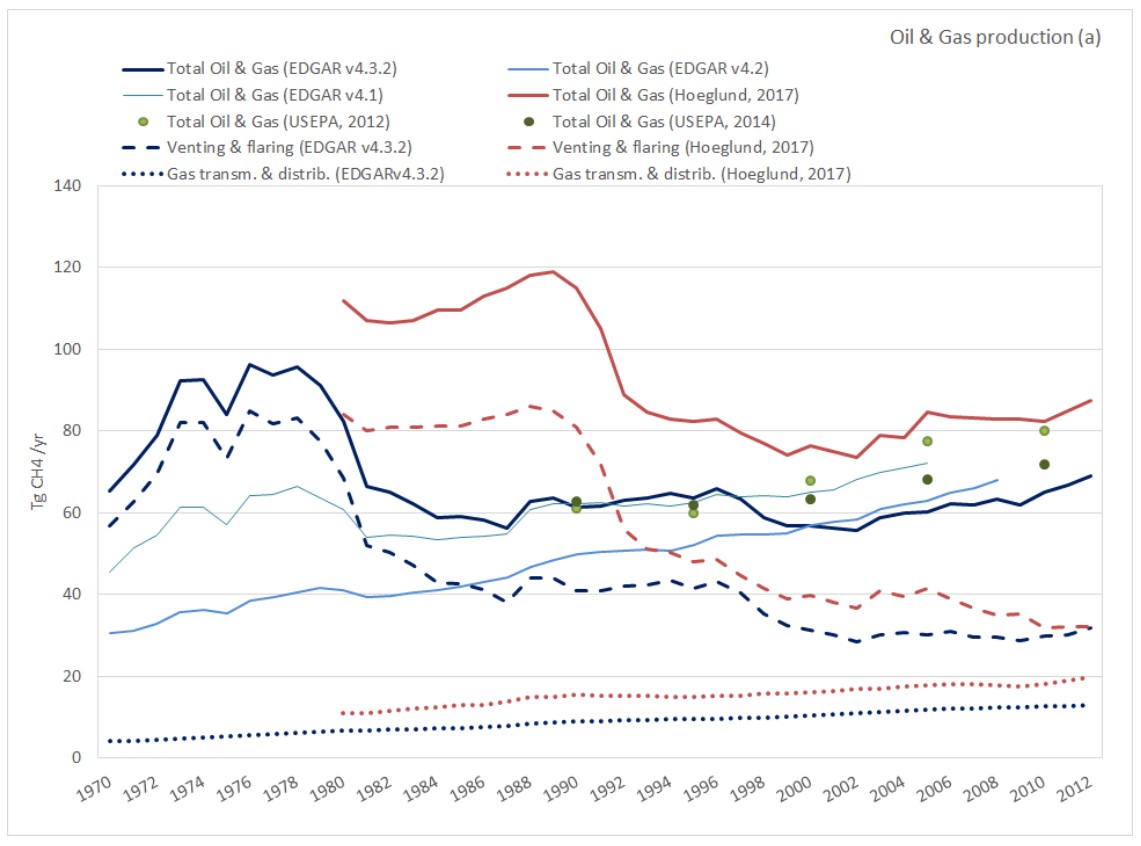

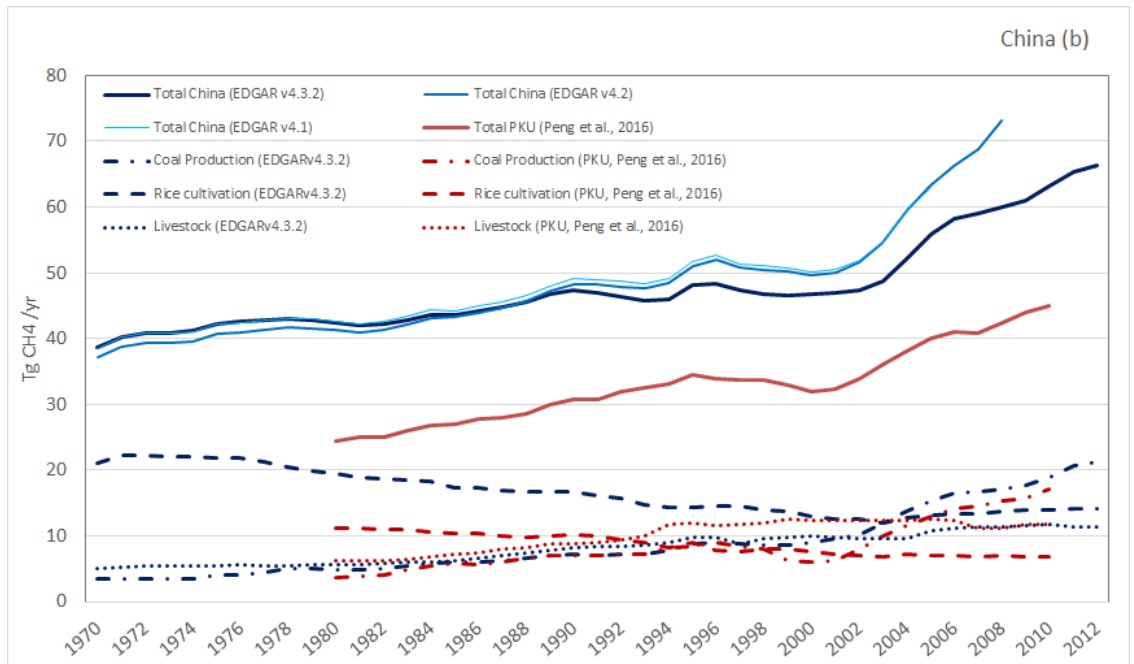

**Figure 7: Intercomparison of CH₄ emissions trends estimated by EDGAR and by others with: (a) details for the CH4 venting for oil and gas extraction, transmission and distribution with data of Höglund-Isaksson (2017) and (b) details for China's sector-specific emissions with data of Peng et al. (2016)**

**Table 5: Intercomparison of the global *[EU]* total Tg N2O in 2005 by EDGARv4.3.2 and by other European and global inventories: The European N Assessment of Leip et al. (2011) for EU27, GAINS Europe of Höglund-Isaksson et al. (2010) and GAINS global of Winiwarter et al. (2018), global total of USEPA (2012). The global values are bold, the European values are given in *[italics]* between brackets.**

| N2O totals in Tg/yr for 2005 global *[EU]* | EDGARv4.3.2 global *[EU27]* | N-Budget *[EU27]* | GAINS global *[EU27]* | USEPA (2012) Global |
|---|---|---|---|---|
| timeseries | 1970-2012 | 2000-2007 | 1990-2015 (projected to 2030) | 1990-2005 (projected to 2030) |
| spatial resolution | 0.1° x 0.1° | 1km x1km | | |
| temporal resolution | Monthly | Annual | 5-yearly | Annual |
| Geocoverage | 226 countries *[27 countries in Europe in 2005]* | *[27 countries in Europe in 2005]* | 172 countries/regions *[27 countries in Europe in 2005]* | global |
| Agriculture | **4.63 (±3.6)** *[0.43 (±0.23)]* | *[0.68]* | **5.71** *[0.87]* | **1.95** |
| Non-Agriculture | **2.54 (±2.5)** *[0.37 (±0.35)]* | *[0.31]* | **1.97** *[0.44]* | **8.91** |
| Total | **7.16 (±6.7)** *[0.80 (±0.45)]* | *[1.08]* | **7.68** *[1.30]* | **10.86** |

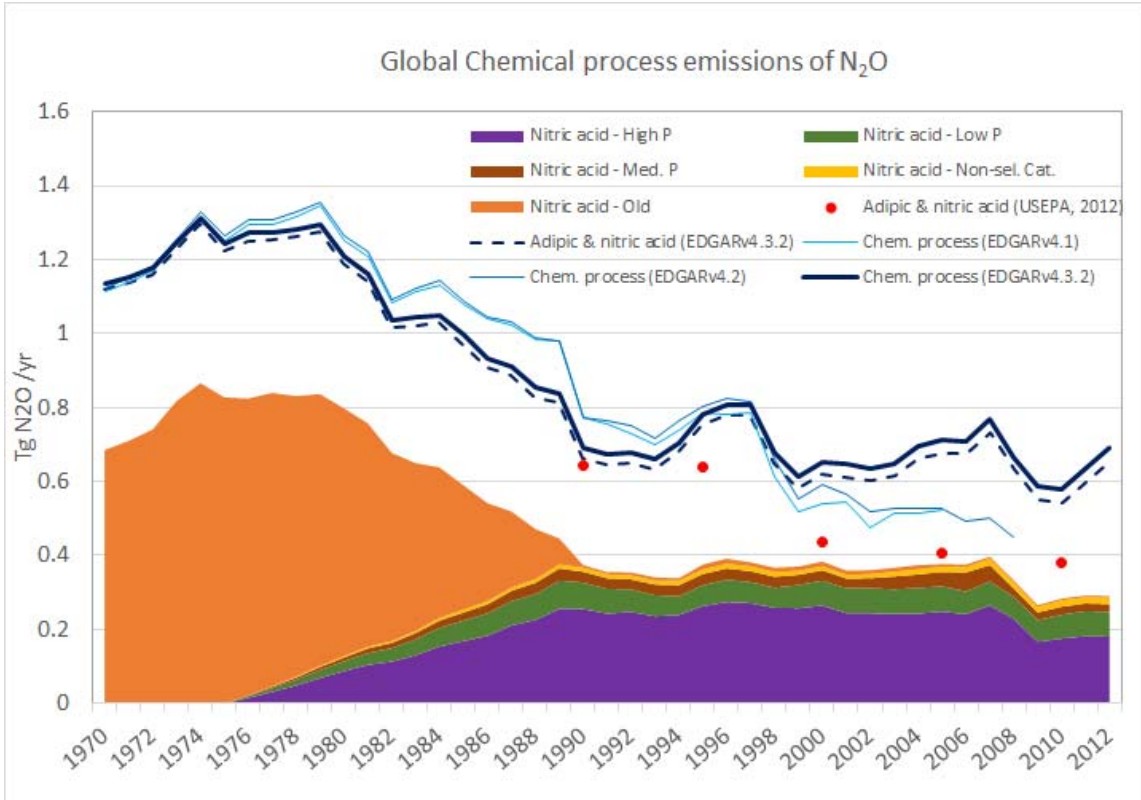

**Figure 8: Global N2O emissions trends for chemical processes, which are mainly originating from Nitric and Adipic Acid Production (aside of smaller contributions from Glycoxal and Caprolactam Production) The coloured area illustrates the penetration of technology for nitric acid production (with High Pressure plants, Medium Pressure plants, Low Pressure plants, plants with Non-Selective Catalytic Reduction and Old plants) to reduce the emissions.**

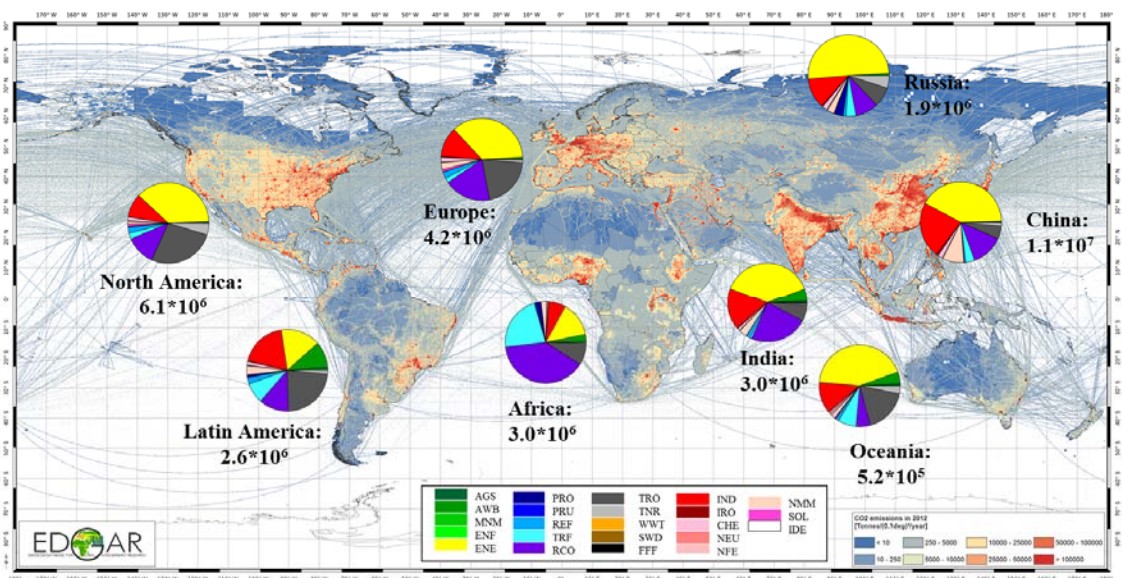

**Figure 9: CO₂ emission grid-map and relative contribution of EDGAR sectors in world regions (pie charts) for 2012.**
The legend for the PIE charts relate to the EDGAR sectors, defined in Table S3: AGS= agricultural soils, AWB=agricultural waste burning, MNM=manure management, ENF=enteric fermentation, ENE=power industry, PRO=fuel production, PRU=production& use of products, REF=oil refineries, TRF=transformation industry, RCO=residential, TRO=road transport, TNR=non-road transport, WWT=waste water, SWD=solid waste disposal, FFF=fossil fuel fires, IND=manufacturing industry, IRO=iron & steel, CHE=chemicals, NEU=non-energy use, NFE=non-ferrous metals, NMM=non-metallic minerals,

SOL=solvents, IDE=indirect emissions. **The represented CO2 emissions include also those from short-cycle carbon (i.e. of e.g. biofuel combustion and agricultural waste burning).**

**Table 6a - Global and regional GHG emissions (in ktons and tons/person) for the year 2012. CO$_{2eq}$ emissions have been calculated including only CO$_2$ from long-cycle carbon only, CH$_4$ and N$_2$O.**

| year 2012 | CO2 long cycle C | CO2 short cycle C | CH4 | N2O | CO2eq (AR5) | CO2eq (AR4) | CO2eq (SAR) | CO2eq (AR4)/cap |
|---|---|---|---|---|---|---|---|---|
| Canada | 5.64E+05 | 5.33E+04 | 4.68E+03 | 1.23E+02 | 7.28E+05 | 7.18E+05 | 7.00E+05 | 20.6 |
| USA | 5.20E+06 | 3.10E+05 | 2.58E+04 | 9.44E+02 | 6.18E+06 | 6.13E+06 | 6.04E+06 | 19.5 |
| Mexico | 4.84E+05 | 5.23E+04 | 5.20E+03 | 3.73E+02 | 7.29E+05 | 7.26E+05 | 7.09E+05 | 5.9 |
| Rest Central America | 1.71E+05 | 9.63E+04 | 3.60E+03 | 8.54E+01 | 2.95E+05 | 2.87E+05 | 2.73E+05 | 3.3 |
| Brazil | 4.73E+05 | 5.20E+05 | 1.92E+04 | 5.63E+02 | 1.16E+06 | 1.12E+06 | 1.05E+06 | 5.5 |
| Rest South America | 6.61E+05 | 1.59E+05 | 1.62E+04 | 4.07E+02 | 1.22E+06 | 1.19E+06 | 1.13E+06 | 5.8 |
| Northern Africa | 4.87E+05 | 1.68E+04 | 7.20E+03 | 1.40E+02 | 7.25E+05 | 7.08E+05 | 6.81E+05 | 4.1 |
| Western Africa | 1.71E+05 | 9.14E+05 | 1.57E+04 | 2.77E+02 | 6.83E+05 | 6.45E+05 | 5.86E+05 | 1.5 |
| Eastern Africa | 5.51E+04 | 5.53E+05 | 1.15E+04 | 3.33E+02 | 4.65E+05 | 4.42E+05 | 4.00E+05 | 1.6 |
| Southern Africa | 4.49E+05 | 3.95E+05 | 8.21E+03 | 1.94E+02 | 7.30E+05 | 7.12E+05 | 6.82E+05 | 3.5 |
| OECD Europe | 3.08E+06 | 3.74E+05 | 1.83E+04 | 7.10E+02 | 3.78E+06 | 3.75E+06 | 3.68E+06 | 9.1 |
| Central Europe | 8.51E+05 | 1.08E+05 | 6.41E+03 | 2.39E+02 | 1.09E+06 | 1.08E+06 | 1.06E+06 | 8.7 |
| Turkey | 3.40E+05 | 3.37E+04 | 3.76E+03 | 1.56E+02 | 4.87E+05 | 4.80E+05 | 4.67E+05 | 6.4 |
| Ukraine + | 3.93E+05 | 2.45E+04 | 3.46E+03 | 1.61E+02 | 5.32E+05 | 5.27E+05 | 5.15E+05 | 9.0 |
| Asia-Stan | 4.52E+05 | 6.00E+03 | 7.75E+03 | 1.12E+02 | 6.99E+05 | 6.79E+05 | 6.50E+05 | 10.6 |
| Russia + | 1.82E+06 | 3.29E+04 | 1.84E+04 | 2.35E+02 | 2.39E+06 | 2.35E+06 | 2.28E+06 | 14.7 |
| Middle_East | 1.84E+06 | 8.65E+03 | 2.05E+04 | 2.17E+02 | 2.48E+06 | 2.42E+06 | 2.34E+06 | 10.7 |
| India + | 2.34E+06 | 1.19E+06 | 4.70E+04 | 1.10E+03 | 3.95E+06 | 3.85E+06 | 3.67E+06 | 2.3 |
| Korea | 6.61E+05 | 1.08E+04 | 2.41E+03 | 5.26E+01 | 7.43E+05 | 7.37E+05 | 7.28E+05 | 9.9 |
| China + | 1.03E+07 | 8.50E+05 | 6.76E+04 | 1.78E+03 | 1.26E+07 | 1.25E+07 | 1.22E+07 | 9.0 |
| South-East Asia | 8.03E+05 | 5.43E+05 | 1.93E+04 | 2.97E+02 | 1.42E+06 | 1.37E+06 | 1.30E+06 | 3.8 |
| Indonesia + | 4.53E+05 | 3.28E+05 | 1.21E+04 | 2.58E+02 | 8.61E+05 | 8.33E+05 | 7.88E+05 | 3.3 |
| Japan | 1.30E+06 | 5.36E+04 | 1.85E+03 | 7.56E+01 | 1.37E+06 | 1.37E+06 | 1.36E+06 | 10.8 |

| | | | | | | | | |
|---|---|---|---|---|---|---|---|---|
| Oceania | 4.67E+05 | 4.90E+04 | 6.49E+03 | 2.07E+02 | 7.04E+05 | 6.91E+05 | 6.68E+05 | 22.7 |
| Internat. Shipping | 6.09E+05 | 1.49E+02 | 4.92E+02 | 8.44E+01 | 6.45E+05 | 6.46E+05 | 6.45E+05 | 0.1 |
| Internat. Aviation | 4.83E+05 | | 3.38E+00 | 2.36E+01 | 4.89E+05 | 4.90E+05 | 4.90E+05 | 0.1 |
| Totals | 3.49E+07 | 6.68E+06 | 3.53E+05 | 9.15E+03 | 4.72E+07 | 4.64E+07 | 4.51E+07 | 6.5 |

**Table 6b - Global sector-specific GHG emissions for the year 2012 (in ktons and tons/person). $CO_{2eq}$ emissions have been calculated including only $CO_2$ from long-cycle carbon only, $CH_4$ and $N_2O$. *Note that emissions from the Supersonic aviation are available only till the year 2003, when the Concorde airplanes stopped flying.**

| EDGAR SECTOR | DESCRIPTION | CO2 long cycle C | CO2 short cycle C | CH4 | N2O | CO2eq (AR5) | CO2eq (AR4) | CO2eq (SAR) | CO2eq(AR4)/cap |
|---|---|---|---|---|---|---|---|---|---|
| AGS | Agricultural soils | 1.6E+05 | | 3.8E+04 | 5.0E+03 | 2.5E+06 | 2.6E+06 | 2.5E+06 | 0.36 |
| AWB | Agricultural waste burning | | 1.0E+06 | 1.8E+03 | 4.6E+01 | 6.2E+04 | 5.9E+04 | 5.2E+04 | 0.01 |
| CHE | Chemical processes | 6.8E+05 | | 2.8E+02 | 6.9E+02 | 8.7E+05 | 8.9E+05 | 9.0E+05 | 0.13 |
| ENE | Power industry | 1.4E+07 | 4.9E+05 | 3.8E+02 | 2.8E+02 | 1.4E+07 | 1.4E+07 | 1.4E+07 | 1.95 |
| ENF | Enteric fermentation | | | 1.0E+05 | | 2.9E+06 | 2.6E+06 | 2.2E+06 | 0.37 |
| FFF | Fossil Fuel Fires | 4.7E+04 | | 1.5E+02 | 7.5E-01 | 5.2E+04 | 5.1E+04 | 5.1E+04 | 0.01 |
| FOO_PAP | Food and Paper | | | | | | | | 0.00 |
| IND | Combustion for manufacturing | 5.5E+06 | 7.4E+05 | 5.6E+02 | 7.6E+01 | 5.6E+06 | 5.6E+06 | 5.6E+06 | 0.79 |
| IRO | Iron and steel production | 2.2E+05 | | 5.2E+01 | | 2.2E+05 | 2.2E+05 | 2.2E+05 | 0.03 |
| MNM | Manure management | | | 1.2E+04 | 3.4E+02 | 4.2E+05 | 4.0E+05 | 3.5E+05 | 0.06 |
| NEU | Non energy use of fuels | 2.5E+04 | | | | 2.5E+04 | 2.5E+04 | 2.5E+04 | 0.003 |
| NFE | Non-ferrous metals production | 8.1E+04 | | | | 8.1E+04 | 8.1E+04 | 8.1E+04 | 0.01 |
| NMM | Non-metallic minerals production | 1.7E+06 | | | | 1.7E+06 | 1.7E+06 | 1.7E+06 | 0.24 |
| PRO | Fuel exploitation | 2.2E+05 | | 1.1E+05 | 3.3E+00 | 3.2E+06 | 2.9E+06 | 2.5E+06 | 0.41 |
| PRU_SOL | Solvents and products use | 1.7E+05 | | | 8.6E+01 | 1.9E+05 | 1.9E+05 | 2.0E+05 | 0.03 |
| RCO | Energy for buildings | 3.3E+06 | 3.4E+06 | 1.4E+04 | 2.7E+02 | 3.7E+06 | 3.7E+06 | 3.6E+06 | 0.52 |
| REF_TRF | Oil refineries and | 1.8E+06 | 8.7E+05 | 6.0E+03 | 2.1E+01 | 2.0E+06 | 1.9E+06 | 1.9E+06 | 0.27 |

| | | | | | | | | |
|---|---|---|---|---|---|---|---|---|
| | Transformation industry | | | | | | | |
| SWD_INC | Solid waste incineration | 1.1E+04 | 1.5E+04 | 1.3E+03 | 4.0E+00 | 4.9E+04 | 4.5E+04 | 4.0E+04 | 0.01 |
| SWD_LDF | Solid waste landfills | | | 2.9E+04 | 1.1E+01 | 8.2E+05 | 7.3E+05 | 6.2E+05 | 0.10 |
| TNR_Aviation_CDS | Aviation climbing&descent | 2.9E+05 | | 2.0E+00 | 8.1E+00 | 2.9E+05 | 2.9E+05 | 2.9E+05 | 0.04 |
| TNR_Aviation_CRS | Aviation cruise | 3.9E+05 | | 2.7E+00 | 1.1E+01 | 3.9E+05 | 3.9E+05 | 3.9E+05 | 0.06 |
| TNR_Aviation_LTO | Aviation landing&takeoff | 9.3E+04 | | 6.5E-01 | 2.6E+00 | 9.4E+04 | 9.4E+04 | 9.4E+04 | 0.01 |
| *TNR_Aviation_SPS | Aviation supersonic | | | | | | | | |
| TNR_Other | Railways, pipelines, off-road transport | 2.6E+05 | 7.5E+02 | 8.7E+00 | 3.8E+01 | 2.7E+05 | 2.7E+05 | 2.7E+05 | 0.04 |
| TNR_Ship | Shipping | 7.8E+05 | 1.6E+02 | 7.1E+01 | 2.0E+01 | 7.9E+05 | 7.9E+05 | 7.9E+05 | 0.11 |
| TRO | Road transportation | 5.4E+06 | 1.7E+05 | 8.0E+02 | 2.3E+02 | 5.5E+06 | 5.5E+06 | 5.5E+06 | 0.78 |
| WWT | Waste water handling | | | 3.8E+04 | 3.5E+02 | 1.2E+06 | 1.1E+06 | 9.1E+05 | 0.15 |
| IDE | Indirect emissions | | | | 6.2E+02 | 1.6E+05 | 1.8E+05 | 1.9E+05 | 0.03 |
| N2O | Indirect N2O emissions | | | | 1.1E+03 | 2.8E+05 | 3.2E+05 | 3.3E+05 | 0.04 |

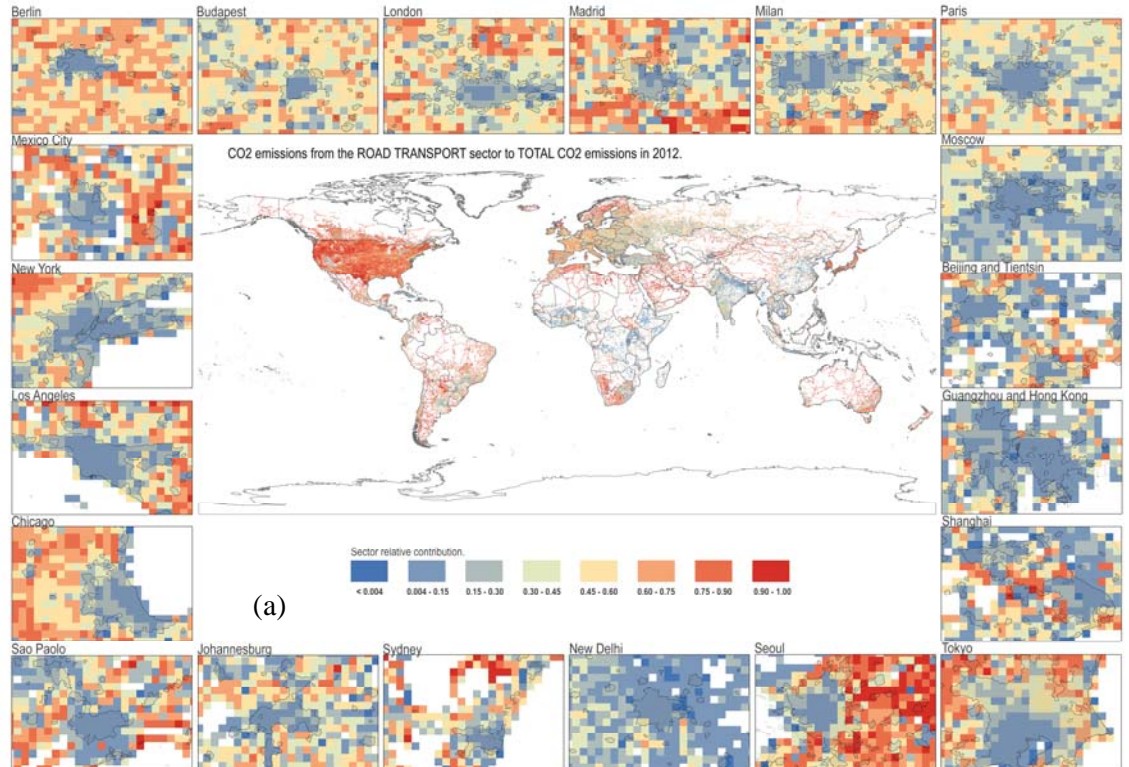

**Figure 10a: Zoom of CO₂ emission grid-maps over cities, representing the share of the road transport within the cities. The represented CO₂ emissions include also those from short-cycle carbon.**

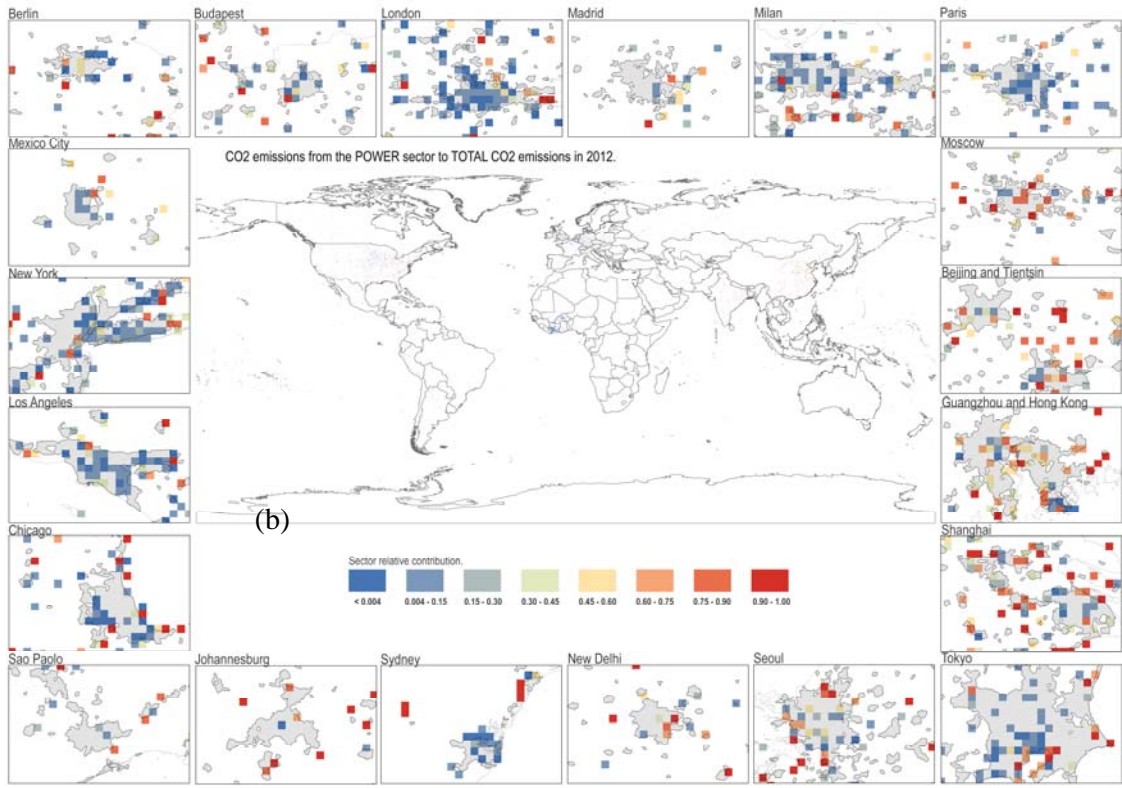

**Figure 10b: Zoom of CO₂ emission grid-maps over cities, representing the share of the power plants within the cities. The represented CO₂ emissions include also those from short-cycle carbon.**

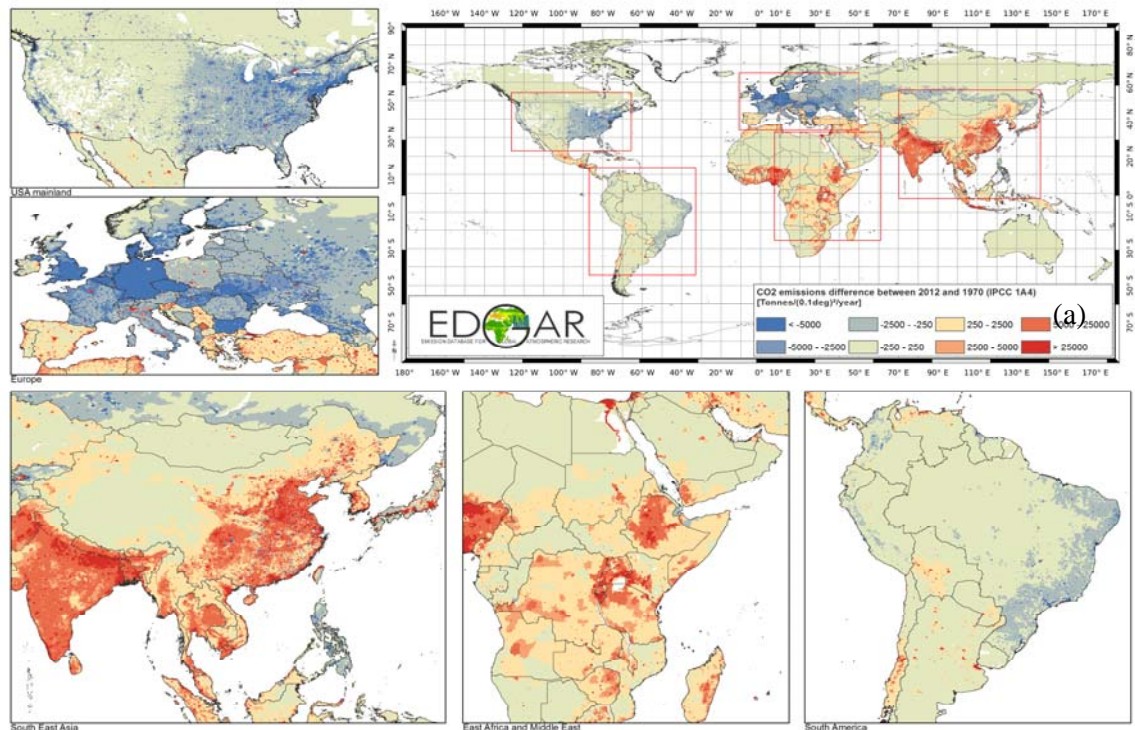

**Figure 11a: Difference in CO₂ emissions from buildings between 2012 and 1970. The represented CO₂ emissions include also those from short-cycle carbon. The figures for the long-cycle and short-cycle carbon separately are taken up in the Supplementary, S5.**

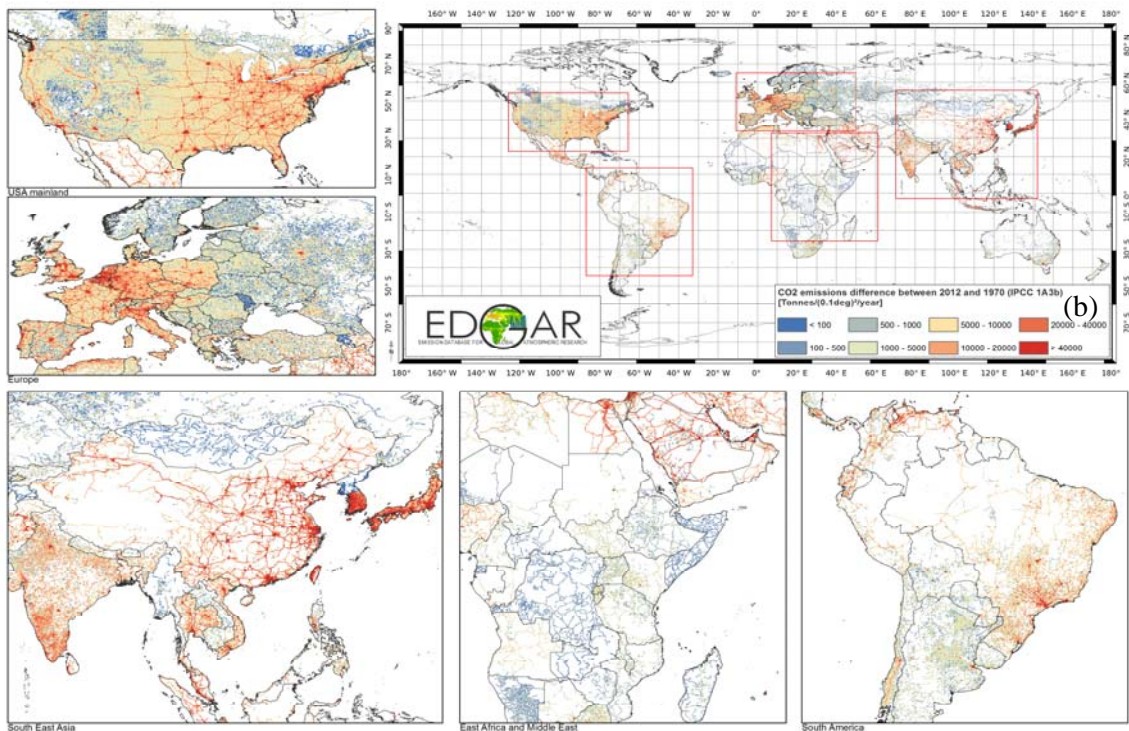

**Figure 11b: Difference in CO₂ emissions from transport between 2012 and 1970. The represented CO₂ emissions include also those from short-cycle carbon. The figures for the long-cycle and short-cycle carbon separately are taken up in the Supplementary, S5.**

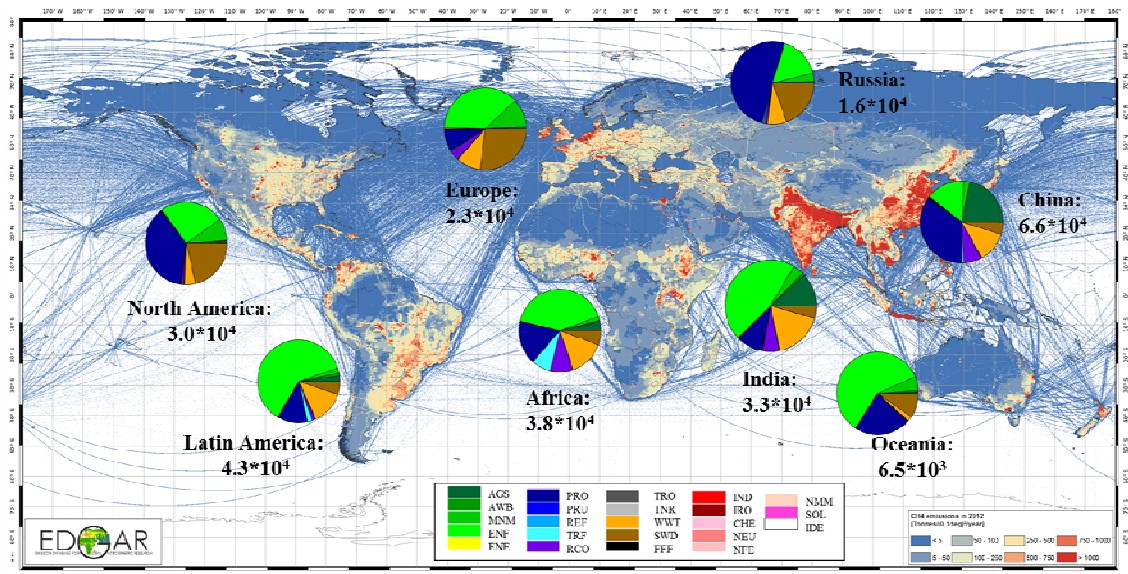

**Figure 12: CH₄ emission grid-map and relative contribution of EDGAR sectors in world regions (pie charts) for 2012.**
The legend for the PIE charts relate to the EDGAR sectors, defined in Table S3: AGS= agricultural soils, AWB=agricultural waste burning, MNM=manure management, ENF=enteric fermentation, ENE=power industry, PRO=fuel production, PRU=production& use of products, REF=oil refineries, TRF=transformation industry, RCO=residential, TRO=road transport, TNR=non-road transport, WWT=waste water, SWD=solid waste disposal, FFF=fossil fuel fires, IND=manufacturing industry, IRO=iron & steel, CHE=chemicals, NEU=non-energy use, NFE=non-ferrous metals, NMM=non-metallic minerals, SOL=solvents, IDE=indirect emissions.

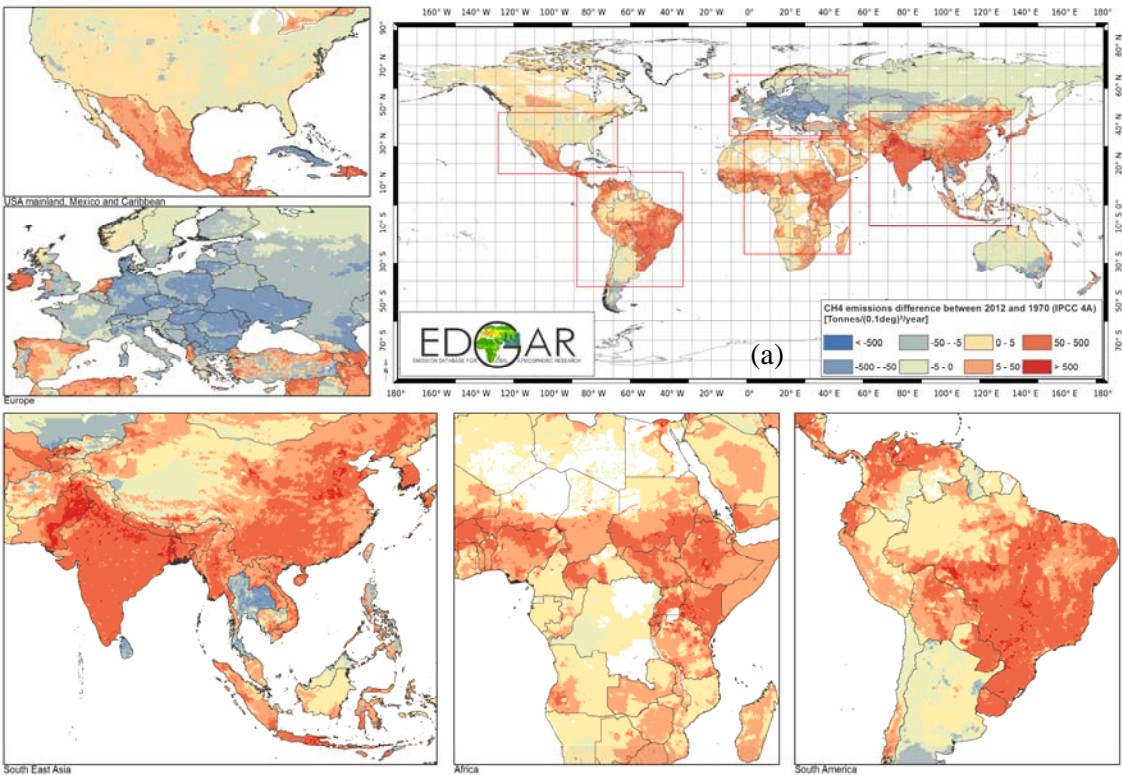

**Figure 13a: Difference in CH₄ emissions from enteric fermentation between 2012 and 1970.**

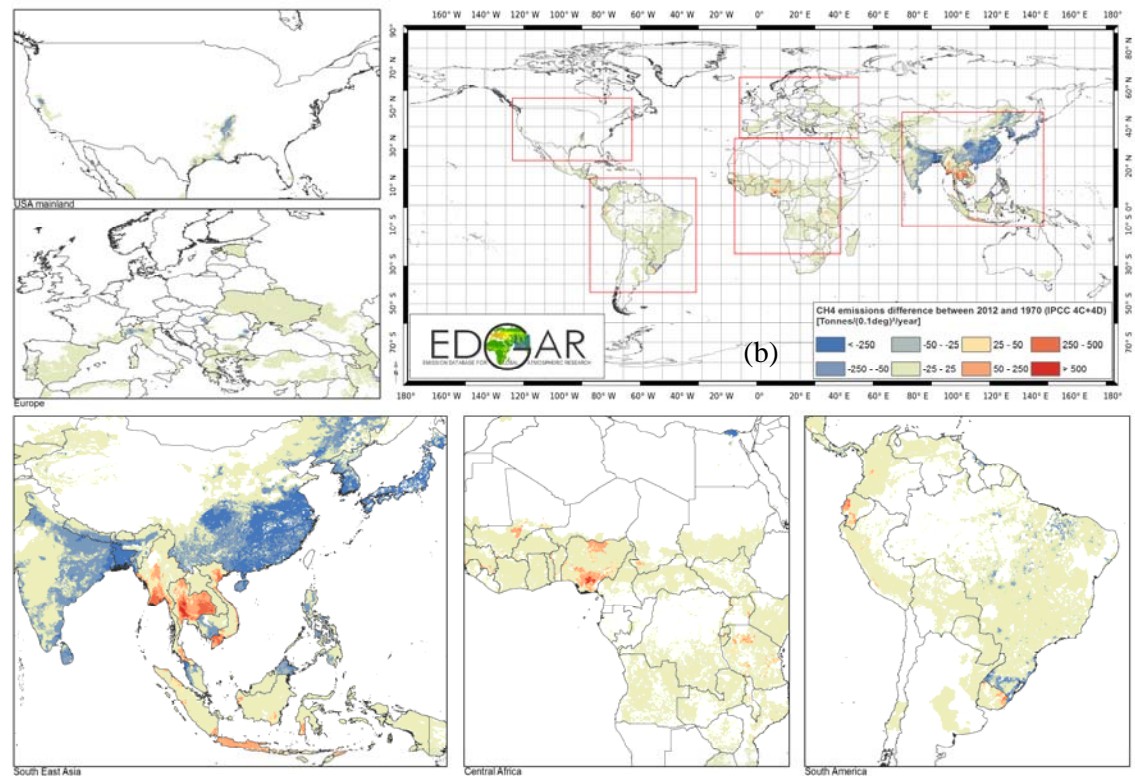

Figure 13b: Difference in CH$_4$ emissions from rice cultivation between 2012 and 1970.

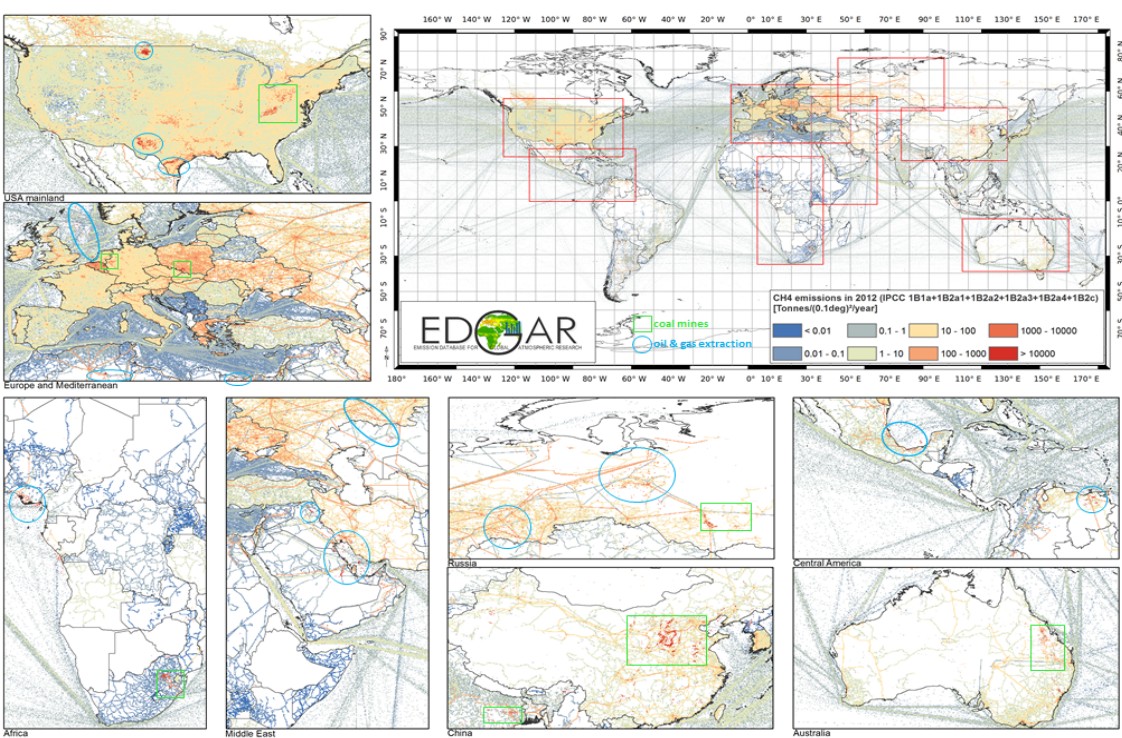

5    Figure 14: CH$_4$ emissions from fossil fuel production in 2012 with zoom on areas with intense coal mining (within green frame) and gas&oil production activities with venting (within blue circle). The shipping lines are representing the CH$_4$ leakage during transmission of oil tanker transport as fugitive emissions from the fuel and not as combustion emissions from the tanker.

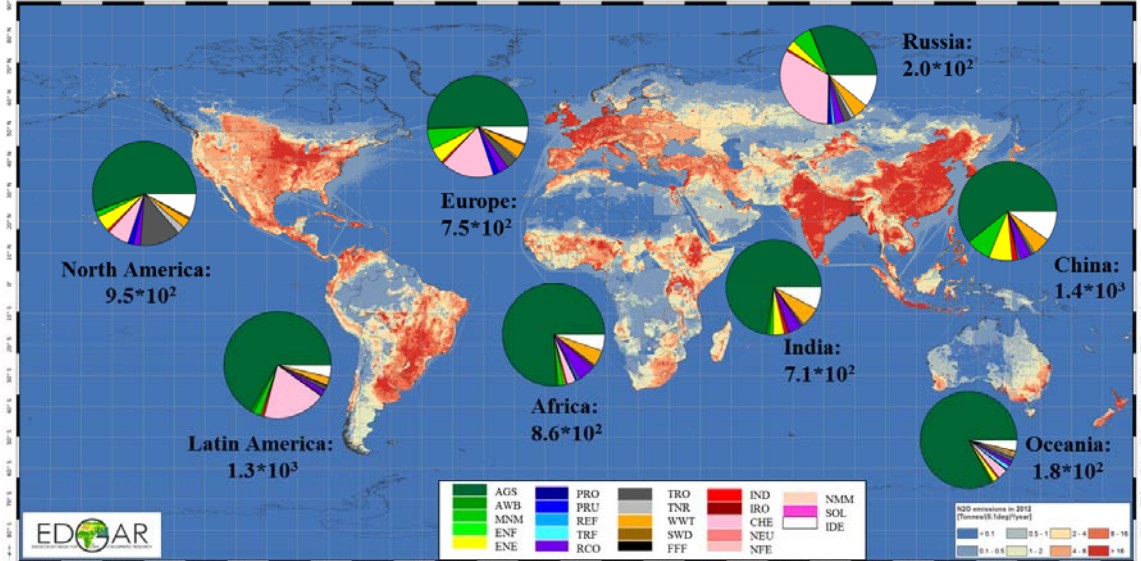

**Figure 15: N₂O emission grid-map and relative contribution of EDGAR sectors in world regions (pie charts) for 2012.**
The legend for the PIE charts relate to the EDGAR sectors, defined in Table S3: AGS= agricultural soils, AWB=agricultural waste burning, MNM=manure management, ENF=enteric fermentation, ENE=power industry, PRO=fuel production, PRU=production& use of products, REF=oil refineries, TRF=transformation industry, RCO=residential, TRO=road transport, TNR=non-road transport, WWT=waste water, SWD=solid waste disposal, FFF=fossil fuel fires, IND=manufacturing industry, IRO=iron & steel, CHE=chemicals, NEU=non-energy use, NFE=non-ferrous metals, NMM=non-metallic minerals, SOL=solvents, IDE=indirect emissions.

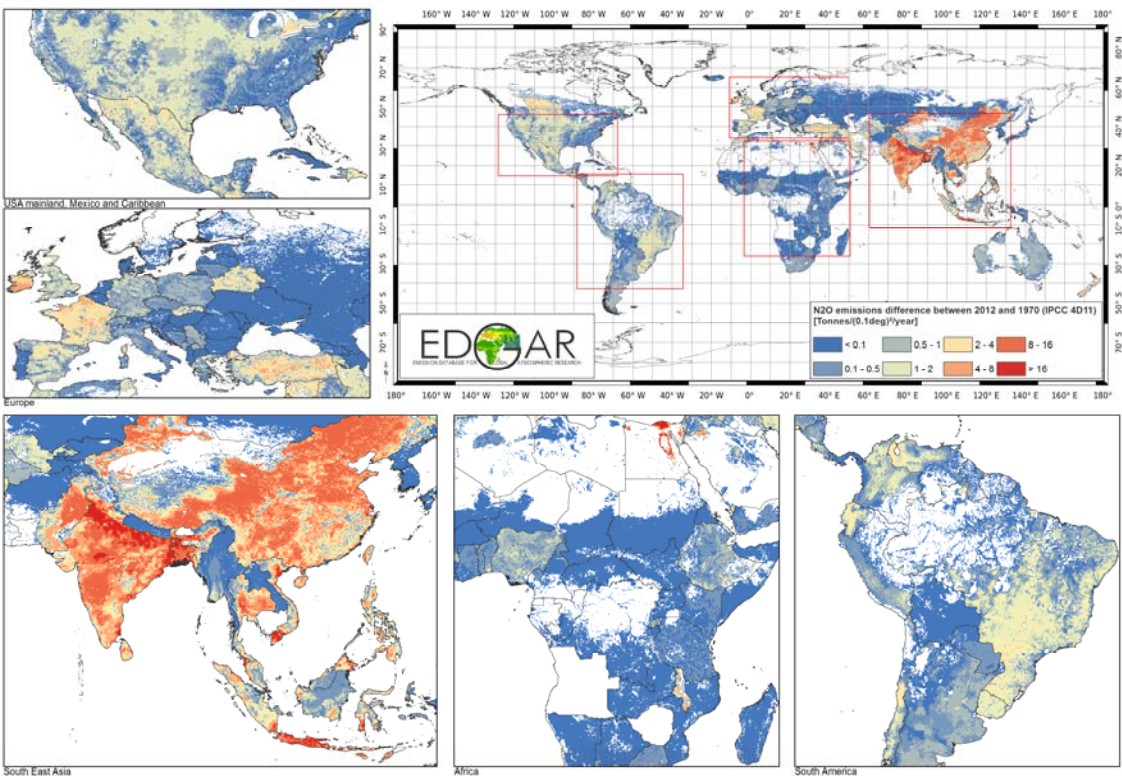

**Figure 16: Difference between 2012 and 1970 in N₂O emissions from fertiliser use on agricultural soils.**