# Peer review of "EDGAR v4.3.2 Global Atlas of the three major Greenhouse Gas Emissions for the period 1970-2012."

_Earth System Science Data, 2018_

## Referee Comment (RC1) · Carlson (Referee) · 31 Jan 2019

Thank the authors for substantial and extensive changes in the present version compared to the first submission. The description now seems more accurate, less promotional, more carefully documented, with fewer loose ends, and overall more useful. The 'strengths and weaknesses' section proves very helpful as does the evaluation and limitations section. Because these sections show the authors' approach and knowledge of the subject, they substantially strengthen the overall presentation.

The authors point out in several places their inability to evaluate uncertainty of source information. Eventually, perhaps through the CHE or VERIFY projects as described, an

integrated uncertainty assessment will prove necessary. For now, in this description, these authors have provided a much clearer guide to and recognition of residual and on-going uncertainty issues.

A few small technical points:

Page 3: text and footnote confusion and omissions at the bottom of page 3 continuing into Page 4. Please revise. Similar problem at the bottom of Page 4.

Page 7 line 29 and following: "We do not recommend an uncertainty analysis of the proxy data itself, but a sensitivity assessment of the representativeness of the selected proxy data using. This needs atmospheric transport modeling and is taken up in the CHE project." Some text missing here? Reader first encounters the CHE acronym here, needs more information about the CHE project, not defined and explained until Page 8.

Also places in the supplement, e.g. Page 15 line 21, where, because supplement now contains text cut from the main narrative and pasted here, the supplement refers to itself?

Authors will need to provide multiple eyes and careful attention to acronyms, references, footnotes, etc. during typesetting and proofreading stages.

---

## Author Comment (AC1) · 18 Mar 2019

The authors thank Prof. Carlson for the positive and constructive review and confirm that the uncertainty analysis is progressing under the H2020 projects CHE and VER-IFY.

Concerning the small technical points: Page 3 and 4: Unfortunately the footnote 5 has been spread over two pages. This was not the case in the original pdf-file that was submitted, but escaped the attention of the authors when a new pdf was created. The authors will pay extra attention to this for the typesetting of the text in the ESSD journal.

[Figure]

Page 7: As indicated by the reviewer, the sentence needs to be corrected and the authors propose the following text: "Before undertaking an uncertainty analysis of the proxy data itself, we do recommend a sensitivity assessment of the representativeness of the selected proxy data using atmospheric transport model simulations. This is initiated in further work, as stipulated in the next section 2.5."

Supplement: Indeed page 15, line 21 refers to Fig. S1 in the Supplementary, which is superfluous. The authors propose instead to change the reference in the brackets by "(see Fig. S1)". The authors did not find any other self-reference in the Supplementary, but will pay extra attention to acronyms, references and footnotes for the entire manuscript. All co-authors will be solicited to help carefully proofreading the article.

Please also note the supplement to this comment:
https://www.earth-syst-sci-data-discuss.net/essd-2018-164/essd-2018-164-AC1-supplement.pdf

―――――――――――――――――――――――

**Supplement:**

**Comments on: "EDGAR v4.3.2 Global Atlas of the three major Greenhouse Gas Emissions for the period 1970-2012" (essd-2018-164) by G. Janssens-Maenhout et al.**

REPLY TO THE REVIEW OF JANUARY 2019 BY PROF. D.J. CARLSON

The authors thank Prof. Carlson for the positive and constructive review and confirm that the uncertainty analysis is progressing under the H2020 projects CHE and VERIFY.

Concerning the small technical points:

*Page 3 and 4:* Unfortunately the footnote 5 has been spread over two pages. This was not the case in the original pdf-file that was submitted, but escaped the attention of the authors when a new pdf was created. The authors will pay extra attention to this for the typesetting of the text in the ESSD journal.

*Page 7:* As indicated by the reviewer, the sentence needs to be corrected and the authors propose the following text: "Before undertaking an uncertainty analysis of the proxy data itself, we do recommend a sensitivity assessment of the representativeness of the selected proxy data using atmospheric transport model simulations. This is initiated in further work, as stipulated in the next section 2.5."

*Supplement:* Indeed page 15, line 21 refers to Fig. S1 in the Supplementary, which is superfluous. The authors propose instead to change the reference in the brackets by "(see Fig. S1)". The authors did not find any other self-reference in the Supplementary, but will pay extra attention to acronyms, references and footnotes for the entire manuscript. All co-authors will be solicited to help carefully proofreading the article.

---

## Referee Comment (RC2) · Tomohiro Oda (Referee) · 28 Mar 2019

Dear authors of the manuscript,

  This manuscript describes the updated version (v4.3.2) of the EDGAR emission dataset with a focus on $CO_2$, $CH_4$ and $N_2O$.  The importance of the EDGAR dataset for the science community is no need for discussion.  The EDGAR dataset is a unique historical gridded emissions dataset that has been extensively used in the atmospheric chemistry, carbon cycle, climate, and many other relevant atmospheric research communities.  Over the years, the EDGAR team has developed and maintained the global framework that offers spatially-explicit country level sectoral emissions prepared in a systematic and consistent manner across different compounds.  As the authors claimed, the EDGAR dataset should help connecting science and policy and can play a critical role in future emission verification support systems which are currently planned and studied by the research community.

  I would like to note that this manuscript has been significantly improved from the previous version of the EDGAR manuscript as published as Janssens-Maenhout et al. (2017) in ESSD Discussion.  While Janssens-Maenhout et al. (2017) did a great job to put the considerable amount of information regarding the data used in the EDGAR dataset development together into a manuscript, I personally felt it had some weakness as a scientific contribution to ESSD.  My main concern was the level of the information provided in Janssens-Maenhout et al. (2017) was not enough to maintain the traceability of the study and promote productive scientific discussions through the use of EDGAR dataset.  This manuscript seems to be much closer to what ESSD and its audience would like to read (which Dr. Carlson and myself tried to clarify in the editorial comment published as Carlson and Oda (2018)).  I found the authors particularly improved the presentation in the Supplementary Information.  The text newly added was helpful to understand the emission calculation and modeling processes better.

  In this review, before I recommend this manuscript for publication, I (as a referee as well as one of many EDGAR emission dataset users) would like to discuss several things that I believe the authors can further address in order to improve this manuscript as a scientific contribution to ESSD.  The following part will discuss more in detail.  I am looking forward to receiving your response.  I hope my review comments will be useful.

Sincerely,
Tomohiro Oda (tomohiro.oda@nasa.gov)

1. Detailed comments/suggestions/discussions

Data tables
  First, I would like to propose to improve the data tables.  In many places (in the tables and also in the main text), data source references (where you get the data) and journal references (documents that explain the data) are mixed up.  Those two can be summarized in tables in a more systematic way (see Table 4. https://www.earth-syst-sci-data.net/8/605/2016/essd-8-605-2016.pdf as an example).  I also found that data tables do not indicate the edition of the data used.  As we are all aware, the use of different editions of data can be a major source of differences (or error and uncertainty) in resulting

emission estimates.  What I request here is important for traceability and let the data users to do a fair interpretation of the EDGAR dataset.

What's new and/or different from the previous and existing EDGAR datasets?

To improve the readability of this manuscript, I would like to propose to highlight the differences from previous version of the EDGAR (v4.2).  This is an updated version of the EDGAR dataset that was constructed in a very similar manner as other pollutant estimates.  I would imagine many readers of this manuscript also used previous and/or existing other versions of the EDGAR dataset.  Given that, it would be very helpful for the readers (especially existing data users) to highlight more the differences from the dataset they have been familiar with.  In this way, this manuscript should be able to help educating the readers efficiently.  I found Section 4 in the Supplementary Information (especially, Fig. S3) very useful.  I believe the authors should be able to improve other parts of the manuscript in the same way.

Evaluation of gridded maps

Similar comments go to the resulting gridded map section.  Because of the nature of emission spatial modeling, it is challenging to present a meaningful uncertainty analysis for gridded estimates.  As pointed out by the authors, I also agree that the gridded uncertainty estimates seen in the recent literature published are often large and hard to interpret.  Also, it is questionable to me if those uncertainty estimates would inform transport modeling and inverse modeling studies in a meaningful way.  But I do see the benefit of those analysis as an evaluation of gridded emission estimates as we could at least inform the data users on the differences we should expect in relative to other emission datasets (or previous version of them).  Especially in this study, showing the differences from the previous version of EDGAR (as done for emission estimates in the Fig. S3) should summarize and highlight the changes the authors have made to the dataset.  Since the changes we would see are combinations of the changes in emission estimates and spatial modeling, they might not be easy to interpret.  However, we would like to see the impact of the changed the authors made (supposedly this is the improvement from the previous version).  I would be curious to see the magnitude of the emissions differences in comparison to Figure 10a for example.  Such comparison would loosely tell us the sensitivity of the urban emission estimates to the modeling approach.  The comparison to other studies have done by previous studies (e.g. Maasakker et al. 2016; Gately and Hutyra, 2017; Oda et al. 2018).  The authors could simply use those differences to discuss the resulting gridded maps.

Hot spot analysis

I am a little bit confused by this analysis.  The EDGAR gridded emissions are based on spatial disaggregation of country-level sectoral emissions.  Thus, the emissions at grid level are obtained by scaling the proxy data with the sectoral total emissions (as shown in formula (2)).  It is clear that the urban emissions are not mechanistically modeling, unlike Gurney et al. (2018).  Thus, the spatial patterns and changes in emissions are only explained by changes of the proxy data and the sectoral totals.  The urban hot spot is a result of emission modeling and theoretically do not show the local emission drivers.  I would imagine EDGAR does a  better job than the carbon cycle FF dataset like CDIAC or ODIAC, but still it is disaggregated emissions.  Also, the spatial resolution is pretty coarse when looked

at urban CO2 simulations studies.  I felt the authors should have touched caveats the hot spot analysis has.

   Another thing I would be curious to hear from the authors is the readiness level of the gridded emissions for policy application (i.e. the Paris Agreement, PA).  In my opinion, many of the gridded emission estimates including this study (and also mine) still need a lot of development and improvement to support the PA in a scientific way we wish.  As mentioned earlier, many of the gridded dataset do not have a capability of showing subnational emission reductions in their emission field (no local drivers).

Be more quantitative

   At least we emission data developers should do is to describe the emission modeling procedure and the data used in details.  As I acknowledged earlier, the authors have greatly improved the presentation especially in the Supplementary Information.  However, I would like to request the authors to be more quantitative.  For example, there must have been a QA/QC process in the use of the CARMA power plant database.  Careful data users already knew the power plant database has been used in the EDGAR dataset development as it has been described in the EDGAR website.  What we the audience of this manuscript would like to learn from this manuscript is how differently the power plant database was used compared to other studies and the differences we expect to see.  The authors might be able to tell certain thresholds for selecting the power plant entries in the CARMA.  For example, as I don't believe the total emissions from CARMA and the EDGAR total sectoral emissions matches well (maybe they do), I believe there were some ad-hoc adjustments to CARMA (or EDGAR) in mapping power plant emissions.  Such details should allow the data users to decide if the EDGAR dataset is a right choice to answer their research questions.

2. Line by line comments

P1, L20:  "…disaggregated to IPCC-relevant source-sector levels".  This does not seem to sound correct.  Emissions are calculated sectoral basis and then spatially disaggregated.

P1, L25: fully traceable.  I feel the authors needs a little bit of extra effort to support this.

P1, L25: IPCC-based methodology.   This is confusing.  IPCC (1996) is a guideline.  They don't precisely define the emission calculation.  Maybe IPCC-compliant would be more appropriate?

P1, L26-: Maybe the use of the term "short_cycle" would be helpful for the EDGAR users?

P3, L29: Policy impacts.  I agree at a large spatial scale (e.g. country and regions, w sectors), but do you think the disaggregated gridded emissions can be used to assess the policy impact?  To support this, you'd have to show that your disaggregated emissions are reasonably indicating the local change.  Gridded emissions in EDGAR are disaggregated rather than mechanistically calculated.   If we see a X% emission change over a city A, that changes are because of the change in the total sectoral emissions and/or changes in the proxy.  I am curious to hear how the gridded emissions can be used for informing

policy. As a developer of disaggregated gridded emissions, I feel I need to acknowledge the limitation of the use of disaggregated emissions (see Oda et al. 2019).

P3, L35. Footnote 5. In my opinion, this is worth mentioning in the main text. For example, the errors in the emission ratio analysis could be dominated by the error in the air pollutants side because of this.

P4, L5: and emission disaggregation?

P5, L6: How did you deal with countries like USSR and Germany over the EDGAR period (e.g. activity data, emission estimates and spatial proxy)? Which country identifier data did the authors use?

P5, L14: So the emission seasonality is not country specific in EDGAR. Is that correct?

P5, L15: CARMA indicates only four years with their two versions of the database. Did you use the emission estimates as they are? If so, how did you match with the CARMA country total CO2 emissions with the EDGAR sectoral total? How did you use those information for intervening years? What was the quality control?

P6, L2: Just to be clear, do the authors recommend those three BB products to supplement the EDGAR emissions because of some reasons and/or compatibility?

L6, L31: I assume the authors meant to say EDGAR wants to avoid model-based estimates, but try to stick to the data based (or EDGAR way?) emissions.

P7, L5: Table S4a only provides the references for data sources. There is no narratives. So it is unclear that how the temporal profiles were constructed and what they are representing.

P7, L11: Huang et al. (2018). I could not find this reference.

P7, L11: I don't understand this. The temporal profiles presented in Andres et al. (2011) were based on a different approach than this study.

P7, L15: Not just global models, the data users are using the EDGAR dataset for regional and even local models, too.

P7, L18: Linear -> Line (?)

P7, L21: Does the area average change over the time (1970-2006, for example)? Probably no?

P7, L25: Where it is *reported*. Probably it would be better to say like "where it is likely located" as the EDGAR is based on the emission disaggregation. Point source emissions could be allocated to the geographical locations reported by databases such as CARMA. But there is no linkage between the

emission at the country, sectoral level and the power plant locations.  For line and area sources, the locations are estimated (or modeled) rather than specified.

P7, L27: Just to make sure… Oda et al. (2018) also uses CARMA for point sources.

P7, L28: I think so, too!  I'd suggest to add "sectoral" before "emissions".  Those two emissions are both based on CDIAC country totals that are calculated based on fuel use.  Thus, the emission disaggregation problem in CDIAC and ODIAC is fundamentally different than that of EDGAR.

P7, P30: I agree with the authors that the uncertainty analysis for the proxy data themselves do not provide what we want as an emission uncertainty that are useful for inverse modeling and/or data assimilation.  But I do think that such sensitivity test can be done w/o transport models as demonstrated by previous studies.  Also, I imagine what the uncertainty analysis that the authors have in their minds will be highly depending on the models and observations used/assumed.  Aside from the uncertainty analysis, an important missing component in this manuscript is an evaluation of gridded maps that should demonstrate the improvement and/or differences/changes in emission spatial representations from existing emission datasets and previous EDGAR datasets.  In my opinion, such evaluation should be done if this manuscript is going to be a scientific paper, rather than a tech report.

P7, P31: The authors should introduce the CHE project.

P8, L5: To me, this sentence does not seem to fit here as the author returned to the national level uncertainty analysis in the next sentence.

P8, L11: I am confused.  The eq. (4) includes all three gases, while Table 2 (not Table 3, 4, and 5)  shows uncertainty estimates for three gases.

P9, L3: So those two uncertainty estimates are not compatible.

P9, L12: Note Andres et al. (2016) limited the result by saying CASE FOR CDIAC.

P9, L17: I feel probably this needs to be elaborated a little bit.  Firs of all, it is unclear that what the authors meant by saying the complete uncertainty and how it is achieved.  Again, given the nature of the gridded maps and the lack of the evaluation data, it Is difficult to validate the gridded maps (e.g. Andres et al. 2016; Oda et al. 2018).  It is important to combine top-down and bottom approaches and thus good error estimations are necessary.  But as mentioned earlier, what ESSD and the audience of ESSD expect this manuscript to present is not necessarily the covariance matrix for data assimilation, but a reasonable sense of how much we could trust the gridded maps.

P9, L28: TD for scientists.  Maybe TD was just for scientists, but I believe it is required for policy making as we want to assure the accuracy of bottom-up emission estimates.  Emission inventories are subject to

systematic biases.  That is a huge problem under the Paris Agreement as the emission estimates should be consistent with what we emitted to the atmosphere (Oda et al. 2019).

P9, L35: Studies listed in Table S5 are great examples of how atmospheric modeling can inform us about the errors and biases in emission estimates.  But I would like to point out that none of them seems to provide the "complete uncertainty".  None of them approached to the grid level uncertainty (excepting the power plants), either.  This sentence reads the use of atmospheric modeling is a perfect solution, but the authors should acknowledge that there are limitations, too.  The sensitivity of the emission spatial representations should depend on the spatial resolution of the models and observation systems assumed on the top of the errors in the transport modeling (meteorology, chemistry, etc.).

P10, L1: I would imagine $CO_2$ is a very different case compared to $NO_x$, $SO_2$, and $CH_4$.

P10, L21: Elvidge et al. (2009) is based on the DMSP lights, but what the download link provides is a different data product.  Elvidge et al. (2009) does not have time series if I remember correctly.  Also, Elvidge et al. (2009) does not cover all the countries with gas flare emissions.  How did you manage the gas flare emissions w/o nightlight location estimates?  Need to explain.

P12, L14: Verify.  Maybe it would be better to say like "detecting the biases in emission inventories"?

P15, L7-:  As mentioned earlier, the authors should provide an evaluation of the gridded maps.  This is the results of the multiplication of the total sectoral emissions and normalized proxy data.  From previous studies, we expect these gridded maps should show different emission patterns from other studies.  For example, Gately and Hutyra (2017) has shown the large discrepancy between their spatially-explicit urban emission estimates and EDGAR.  I do not request the authors to validate these gridded maps, but help the audience of the ESSD to interpret these gridded maps in order not to over-interpret these maps beyond their limitations.  It could be done by acknowledging the emission spatial differences reported by previous studies.

P16, L2: What was the QA/QC done for power plant data especially for geolocation?  Are they verified?  What was the threshold for emission intensity?

P18, L2: I (I am sure Dr. Carlson, too) would like this manuscript to be one of the good examples of ESSD articles!

P18, P22: This is not about EDGAR, but atmospheric models.  I feel this is misplaced here (maybe I might be wrong…).  I would add this is what we expect to atmospheric inversions to do, but it is not happening in the way we hope quite yet.

P19, L7: Do the authors limit to the interpretation of data from satellites, but not other platforms?  The EDGAR team has experienced several success with satellite measurements for other compounds.  But

data form other platforms such as ground-based sites (e.g. radiocarbon measurements) and aircrafts should not be excluded (I assume the authors did not mean to).

P19, L11: Regional specificity, too? This had to be sacrificed in order to achieve a global systematic framework. I think this is consistent with the first bullet point at L15.

P19, L17: top-down -> disaggregation or downscaling? (just to avoid a potential confusion with atmospheric top-down)

P19, P20: Would you want to add a caution that gap-fill could sacrifice the consistency in emissions that EDGAR maintained? For CO2, for example, we don't want to do a mosaic emission approach as done for air pollutants as the consistency between the global and regional budget is important.

P19, L31: I assume this is a general statement for the EDGAR dataset as differences due to the technology types do not make significant differences according to the authors.

P19, L36: Given the fact that the authors highlight the importance of the point source information, I feel the description of the point source data and modeling need to be improved a bit (see relevant comments listed earlier).

P20, L9: I am confused here. What the authors described here does make sense, but this would turn EDGAR into what we can't define as an inventory (more like a model). Is that the direction of the EDGAR development?

P20, L31: Similar comment to the above. The approach is valid, but what about the large uncertainty associated with the CO2 emission estimates? This is CO2, not SO2. Would you be happy to include those estimates as a part of EDGAR dataset? This does not seem to be very consistent with the EDGAR's basic principle mentioned earlier.

P21, L1-: Data availability section needs to be improved.

P40, Table 3. Note the latest version of the ODIAC data product (ODIAC2018) has been available. I suspect that the authors might be comparing the CO2 estimates based on the different year edition of the statistical data which do not allow us to do a fair comparison.

P48, Figure 10a and 10b: The quality of the figure needs to be improved. The numbers of the main panel are hard to read even enlarged.

P49: Figure 11a and 11b: I'd suggest to do the v4.3.2 minus v4. It is not going to be a clean comparison. But it should demonstrate the improvement and/or changes over different versions of EDGAR (which is the main claim of this manuscript). The same comment goes to CH4 and N2O maps, too.

Supplement Information

I could not find the information about technology specific proxy data. Are those only used for air pollutants?

P2, L2: How did the authors deal with countries such as Germany and Former Soviet Union in the EDGAR calculation and mapping?

P14, P4: Did the authors linearly interpolate the population in time?

P14, L13: How the geolocation errors were managed in the QA/QC process? Did the authors consider commission/decommission of plants?

P15, L4: NOAA gas flaring nightlight data used in Oda et al. (2018) only convers 60+ countries and thus the emissions needed to be distributed as a part of area sources in ODIAC. How was it done in EDGAR?

P15, L12: Are this weighting factors listed somewhere? Did you use population to get an additional spatial weighting function?

P15, L24: km 101 until the last 101 km?

P15, L32: Is Friedrich and Reis (2004) accurate reference for this? Friedrich and Reis (2004) was a compilation of air pollutants studies. So I assume the authors used a temporal profile of an air pollutant(s) as a proxy. I also could not find residential emission temporal profile plot in Friedrich and Reis (2004).

P15, L33: I think the authors should at least explain how the temporal profiles were constructed. Are they averages for multiple year data (if so, error bar?) or single year? Where did the underlying data come from (only from Europe)?

P21: L2: CARMA base years should be 2000 & 2006 (earlier version) or 2004 & 2009 (for v3.0). No?

P21, L12: This is a bit stretch… the authors should carefully read Nassar et al. (2017). I am curious to ask the EDGAR team is willing to incorporate the satellite-based power plant estimates in the emission dataset. It seems to be ok for SO2 (large errors), but I am not sure about CO2.

P21, Figure S2. Is it long term averages of European sectoral temporal profiles? What do we see here?

P22, Fig S2b: The authors need to describe how each one of them was constructed and explain the differences in terms of modeling approach and data used.

P22, L6: We need these for gridded maps.

P23, L15: Were the emissions CARMA indicated use as they were?  The linkage between CARMA and the EDGAR sectoral total emission as the two should not match perfectly w/o any adjustment.

References:

- Andres R. J., Gregg J. S., Losey, L., Marland, G., and Boden, T. A.: Monthly, global emissions of carbon dioxide from fossil fuel consumption, Tellus B, 63, 309-0327, https://doi.org/10.1111/j.1600-0889.2011.00530.x, 2011.
- Gately, C. K., & Hutyra, L. R. (2017). Large uncertainties in urban-scale carbon emissions. Journal of Geophysical Research: Atmospheres, 122, 11,242–11,260. https://doi.org/10.1002/2017JD027359
- Janssens-Maenhout, G., Crippa, M., Guizzardi, D., Muntean, M., Schaaf, E., Dentener, F., Bergamaschi, P., Pagliari, V., Olivier, J. G. J., Peters, J. A. H. W., van Aardenne, J. A., Monni, S., Doering, U., and Petrescu, A. M. R.: EDGAR v4.3.2 Global Atlas of the three major Greenhouse Gas Emissions for the period 1970–2012, Earth Syst. Sci. Data Discuss., https://doi.org/10.5194/essd-2017-79, 2017.
- Maasakkers et al. Gridded National Inventory of U.S. Methane Emissions, *Environ. Sci. Technol.*, 2016, *50* (23), pp 13123–13133, DOI: 10.1021/acs.est.6b02878
- Oda, T. and Maksyutov, S.: A very high-resolution (1 km×1 km) global fossil fuel $CO_2$ emission inventory derived using a point source database and satellite observations of nighttime lights, Atmos. Chem. Phys., 11, 543-556, https://doi.org/10.5194/acp-11-543-2011, 2011.
- Oda, T., Maksyutov, S., and Andres, R. J.: The Open-source Data Inventory for Anthropogenic $CO_2$, version 2016 (ODIAC2016): a global monthly fossil fuel $CO_2$ gridded emissions data product for tracer transport simulations and surface flux inversions, Earth Syst. Sci. Data, 10, 87-107, https://doi.org/10.5194/essd-10-87-2018, 2018.
- Oda, T. et al.: Errors and uncertainties in a gridded carbon dioxide emissions inventory, accepted for *Mitigation and Adaptation Strategies for Global Change*

---

## Referee Comment (RC3) · Robbie Andrew (Referee) · 1 Apr 2019

I have only a few comments on this article for purposes of clarification, and focus on the methodology section.

Page 4, lines 19-25: This is a very long opening sentence. I would suggest preceding it with something simpler like "Annual national emissions are calculated using national activity data and emission factors, modified by a number of additional parameters." Page 4, line 29: what does "uncontrolled" mean in this context? Do you mean unverified or default? Is this important clarification or could it simply be dropped here? Page 5,lines l7-8: The sentences beginning "The yearly accounting..." needs to be rewritten.

[Figure]

It took me several goes to understand what was meant here. Either rewrite for clarity, or consider removing/relocating: it's not clear why this point is necessary here. Annual data is clearly useful, without resorting to justification by how it might be superior to sub-annual data. Page 5, line 14: should this be "latitudinal band"? Page 5, line 18: I suggest striking "-related" and retain simply "energy sector". While Table 1b is clearly introduced, Table 1a is not. A simple "Table 1a provides..." on page 5, about line 18, would help enormously here, particularly since it's not immediately clear whether the table 1a referred to is in this paper or the IPCC guidelines. Page 6, line 35: I'm confused as to why the 1996 codes should be used when everywhere else the 2006 guidelines are referred to. page 8, line 6: The project is called "CO2 Human Emissions"; remove "of". page 8, line 13: please rewrite "countries adhering to 24OECD90 countries" as, for example, "the 24 member countries of the OECD in 1990" or "the 24 OECD90 countries", similarly 16EIT90, etc. Page 8, line 24: I had a quick look at the cited article and could find no reference to CO2. Did the authors mean carbon monoxide emissions from biofuel combustion? If the cited article doesn't specifically say that CO2 emissions from biofuel are difficult to estimate, perhaps this should be reworded. Perhaps, for example, the meaning is that reliable biofuel consumption data are difficult to obtain (indicated on page 6506 of the reference), and the authors here make the conclusion that it is difficult to estimate reliable CO2 emissions from biofuel combustion?

I wonder why Elvidge et al 2009 is used for flaring when that team has produced several updates since then. Do the subsequent releases resolve the apparent difference between night-light derived flaring and national inventory reports/CDIAC?

Comparison is made with IEA's emissions estimates, and the comment is made that that differences are largely because of the different emission factors used. But this comparison is with an old (2014) edition of the IEA dataset, and IEA have subsequently switched to using the 2006 guidelines' emission factors. Some comment must be made to this effect such that the reader is not misled. Given the same underlying energy data

and same emission factors, one would expect for the bulk of $CO_2$ emissions categories that the emissions in EDGAR would be identical to those in IEA.

I congratulate the authors on the breadth of this work and the article itself. I must however register a minor complaint in response to the statement made at the top of page 18. While this paper is very welcome documentation for a very important global dataset, it cannot be said to be replicable given the descriptions herein, and therefore falls some way short of being "full, transparent and inclusive documentation." I do hope that this article will be updated in future following the release of a new version of the dataset, and that attention will be given to improving the transparency of the methodology used.

---

## Author Comment (AC2) · 25 Apr 2019

Dear Prof. Oda,

The authors thank Prof. Oda for the positive and constructive review and have put extra efforts for providing full transparent and inclusive documentation of the EDGARv4.3.2 dataset, which we hope supports the future GHG monitoring and verification capacity.

1. Detailed comments/suggestions/discussions Data tables The table 1b has been further improved by splitting the columns for the activity data and emission factor data into four columns in total: the AD data source (carefully mentioning the edition/ version of

the dataset used); the AD data reference; the EF data source; the EF data reference, where the second column addresses the reference. Tables S4a and S4b of the supplementary have also been improved in a similar way with the temporal data source respectively gridmap data source split from the data reference.

Differences with previous EDGAR datasets The authors agree to inform the readers of the difference between the different EDGARv4 datasets. Therefore, section 4 of the Supplementary Information addresses the differences between EDGARv4.3.2 and previous versions v4.2 and v4.1, which have not been documented in a publication but which have been used by atmospheric modellers. The differences shown in Figures S3 are explained by the continuous improvement the EDGARv4 database has gone through since its first release. Such improvements are detailed in the revised manuscript via an explicit reference in the main manuscript at the end of the section 1 "Historical evolution". We wish to stress that the EDGARv4.3.2 is the result of a steady improvement of the EDGARv4 database over more than a decade, also thanks to the feedback of users. In particular we note that: - For the main differences between EDGARv4.2 and v4.1 we refer to http://edgar.jrc.ec.europa.eu/Main_differences_between_EDGARv42_and_v41.pdf. - For the main differences between EDGARv4.3.2 and v4.2 we refer to the Supplementary of the paper, section 3 and Table S5 with the findings of studies, using EDGARv4 as input. For Table S5 we refer to the Supplement here.

Evaluation of gridded maps The authors agree that the spatial distribution is the major cause of the differences in the gridmaps and we therefore propose to include an overview of the improvement in the gridding with the table below (which is included in the revised manuscript, Table S5). Section 4 of the Supplementary refers to the findings of Gately & Hutyra. (2017) and is expanded with the findings of Maasakkers et al. (2016) and Oda et al. (2018) as follows: "Improvement of the spatial distribution of the fossil fuel production emissions in EDGARv4.2 was shown to be necessary for USA by Maasakkers et al. (2016) and for China by Saunois et al. (2017) and addressed

accordingly by extending the dataset with extra point sources for the extraction and mining sites. The importance of point and line source data has been also illustrated by Oda et al. (2018) but needs further observation-based verification." As proof of the improvement of the road transport spatial distribution proxy, we show in the figure here below the map of the NOx emissions due to road transport making use of traffic volume data for Europe (right) and the EDGARv4.3.2 road transport (left) gridmaps. We do refrain from deriving insights from cell-to-cell differences or ratios between gridmaps, as we experienced that these are not revealing of useful information on where to improve (due to displacements and skewness, for example). We observe that the spatial changes are in the expected direction, with the same patterns in most EU countries (e.g. UK, Germany, Poland) but also differences (e.g. in Italy, where the road transport network between cities needs to be more pronounced). In our view, the magnitude of the improvement can only be assessed and quantified by confronting to the observed data that we want to represent with the spatial distribution.

Hot spot analysis Indeed EDGARv4.3.2 is not mechanistically modeling urban emissions, unlike Gurney et al. (2018) but do start with sectoral country totals, to avoid the need of selecting a definition for an urban area. For CO2 emissions, which are dominated by fuel combustion, the national fuel statistics, driving the emissions, are known with a much smaller uncertainty than what is available at city level by e.g. the Covenant of Mayor data in Europe. Whereas subnational or urban emission gridmaps might be more subject to the uncertainty on the activity data that are to be defined as representative for the local area, the uncertainty in the EDGAR gridmaps is mostly determined by the assumptions on the representativeness of the selected spatial distribution proxy for the entire country. With increasing granularity of the spatial distribution per sector and using where point source data, this uncertainty reduces. EDGARv4.3.2 uses 297 distinct datasets for the different subsectors. At least for Europe, the authors believe that the hot spot analysis remains useful, in particular because of the use of the many point sources for any industrial or commercial activity. The authors would assume a better spatial representation of the emissions than what is obtained with e.g.

the CCFFDAS model or even with the ODIAC model, but the validity can only be proven by an observation-based verification with e.g. space-borne XCO2 measurements as a next step. The authors realise that this goes hand in hand with the improvement of the temporal profiles and want to refer to the recent work we submitted to ESSD by Crippa et al. (2019) .

Policy application The European Commission (EC)'s in-house global emissions database EDGAR is, since more than a decade, known and used by the EC's Directorate General Climate Action (DG CLIMA). As such, DG CLIMA has been using emission estimates for world regions/countries in preparation of the climate negotiations at the COP (e.g. presentation of Director General Jos Delbeke in 2012, Staff Working Documents in 2014 and successive years). Most recently DG CLIMA is increasingly asked to look into subnational emission inventories, such as inventories at urban or province-level scale. These can provide actionable information on the implementation of local GHG reduction measures. For the readiness level of the gridded emissions for policy application, the authors refer to the air quality. The air quality (in Europe addressed by a first directive in 1970) and transboundary air pollution (addressed under the UNECE Convention on Long-Range Transboundary Air Pollution - CLRTAP of 1979 (in force since 1983) focused in a first step on emission inventories of air pollutants and the monitoring of the time-series. In a second step, gridmaps were requested and nowadays the Parties need to provide these at 0.1deg resolution on an annual basis. The European Commission – Directorate General Environment appreciated the delivery of default emission gridmaps for the European Commission Directive on the Pollutant Release Transfer Register (E.PRTR) and supported EDGAR with extra funding for further use of the emission gridmaps by the CLRTAP Task Forces of Emission inventories & Projections – TFEIP and of Hemispheric Transport of Air Pollution – TF HTAP. Nowadays, as one of its activities the Copernicus Atmospheric Monitoring Service assesses the bottom-up gridded emissions (and in particular local exceedances of pollution levels) with top-down measurements.

Quantitative information on CARMA corrections The authors agree that the power plants are very important point sources and that the CARMAv3.0 dataset has been carefully screened with an internal QA/QC procedure to avoid large errors. We are not allowed to disclose the CARMAv3.0 dataset, because that is not our proprietary and unfortunately no longer online available. For the sake of transparency, we summarized in the table below the different steps undertaken to convert it to our EDGRv4.3.2 spatial proxy dataset for the power sector. CARMAv3.0:68931 power/heat plants Corrected CARMAv3.0:200 points have been corrected for the missing or inverted coordinates For China: 1200 points have been added manually with internal resources For Russia: 50 points have been added manually with internal resources Resulting EDGARv4.3.2 proxy for power plants aggregated to 0.1x0.1: 16931 cells, of which: 4610 cells are defined for Auto-producers 5199 cells are defined for COAL 3304 cells are defined for GAS 3818 cells are defined for OIL

2. Line by line comments Manuscript Supplementary Information We refer to the Supplement here for the replies (point by point).

We'll upload the revised manuscript and online supplementary information after having completed the review of Dr. R. Andrew as well. Thanks for your understanding and continued interest. Best regards, Greet.

Please also note the supplement to this comment:
https://www.earth-syst-sci-data-discuss.net/essd-2018-164/essd-2018-164-AC2-supplement.pdf
* * *
[Figure]

**Fig. 1.** Comparison of the road transport in Europe gridded with EDGARv4.3.2 proxy and with traffic volume

[Figure]

**Fig. 2.** Difference EDGARv4.3.2 and EDGARv4.2 emissions of CO2 for road transport (in t/yr per gridcell of 0.1deg)

**Supplement:**

**Comments on: "EDGAR v4.3.2 Global Atlas of the three major Greenhouse Gas Emissions for the period 1970-2012" (essd-2018-164) by G. Janssens-Maenhout et al.**

REPLY TO THE REVIEW OF MARCH 2019 BY PROF. T. ODA

The authors thank Prof. Oda for the positive and constructive review and have put extra efforts for providing full transparent and inclusive documentation of the EDGARv4.3.2 dataset, which we hope supports the future GHG monitoring and verification capacity.

**1. Detailed comments/suggestions/discussions**
**Data tables**
The table 1b has been further improved by splitting the columns for the activity data and emission factor data into four columns in total: the AD data source (carefully mentioning the edition/ version of the dataset used); the AD data reference; the EF data source; the EF data reference, where the second column addresses the reference.
Tables S4a and S4b of the supplementary have also been improved in a similar way with the temporal data source respectively gridmap data source split from the data reference.

**Differences with previous EDGAR datasets**
The authors agree to inform the readers of the difference between the different EDGARv4 datasets. Therefore, section 4 of the Supplementary Information addresses the differences between EDGARv4.3.2 and previous versions v4.2 and v4.1, which have not been documented in a publication but which have been used by atmospheric modellers. The differences shown in Figures S3 are explained by the continuous improvement the EDGARv4 database has gone through since its first release. Such improvements are detailed in the revised manuscript via an explicit reference in the main manuscript at the end of the section 1 "Historical evolution". We wish to stress that the EDGARv4.3.2 is the result of a steady improvement of the EDGARv4 database over more than a decade, also thanks to the feedback of users. In particular we note that:
- For the main differences between EDGARv4.2 and v4.1 we refer to http://edgar.jrc.ec.europa.eu/Main_differences_between_EDGARv42_and_v41.pdf.
- For the main differences between EDGARv4.3.2 and v4.2 we refer to the Supplementary of the paper, section 3 and Table S5 with the findings of studies, using EDGARv4 as input.

**Evaluation of gridded maps**
The authors agree that the spatial distribution is the major cause of the differences in the gridmaps and we therefore propose to include an overview of the improvement in the gridding with the table below (which is included in the revised manuscript, Table S5). Section 4 of the Supplementary refers to the findings of Gately & Hutyra. (2017) and is expanded with the findings of Maasakkers et al. (2016) and Oda et al. (2018) as follows: "Improvement of the spatial distribution of the fossil fuel production emissions in EDGARv4.2 was shown to be

necessary for USA by Maasakkers et al. (2016) and for China by Saunois et al. (2017) and addressed accordingly by extending the dataset with extra point sources for the extraction and mining sites. The importance of point and line source data has been also illustrated by Oda et al. (2018) but needs further observation-based verification."

**Table S5 – Improvement of the spatial proxy data from EDGARv4.2 to EDGAR v4.3.2: indicating those sectors that were considerably improved for the spatial distribution of the sectoral emission totals per country.**

| EDGAR sector | Sector description | Gridmaps v4.2 | Gridmaps v4.3.2 |
|---|---|---|---|
| **Total** | | 114 maps | 297 maps (also sometimes changing over the 42 yr time period) |
| **AGS/ ENF/ MNM** | **Agriculture** | Per type of animal: FAOgeonetwork (2007) | Per type of animal: FAOgeonetwork (2014) |
| | | Per type of crop: FAOgeonetwork (2007) | Per type of crop: FAOgeonetwork (2014) |
| **CHE/ IRO/ NFE** | **Production of chemicals** | In-house EDGAR proxy, gapfilled with **population** as default | In-house EDGAR proxy per type of non-ferrous metal or per type of chemical product, improved with **point source data** from EPRTRv4.2 for Europe, USGS for the rest of the world, gapfilled with **urban population** as default |
| **NMM** | **Cement** | In-house EDGAR proxy, gapfilled with **population** as default | In-house EDGAR proxy, improved **with point source data** from EPRTRv4.2 (2012) for Europe, and CEC (2015) for the rest of the world with global cement shares per province for China and with **urban population** as default |
| **ENE** | **Power industry** | 53981 Power plants (public + autoproducers) from CARMA (2007) without distinction for fuel type | 68931 Power plants (public + autoproducers) from CARMAv3.0 (2012), distinguished per fuel type and with correction for missing or inverted coordinates and additions for China and Russia |
| **PRO** | **Fuel exploitation** | In-house EDGAR proxy, with distinction per fuel type and surface and ground mining of brown and hard coal. | In-house EDGAR proxy per fuel type, improved for coal mining with **point source data** from EPRTRv4.2 (2012) for Europe, and Liu et al. (2015) for China |
| **REF** | **Oil refineries** | In house EDGAR proxy, with gas flaring from **Elvidge (2009),** and gapfilling with **population** as default | In-house EDGAR proxy, improved **with point source data** from EPRTRv4.2 (2012) for Europe, USGS (2014), gas flaring from **NOAA (2015),** oil terminals from World Port (2015), gapfilled with **rural population** |
| **SWD** | **Landfills and waste incinerators** | In-house EDGAR proxy, gapfilled with population | In-house EDGAR proxy, improved **with point source data** from EPRTRv4.2 for Europe, gapfilled with **urban population** |
| **TNR_ Aviation** | **Aviation** | In-house EDGAR proxy with distinction between take-off/landing, climb-out/descend, cruise, supersonic based on **AERO2K** dataset (Eyers et al. 2004) | In-house EDGAR proxy with distinction between take-off/landing, climb-out/descend, cruise, supersonic based on **Airline Route Mapper** (http://arm.64hosts.com/) |
| **TNR_ Ship** | **Shipping** | EDGAR proxy based on Wang et al. (2008) | In-house EDGAR proxy based on Wang et al. (2008) improved with **LRIT** information (Trombetti, 2017) for European seas |
| **TRO** | **Road transport** | Convolution of **roads** of OpenstreetMap **(2008)** and the **population** map for all vehicles except trucks; roads of OpenstreetMap (2008) for trucks | New OpenStreetMap **(2015) with distinction between 4 different road type classes**, used differently for the different types of vehicles (trucks, passenger cars, busses, twowheelers) |

| WWT | Waste water handling | Total population for developed countries, urban population for developing countries | EDGAR proxy improved with point source data from EPRTRv4.2 (2012) and CEC (2015), gapfilled with urban population |
|------|------|------|------|

As proof of the improvement of the road transport spatial distribution proxy, we show in the figure here below the map of the NOx emissions due to road transport making use of traffic volume data for Europe (right) and the EDGARv4.3.2 road transport (left) gridmaps. We do refrain from deriving insights from cell-to-cell differences or ratios between gridmaps, as we experienced that these are not revealing of useful information on where to improve (due to displacements and skewness, for example). We observe that the spatial changes are in the expected direction, with the same patterns in most EU countries (e.g. UK, Germany, Poland) but also differences (e.g. in Italy, where the road transport network between cities needs to be more pronounced). In our view, the magnitude of the improvement can only be assessed and quantified by confronting to the observed data that we want to represent with the spatial distribution.

[Figure]

Fig. 1: Visual comparison of the distribution of road transport emissions, using the EDGARv4.3.2 proxy (based on 4 classes of roads) and using the traffic volume from the Trans-Tools model.

**Hot spot analysis**

Indeed EDGARv4.3.2 is not mechanistically modeling urban emissions, unlike Gurney et al. (2018) but do start with sectoral country totals, to avoid the need of selecting a definition for an urban area. For CO2 emissions, which are dominated by fuel combustion, the national fuel statistics, driving the emissions, are known with a much smaller uncertainty than what is available at city level by e.g. the Covenant of Mayor data in Europe. Whereas subnational or urban emission gridmaps might be more subject to the uncertainty on the activity data that are to be defined as representative for the local area, the uncertainty in the EDGAR gridmaps is mostly determined by the assumptions on the representativeness of the selected spatial distribution proxy for the entire country.

With increasing granularity of the spatial distribution per sector and using where point source data, this uncertainty reduces. EDGARv4.3.2 uses 297 distinct datasets for the different subsectors. At least for Europe, the authors believe that the hot spot analysis remains useful, in particular because of the use of the many point sources for any industrial or commercial activity. The authors would assume a better spatial representation of the emissions than what is obtained with e.g. the CCFFDAS model or even with the ODIAC model, but the validity can only be proven by an observation-based verification with e.g. space-borne XCO2 measurements as a next step. The authors realise that this goes hand in hand with the improvement of the temporal profiles and want to refer to the recent work we submitted to ESSD by Crippa et al. (2019)[1].

**Policy application**

The European Commission (EC)'s in-house global emissions database EDGAR is, since more than a decade, known and used by the EC's Directorate General Climate Action (DG CLIMA). As such, DG CLIMA has been using emission estimates for world regions/countries in preparation of the climate negotiations at the COP (e.g. presentation of Director General Jos Delbeke in 2012, Staff Working Documents in 2014 and successive years). Most recently DG CLIMA is increasingly asked to look into subnational emission inventories, such as inventories at urban or province-level scale. These can provide actionable information on the implementation of local GHG reduction measures.

For the readiness level of the gridded emissions for policy application, the authors refer to the air quality. The air quality (in Europe addressed by a first directive in 1970) and transboundary air pollution (addressed under the UNECE Convention on Long-Range Transboundary Air Pollution - CLRTAP of 1979 (in force since 1983) focused in a first step on emission inventories of air pollutants and the monitoring of the time-series. In a second step, gridmaps were requested and nowadays the Parties need to provide these at 0.1deg resolution on an annual basis. The European Commission – Directorate General Environment appreciated the delivery of default emission gridmaps for the European Commission Directive on the Pollutant Release Transfer Register (E.PRTR) and supported EDGAR with extra funding for further use of the emission gridmaps by the CLRTAP Task Forces of Emission inventories & Projections – TFEIP and of Hemispheric Transport of Air Pollution – TF HTAP. Nowadays, as one of its activities the Copernicus Atmospheric Monitoring Service[2] assesses the bottom-up gridded emissions (and in particular local exceedances of pollution levels) with top-down measurements.

**Quantitative information on CARMA corrections**

The authors agree that the power plants are very important point sources and that the CARMAv3.0 dataset has been carefully screened with an internal QA/QC procedure to avoid large errors. We are not allowed to disclose the CARMAv3.0 dataset, because that is not our proprietary and unfortunately no longer online available. For the sake of transparency, we
* * *
[1] Crippa, M., Solazzo, E., Huang, G., Guizzardi, D., Koffi, E. Muntean, M., Schieberle, C., Friedrich, R., Janssens-Maenhout, G.: Towards time varying emissions: development of high resolution temporal profiles in the Emissions Database for Global Atmospheric Research, Earth Syst. Sci. Data Discuss., https://doi.org/10.5194/essd-2019-47, in review, 2019

[2] https://atmosphere.copernicus.eu/

summarized in the table below the different steps undertaken to convert it to our EDGRv4.3.2 spatial proxy dataset for the power sector.

| Steps on the data source | Description of the content |
|---|---|
| CARMAv3.0 | 68931 power plants |
| Corrected CARMAv3.0 | 200 points have been corrected for the missing or inverted coordinates
For China 1200 points have been added manually with internal resources
For Russia 50 points have been added manually with internal resources |
| Resulting EDGARv4.3.2 proxy for power plants | All points are aggregated to 0.1x0.1 cells. These are in total 16931 cells, of which:
4610 cells are defined for Auto-producers
5199 cells are defined for COAL
3304 cells are defined for GAS
3818 cells are defined for OIL |

**2. Line by line comments**
*Manuscript*

P1, L20: The wording "disaggregated" is replaced with "broken down" to avoid confusion with the spatial disaggregation.

P1, L25: The wording "fully traceable", has been replaced by "transparent to the extent possible", to avoid false expectations. (Underlying datasources such as activity data and proxy data can not be shared, as these are not proprietary of JRC.

P1, L25: The wording "IPCC-based methodology" is replaced by "IPCC-compliant methodology", as suggested.

P1, L26-: The authors agree that terms "short_cycle" and "long-cycle" are helpful for the EDGAR users, but prefer to introduce these in section 1.

P3, L29: The authors do think so. Arguments are given under "policy application" paragraph above. Limitations are addressed in section 6.2 of the manuscript.

P3, L35. Footnote 5 is taken up in the main text as suggested.

P4, L5: The authors agree and added "emission disaggregation" to this sentence.

P5, L6: EDGAR provides bottom-up inventories for any activity within the territory (conform IPCC GL). The boundaries of the territories are the current political boundaries of the countries, as delineated by the European Commission (http://publications.europa.eu/code/en/en-5000500.htm). While Germany is the aggregate of West and East Germany, a split-up was

needed for the former Soviet Union and former Yugoslavia. The pre-1990 data were allocated to the countries using the same share of sector- and fuel-specific country shares in the activitiy data from the countries in 1990. The authors refer to the footnote 1 of Table 1b.

P5, L14: The authors confirm that the emission seasonality is not country-specific in EDGARv4.3.2.

P5, L15: We did use the three different reference years of the CARMAv3.0 datasets (2004, 2009 and "future-2014"). For the intervening years, we used the reference year that was the closest. The total emission was distributed making use of the intensity parameter, given by CARMAv3.0. The authors refer to the explanation in section 3 of the Supplementary. For more details on the QA/QC and the corrections made, we refer to the "Quantitative information on CARMA corrections" paragraph above.

P6, L2: EDGARv4.2 has included Large-Scale Biomass burning (incl. post-burn effects) for forests from GFED (Van der Werf et al., 2010) and for peat land from Joosten (2009)[3]. This has led to confusion in the IPCC AR5, because this subset covered only partially the emission sources of the land use, land-use change and forestry, whereas it was compared to datasets which covered the LULUCF sector in a more comprehensive way. Petrescu et al. (2012) calculated the emission sources and sinks of the forest land (remaining forest land) with gains (from forest growth) and losses (from harvest, net deforestation and fires) following IPCC (2003) and (2006) methodology but had to conclude that the different losses can not be superposed without risk of double-counting. Careful top-down analysis would be needed to resolve this problem, which is outside the scope of the bottom-up inventory of EDGAR. Therefore it was decided for EDGARv4.3 not to cover the LULUCF sector, but the agricultural field burning and the crop waste burning, based on the agricultural area and crop yield as activity data remain included.

P6, L31: The authors meant indeed to say that EDGAR follows a Tier 1 or Tier 2 approach of the IPCC (2006) Guidelines with sub-activity data and avoid modeling the processes. The reasons are the missing parameters for some world-regions and the consistency of applying one selected Tier level for all world countries in EDGAR.

P7, L5: Temporal profiles in EDGAR have been developed in 2010 for the FP6 and FP7 research projects CIRCE and PEGASOS, because the global air quality models needed monthly disaggregated air pollutant emission gridmaps as input. The temporal profiles are a first rough bottom-up estimate of the temporal variation for major sectors, based on the insights of regional air quality models. The authors added this as underlying "narrative" to section 2.3, but refer to the recent work on temporal profiles, that was submitted to ESSD by Crippa et al. (2019)[1].
* * *
[3] Couwenberg J, Dommain R, Joosten H (2009) Greenhouse gas fluxes from tropical peatlands in south-east Asia. Global Change Biology, doi: 10.1111/j.1365-2486.2009.02016.x

P7, L11: Huang et al. (2018) is a report as deliverable for a contractual work. The results have been taken up in the recent publication submitted to ESSD of Crippa et al. (2019). The authors exchange the reference.

P7, L11: The authors agree that the temporal profiles of Andres et al. (2011) are based on a different approach and deleted this reference to avoid confusion.

P7, L15: The authors agree and deleted the word "global".

P7, L18: The authors corrected the term "linear" by "line", as suggested.

P7, L21: The area average for a grid cell with point sources only changes when the point sources for that grid cell change.

P7, L25: The authors agree and exchanged "reported" with "likely located".

P7, L27: The authors are aware that Oda et al. (2018) also uses CARMA for point sources, but we aimed to indicate some alternatives for the other sources. We changed the sentence as follows: "Alternatives can be night light satellite data, as used by Oda et al. (2018) for those emission sources that were not yet covered with point sources (such as the power plants).

P7, L28: The authors agreed to add "sectoral" before "emissions".

P7, L30: The authors agree that the uncertainty analysis for the proxy data themselves are not useful for inverse modeling and/or data assimilation and consider a full analysis of the representatives of the 297 selected proxy datasets beyond the scope of this paper. The sensitivity of the proxy data and the evaluation of grid maps are ongoing, also for the H2020 research projects CHE and VERIFY. The authors want to refer here to the H2020 research project CHE and move the explanation of the CHE project from the next section up to this section. Since we are not in the position to quantify and assess the magnitude of improvement of the different spatial proxies used (like for all inventories, we imagine), we stress the fact that the continuous improvements the EDGAR' s mapping has gone through, has been mainly dictated by policy and scientific demand. An indirect evidence of the improvement of the mapping comes from the applications to air quality model evaluation activities and global modelling. Indeed one of the benefits and strengths of EDGAR is its transversal support to regional and global modelling communities as well as to climate and policy. Recent modelling activities (AQMEII, HTAP among others) have reported improvements for EDGARv4 inputs, which were taken on board in an iterative and informal way and reported significant improvements of model performance.
In addition, we are not familiar with the literature the reviewer refers to in this instance. If the reference is to the last paper suggested in the references list (Oda et al., 2019, accepted for Mitigation and Adaption Strategies for Global Changes), we note that we have no access to that specific manuscript.

P7, P31: The authors introduced the CHE project in the following section dedicated to uncertainty and propose these explaining sentences to the section 2.4 upfront.

P8, L5: The authors agree that this sentence does not fit too much here and propose to move these sentences up to section 2.4.

P8, L11:  The eq. (4) provides the uncertainty for the CO2eq as reported in policy documents for the Paris Agreement and include all three gases (neglecting the F-gases). For the sake of clarity, the unit CO2eq has been added.

P9, L3: The authors agree and aimed to point to the difference: by adding the small sources on waste incineration, urea and liming activities but with much larger uncertainty than the major source on fossil fuel combustion, the resulting uncertainty is a higher (9.0% instead of 8.4%).

P9, L12: The authors take note of this and put "Note Andres et al. (2016) limited the result by saying CASE FOR CDIAC" as footnote.

P9, L17: The authors extended section 2.4 on an assessment of the representativeness of the 297 spatial proxy datasets and the assessment of the representativeness in the CHE and VERIFY projects. A reference to the previous section is taken up here.

P9, L28: The authors agree to delete "and are of prime interest to scientists". However, the enhanced transparency framework for the Paris Agreement was set up with the publicly available bottom-up inventories for all world countries and their NDCs. Only in 2017 the SBSTA recognized officially the use of Earth observation data for assessing the bottom-up inventories.

P9, L35: The authors agree that inverse modeling is also limited and added the following sentence: "Although the posterior feedback on the prior emission gridmaps is very useful, it remains limited because of the uncertainties related to the transport model, the atmospheric chemistry model, the meteorology input and the in-situ or space-borne observations.

P10, L1: Even though the different gases CO2, NOx, SO2, and CH4 behave differently in the atmosphere, they all need to be transported by an atmospheric model using meteorological input. The sources of emission for the first three gases show similarities: they are co-emitted in the case of fossil fuel combustion, at high temperature, when the fuel is coal or heavy residual fuel oil. The location of the point sources for all three gases should be therefore consistent. For CH4 the emission sources are quite different from CO2.

P10, L21: The reference for the EDGARv4.2 flaring gridmap, Elvidge et al. (2009)[4], has been replaced with the new reference: NOAA – NGDC (2015) [5]. We followed the indications on
* * *
[4] DMSP data collected by the US Air Force Weather Agency
[5] Image and Data processing by NOAA's National Geophysical Data Center

https://www.ngdc.noaa.gov/eog/viirs/download_viirs_flares_only.html to extract the locations of flaring from all the light sources.

P12, L14: The term "verify" has been selected in consultation with our policymakers at the DG CLIMA. The authors propose to insert the following footnote: "The term *verify* is selected in consultation with the EC policymakers for Climate and refers to the detection of biases in emission inventories."

P15, L7: While the representativeness of the selected proxy data needs further evaluation, the authors consider this beyond the scope of this paper (In this paper we wanted to focus on documenting v4.3.2 with activity data, emission factors, proxy data). Instead the authors provides a warning for the sensitivity of the proxy data on the results and refer to section 2.4 for the assessment of the representativeness of the proxy data and to section 4 of the supplementary (with extra Table S5) for the continuous improvements made from the EDGARv4 database, tripling the number of proxy datasets and addressing the issues with e.g. road transport distribution as pointed out by Gately & Hutyra (2017).

P16, L2: We refer to the "Quantitative information on the CARMA corrections" paragraph here above.

P18, L2: The authors are grateful for that.

P18, L22: The authors agree to rephrase the sentence as follows: "Although modelling uncertainties and the uncertainties of natural emissions remain large, the atmospheric models provide observationally constrained top-down input and it is expected that inverse models increasingly contribute to the independent verification of the total fluxes." The authors want to refer to the fact that in the IPCC 2019 refinement of the 2006 Guidelines (forthcoming May 13[th], 2019), an extra chapter in volume 1 is taken up on inverse modeling and its use for national GHG inventory reporting

P19, L7: The authors confirm that they did not want to limit the Earth observation data to satellite imagery only, but in fact, as shown by Bergmaschi et al. (2015) the in-situ ground stations are even of larger importance for constraining the prior emissions, in particular for N2O. Therefore the authors propose to replace "satellite data" with "space-borne or in-situ Earth observation data".

P19, L11: EDGAR does include regional specificity (cfr. the 24 geographical groups), but not subnational specificity (because the activity data is national). The authors confirm that this is consistent with the first bullet point at L15.

P19, L17: The authors agree to exchange "disaggregation" with "downscaled".

P19, L20: The authors agree that this gapfilling comes at the expenses of consistency loss and propose to add this with the following additional sentence. "These gapfillings come at the expense of losing consistency within the reported emissions as inventory."

P19, L31: This is indeed a general statement, of less relevance for CO2, but of significance for some CH4 and N2O emitting subsectors (e.g. CH4 from coal mining at surface or underground, N2O from nitric acid processes).

P19, L36: The authors refer to the section on "Quantitative information on CARMA corrections" for the power plant point sources.

P20, L9: Ideally the point sources should be allocated the emissions accounted for that point source, but we do not have a global point source database for all industrial facilities (such as EPRTR for Europe). When using CARMA, the total emissions need to be distributed over the different point source locations, using a parameter, such as emission intensity, capacity of the plant etc. Ideally this should be complemented with the real share of the plant in the national total emissions and its temporal profile. More global point source data is needed for improving EDGAR here.

P20, L31: The strength of EDGAR is that it calculates emissions for GHG and for air pollutants using consistently the same activity data and the spatial proxy data. As such co-emitted species are represented with one single multi-pollutant source. The CHE project is investigating how these ratios (changing over time and space) can be used to characterize and extract the fossil fuel signal from the total in the space-borne observations.

P21, L1: Data availability section is improved with a short overview table of the data.

P40, Table 3. The authors took into account the ODIAC timeseries 2000-2016, that was documented in Oda et al. (2018) and we took the year 2010. Indeed, different release of energy statistics as basis for the emission calculations might be the reason for the difference. We took this up as small footnote to the table (not only for ODIAC but also for BP).

P48, Figure 10a and 10b: High quality figures will be uploaded and delivered for the final paper.

P49: For the road transport emissions, the authors would like to refer to the section "Evaluation of gridded maps", illustrating how the current EDGARv4.3.2 spatial distribution is calibrated to traffic volume. We do not have volume traffic for the rest of the world. We experienced that the previous proxy using a convolution of roads and population did concentrate the emissions too much within the cities, which has been reported also by e.g. Gately & Hutyra (2017). For the sake of transparency, we provide here below the requested difference of CO2 (long-cycle carbon) emissions of road transport for 2005 produced in EDGARv4.3.2 and EDGARv4.2. The changes are not negligible, but we refrain from such analysis and do not propose this for the paper, because we can only give confidence for Europe that the changes went into the right direction with our comparison to the traffic volume.

[Figure]

Fig. 2: Difference of EDGARv4.3.2 and EDGARv4.2 emissions of CO2 (long-cycle carbon) for road transport in 2005.

*Supplement Information*
The authors confirm that technology-specific data are only used for air pollutant emissions (cfr. Crippa et al., ESSD 2018).

P2, L2: This comment was also raised for P5, L6 and explained there. The Authors refer to footnote 1 of Table 1b.

P14, P4: EDGARv4 does not interpolate the population gridmaps (total, urban and rural) over time. The 1990 population gridmaps were used from 1970 to 1992 and for the more recent years, we used the population gridmaps for the reference year that was the closest.

P14, L13: For the QA/QC and the corrections to the CARMAv3.0 point sources, the authors refer to the "Quantitative information on CARMA corrections" paragraph. The EDGAR team did not sufficiently dispose over resources to check the commissioning and decommissioning of all plants.

P15, L4: The NOAA gas flaring nightlight data maps for 1992-2000-2006-2014 were used with the reference year that is closed by for all intervening years. The spatial proxy data were gapfilled with rural population for those countries that were not covered.

P15, L12: These four times six weighting factors (for the 4 different types of roads and the 6 different types of vehicles) are EDGAR specific and calculated such that in Europe the distribution represents well the traffic volume (available for Europe via the EC in-house data of the Trans-Tools transport model). No population data was any longer used. (This makes the big difference between the new gridmaps of road transport and the old ones. We experienced that the road transport emissions inside urban areas can not be scaled with population data.) The authors refer to the "evaluation of gridded maps" paragraph above.

P15, L24: The authors correct this typo and inverted it to 101 km.

P15, L32: The authors confirm that Friedrich and Reis (2004) is not to most suitable reference to the temporal profiles used for the GENEMIS inventory, which was input to the LOTOS model for assessing air pollution (acid rain). The authors exchanged the reference with Lenhart & Friedrich (1995)[6].

P15, L33: This comment was also raised for P7, L5. The authors confirm that the temporal profiles are mainly based on data from the air quality community in Europe. We refer to the additions in section 2.3 and the recent work on temporal profiles, that was submitted to ESSD by Crippa et al. (2019)[1].

P21: L2: The CARMAv3.0 base years are 2004, 2009 and "future", which we all used without interpolation for intervening years, but selecting the closest reference year (2004, 2009 and future=2014).

P21, L12: The authors agree that this is very much forward looking and deleted the sentence.

P21, Figure S2a shows the temporal profiles in EDGARv4.3.2, which are applied to all 42 years without change in the northern hemisphere. These are indeed based on European sectoral profiles from air quality models.

P22, Figure S2b shows a comparison of the temporal profiles, which are again as default applied for any year in the northern hemisphere for the emissions from the energy and industry sector (large scale fuel combustion). For more details on the temporal profiles we refer to the respective papers of each of the datasets cited in the caption of the figure.

P22, L6: The section 4 of the Supplementary has been expanded with a comparison of the spatial proxy dataset used in EDGARv4.2 and EDGARv4.3.2.

P23, L15: The authors use CARMAv3.0 for the allocation of the national totals for the energy sector using a share for each point source within the country, derived from the intensity given by CARMAv3.0. As such no further adjustment is needed afterwards.

The authors are interested in a copy of the new paper:
Oda et al.: Errors and uncertainties in a gridded carbon dioxide emissions inventory, accepted for *Mitigation and Adaptation Strategies for Global Change, 2019*
* * *
[6] Lenhart, L., Friedrich, R., European emission data with high temporal and spatial resolution, Water, Air, and Soil Pollution, Vol. 85, Issue 4, pp. 1897-1902, https://doi.org/10.1007/BF01186111, 1995

---

## Author Comment (AC3) · 29 Apr 2019

Dear Dr. Andrew (dear Robbie),

We would like to thank you for the positive and constructive review. We reply here underneath to the comments you raised point by point.

Regarding the comments raised: Page 4, lines 19-25: The authors agree to include a simple opening sentence and propose: "Annual country-specific emissions are calculated using international activity data and emission factors, updated according latest scientific knowledge and following IPCC (2006) methods." The next sentence starts

then with: "Emissions (EM) ..." (PS: We do distinguish between the term "national = made within the country" and "country-specific = representing the country-territory". This terminology was agreed with the European Commission Directorate-general Climate Action: for which the national inventories are only received by the countries, whereas EDGARv4 is providing country-specific inventories, estimated with international statistics and emission factors, representative for a given territory".)

Page 4, line 29: "Uncontrolled" means: without end-of-pipe abatement. Formula (1) is the standard formula for the emission calculation of any gas, GHG or air pollutant, in EDGAR. While for GHG there are no end-of-pipe abatements active. for air pollutants the end-of-pipe abatement is significant (e.g. filters, catalysts, etc.). In EDGAR the abatement measures as well as the emission recovery (e.g. coalbed CH4 recovery) is calculated explicitly. This provides our policy makers information on the effectiveness of emission reduction measures.

Page 5, lines 7-8: The authors agree to remove the sentence.

Page 5, line 14: The authors agree that "latitudinal region" should be replaced by "latitudinal band".

Page 5, line 18: The authors prefer to keep "energy-related" sectors. These include also the non-energy use of fuel, the petrochemical sector and the manufacturing industry (in particular iron and steel). While the energy sector is managed by the European Commission's Directorate General (DG) Energy, the non-energy industry is also managed by other DGs, such as DG Internal market, Industry, Entrepreneurship and SMEs.

Page 5, line 18: Table 1a: The authors agree to introduce also this table as follows: "Table 1a provides a structured overview of all the emission sources included in the EDGARv4 database."

Page 6, line 35: The 1996 codes were still used, because the national GHG inventories were still reported in 2014 (for the time series 1990-2012) using the 1996 codes. Only

after the reporting of the first Kyoto Protocol period the codes were changed into the 2006 codes.

Page 8, line 6: The project name has been corrected to "CO2 Human Emissions".

Page 8, line 13: The sentences have been rephrased with "the 24 member countries of the OECD in 1990 (24OECD90) and 16 countries with Economies in Transition of 1990 (16EIT90)".

Page 8, line 24: The authors indeed mean: that it is difficult to estimate reliable biofuel combustion activity data and so CO2 emissions. Although the article of Denier van der Gon et al. (2015) does not focus on CO2 but on PM, the energy statistics (fuel wood used) are provided in Table 2, and are directly proportional to the CO2 emissions. Denier van der Gon et al. (2015) concluded "that the wood-burning emissions are much higher than could be accounted for with the emission inventory available at the time". Moreover he indicates that "this is only partly related to emission factor measurements", but that underreporting (of activity data) has been noted, also by EEA collecting national inventories for Europe. The authors agree to rephrase the sentence as follows: "While Denier van der Gon et al. (2015) indicate that biofuel combustion activity (and corresponding short cycle carbon CO2) is difficult to estimate for the different countries in Europe, Tian et al. (2015) estimate the large uncertainties in CH4 and N2O budgets."

Concerning Elvidge et al. (2009): The reference for the EDGARv4.2 flaring, Elvidge et al. (2009), has been replaced with the new reference that is used in EDGARv4.3.2: NOAA – NGDC (2015). We did also consult Elvidge et al. (2016) for the quantification of the venting and flaring trend in the last decade.

Comparison with the IEA emission estimates: This comparison is indeed done with the old (2014) edition of the IEA dataset (as given in the reference), because EDGARv4.3.2 is also based on that and most national GHG inventories time-series till the first Kyoto Protocol period were based on that too. The difference in the calculation for IEA and

EDGAR are indeed the old carbon factors of IPCC (1996) that was used by IEA, while EDGAR has been using always IPCC (2006) carbon factors, but there are also other differences. EDGARv4 supplements the charcoal production activity with fuelwood data of FAO, the venting and flaring activity with satellite data, and the fossil fuel mine gas recovery with UNFCCC data. In addition EDGARv4 calculates the transformation losses which IEA neglects. The authors agree to include a clarifying note to Table 3 on the update of the IEA carbon factors that meanwhile happened.

Concerning the description of the paper as "full, transparent and inclusive documentation": Although the EDGARv4 emission data set with calculation method is - to the extent possible - provided in a transparent and inclusive manner, the underlying activity data and spatial proxy data sets cannot be released due to property rights. Moreover the EDGARv4.3.2 is the result of iterative improvements over almost a decade, which makes it more difficult to keep full transparency. While the EDGARv4.3.2 gridmaps remain replicable within the EDGARv4 database, this can not be said for EDGARv4.2 and EDGARv4.1, because of the large updates and expansion of the proxy datasets (from 114 datasets for v4.2 to 297 for v4.3.2). However, this article should lay the foundation from which further updates document the newer releases of EDGAR. In order not to keep away from false expectations, we propose to end the sentence slightly rephrased as "complete documentation of the EDGARv4 products that has been compiled in the most transparent way possible."

References used: NOAA – NGDC: Image and Data processing by NOAA's National Geophysical Data Center, https://www.ngdc.noaa.gov/eog/viirs/download_viirs_flares_only.html, latest access 2015.

Elvidge, C.D., Zhizhin, M., Baugh, B., Hsu, T/-C., Ghosh, T.: Methods for Global Survey of Natural Gas Flaring from Visible Infrared Imaging Radiometer Suite Data. Energies, Vol. 9 (1), p. 14, doi:10.3390/en9010014

We are finalising the revision of the paper and will upload this tomorrow, after a final check that all comments are included. In addition, I would like to raise that we will update the formula on the uncertainty with the comment you raised by email, in order to avoid any misunderstanding with the absolute uncertainty (in $CO_2eq$). The Table 2 of relative uncertainties remain unchanged.

Thanks very much for your interest and continued feedback. Best regards, Greet.

Please also note the supplement to this comment:
https://www.earth-syst-sci-data-discuss.net/essd-2018-164/essd-2018-164-AC3-supplement.pdf